# Insight predicts subsequent memory via cortical representational change and hippocampal activity

**Maxi Becker** [1,2] ✉, **Tobias Sommer**[3] **& Roberto Cabeza**[1,2]

The neural mechanisms driving creative problem-solving, including representational change and its relation to memory, still remain largely unknown. We focus on the creative process of insight, wherein rapid knowledge reorganization and integration—termed representational change—yield solutions that evoke suddenness, certainty, positive emotion, and enduring memory. We posit that this process is associated with stronger shifts in activation patterns within brain regions housing solution-relevant information, including the visual cortex for visual problems, alongside regions linked to feelings of emotion, suddenness and subsequent memory. To test this, we collect participants' brain activity while they solve visual insight problems in the MRI. Our findings substantiate these hypotheses, revealing stronger representational changes in visual cortex, coupled with activations in the amygdala and hippocampus—forming an interconnected network. Importantly, representational change and hippocampal effects are positively associated with subsequent memory. This study provides evidence of an integrated insight mechanism influencing memory.

Creativity and memory have a synergistic relationship. Finding creative solutions depends on having relevant preexistent knowledge, and, conversely, novel solutions are typically well-remembered, promoting new knowledge of the world and future adaptation. *Insight* is a fundamental process in creative problem-solving that occurs when a problem is solved via a novel approach by reorganizing preexistent knowledge structures[1,2]. Insight problem solving, which is often studied in the context of scientific discoveries[3], has been associated with boosted subsequent long-term memory, sometimes after a single experience, unlike other forms of learning that require multiple repetitions[4–8]. The study of insight can shed light on mechanisms of learning and memory formation, and has practical implications in educational settings, as insight-driven learning experiences can be effective in improving knowledge retention[9,10].

This raises the question of how the insight process works and why it is often associated with better memory. The insight process has cognitive and evaluative components. The main cognitive component

is associated with *representational change* (RC) whereby internal (conceptual or perceptual) representations of the solution are reorganized and integrated into a coherent whole[11,12]. This process is assumed to involve a change in the pattern of currently activated cell assemblies coding the problem in domain-specific cortex, but neuroscientific evidence is currently lacking[13,14]. As an example, in verbal problem-solving, RC may encompass semantic reinterpretation or the regrouping of conceptual relationships[15–17]. In contrast, for visual problems that demand object recognition, RC may involve breaking up perceptual chunks into their constituent elements to make them available for a novel recombination[13,18,19]. Multivariate functional magnetic resonance imaging (fMRI) can identify representation-related activation patterns in regions such as visual cortex[20,21], which can be used to detect RC.

The evaluative component (or *AHA! experience*) is characterized by positive emotions and the conviction that the solution arrived suddenly and is certainly correct[7,22]. Note, this component involves

¹Humboldt University Berlin, Department of Psychology, Berlin, Germany. ²Duke University, Center for Cognitive Neuroscience, Durham, NC 27708, USA. ³University Medical Center Hamburg-Eppendorf, Institute of Systems Neuroscience, Hamburg, Germany. ✉e-mail: maxi_becker@gmx.net

cognitive aspects as it assesses the solution and has been associated with activations in amygdala and hippocampus[23–27]. Activation in the amygdala has been suggested to play a role in processing positive emotions[28], and the detection of novelty or surprise, such as during the sudden arrival or unexpected content of a solution, has been associated with hippocampal activation. It is well-known that novelty[29,30] and surprise[31–33] involve the hippocampus, particularly its anterior formation. Importantly, at the time of insight, cognitive and evaluative elements are unlikely to function in isolation but rather in a closely interconnected manner, facilitating the efficient processing of solution-relevant information[12]. Finally, the neural mechanisms of this *insight-related better memory* remain largely unknown[8,34]. However, both insight components likely contribute to this effect, as indicated in behavioral studies[4,6,8,35,36]. There is evidence that the quality of visual cortex representations during the perception of visual stimuli predicts later memory[37], and the fundamental contributions of hippocampus in declarative memory encoding has been continuously demonstrated (for reviews[38,39]).

In sum, insight during problem solving likely reflects stronger multivariate pattern changes associated with enhanced solution-relevant representations. This increased RC (or conceptual update) is efficiently integrated into the solution process—a series of cognitive steps from problem representation and active visual or memory search to subsequent solution retrieval[11]—leading to awareness of the solution. This realization is often accompanied by an emotional and suddenness/surprise response in hippocampus and amygdala resulting in enhanced encoding of the solution and better subsequent memory.

To investigate the neural mechanism of (visual) insight and its association with subsequent memory, the current study scanned participants using fMRI while they performed a visual insight task. In this task, participants attempted to identify (solve) difficult-to-recognize, high-contrast images of real-world objects (*Mooney images*[40], see Fig. 1A, B). Due to ambiguous figure/ground separation, black/white contours in these images are often grouped into meaningless shapes, hindering successful object identification. Perceptually rechunking or regrouping these contours can lead to sudden object identification, eliciting a type of insight[10,25]. Following Mooney object identification, participants provided ratings of the perceived suddenness, emotion, and certainty about the solution—which were used to index insight—and then selected the semantic category of the hidden object. Five days later, they performed a *recognition task* with confidence measures, and then generated the name of the hidden object.

To evaluate this neural mechanism of (visual) insight and its association with memory, we tested four specific fMRI hypotheses. First, ventral occipito-temporal cortex (VOTC) exhibits visual insight-related RC. We focused on six VOTC regions-of-interest (ROIs) associated with visual object recognition (posterior and anterior fusiform gyrus–pFusG, aFusG; inferior lateral occipital complex–iLOC; anterior, mid, and posterior inferior temporal gyrus–aITG, mITG, pITG)[41]. RC was measured during Mooney object identification by comparing multivariate activity patterns before and after the solution was found, and determined if the pattern change was greater with increasing insight.

Second, the amygdala and hippocampus show insight-related activity. We examined if at the time of the solution, activity in these regions correlated with increasing insight.

Third, solution-relevant information is more efficiently integrated by a "solution network" consisting of VOTC, amygdala, and hippocampus during insight. Given that the entire VOTC visual hierarchy contributes to object identification[42] and they are all closely interconnected, we employed the whole VOTC in this analysis. Efficient information integration was quantified via functional connectivity of the solution network as well as different graph measures assumed to be higher with high versus low insight trials.

Finally, we hypothesized that both insight components (cognitive and evaluative) are associated with better subsequent memory. In particular, we expected that RC (cognitive) in VOTC and hippocampal activity (evaluative) predict insight-related better memory. Insight-related memory was identified by behavioral and neural measures that showed an insight*memory (remembered vs. forgotten) interaction.

## Results

All reported inferential statistics (*Chi²*) are based on nested model comparisons between two (general) linear mixed models where a baseline model is compared to a full model incorporating the independent variable of interest. For testing interaction effects, the baseline model additionally incorporates two independent variables, while the full model adds their interaction. Effect sizes—either odds ratios (OR) for binary outcomes or standardized beta estimates (ß) for continuous outcomes—correspond to the regression estimate of the respective independent variable or interaction between two or three independent variables in the full model.

### Behavioral results—insight predicts better subsequent memory

A final sample of $N = 31$ participants was analyzed. In line with previous research, insight during the Problem Solving Task (Fig. 1B) was quantified using the three 4-point insight ratings following Mooney object identification[5,22], namely positive emotion ($M = 2.8$, SD = 0.41), suddenness ($M = 3.0$, SD = 0.39) and certainty ($M = 3.16$, SD = 0.34). However, due to their significant factor loadings onto a single latent factor ($\lambda = 0.65$–0.53; $p$s < 0.001, see Fig. S2 in the Supplementary Methods[22,43], the three ratings were combined to one insight measure. All statistical analyses (except for functional connectivity and graph analyses) were conducted using a continuous insight measure, which is the sum of suddenness, emotion, and certainty ratings unless specifically mentioned. However, for better interpretability, we present the descriptive data using a median split of the sum of these three ratings (range: 3–12 points). This categorization distinguished between high insight trials (HI-I) and low insight (LO-I) trials.

On average, participants indicated a solution for 68.4% (SD = 16%) of all presented Mooney images. The solution, i.e., the subsequently selected category of the object, was correct in 43.0% (chance adjusted, SD = 13%) of all cases. Of all correctly identified Mooney images, 65.3% were solved with HI-I and 34.7% with LO-I trials (SD = 8%). The full model including insight and run order as predictor for accuracy along with random item and subject intercepts was significantly better than the baseline model without insight (*Chi²*(1) = 94.98, $p < 0.001$, OR = 1.31, 95% CI [1.63, 2.09]). This suggests that accuracy for HI-I was significantly higher than for LO-I, consistent with previous results[44,45]. Of all incorrectly identified Mooney images, 40.5% were solved with HI-I and 59.5% with LO-I trials (SD = 21%, see amount of trials for all conditions in Fig. S4 in the Supplementary Methods).

The median response time for correct solutions was 3.5 s (SD = 0.8 s) where participants were faster during HI-I (2.7 s, SD = 0.81 s) than LO-I solutions (5.1 s, SD = 1.12 s). This difference was significant (*Chi²*(1) = 423.8, $p < 0.001$, ß = −0.22, 95% CI [−0.24, −0.21]), as shown by the significantly improved fit of the full model—which included insight and run order as predictors for logged response time, along with random subject and item intercepts—compared to the baseline model without insight. The median response time for incorrect HI-I solutions was 3.4 s (SD = 1.2 s) and 6.05 s (SD = 1.1 s) for LO-I solutions (see Fig. S3, Supplementary Methods). Response time and accuracy has been shown to covary with insight solutions[46]. We observed a systematic difference in response times and accuracy between the HI-I and LO-I condition but were not interested in differences in difficulty. Consequently, only accurate responses and response time, treated as a covariate of no interest, were incorporated into all subsequent analyses.

Subsequent memory of each Mooney object identification was measured 5 days later. Participants recognized 67.0% (SD = 15%) of the 120 presented Mooney images, regardless of whether they had

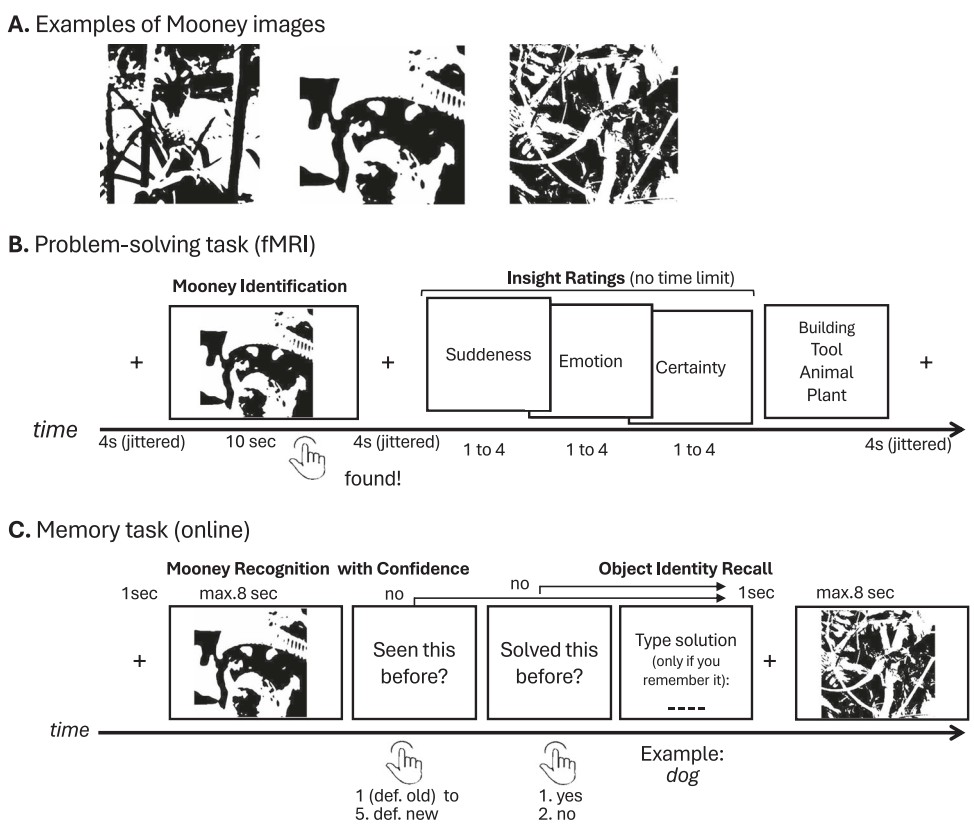

**Fig. 1 | Experimental design. A** shows examples of Mooney images (from left to right: dog, knife, snail, doorknob, spider, tombstone). **B** describes the Mooney paradigm presented in the scanner. **C** illustrates the subsequent memory test 5 days after the solving phase in the MRI.

correctly identified them, with a false alarm rate of 19.3% (SD = 14%). Due to the problem solving context, we were primarily interested in memory for the solution (i.e., object identity recall) but in order to recall the Mooney images, participants had to correctly recognize the image and to remember having solved it first (see Fig. 1C). Hence, subsequent memory for object identity was defined as correct recognition and memory of having solved it and memory for the solution, i.e., the name of the depicted object. Out of the 120 presented Mooney images, participants correctly solved and subsequently correctly recognized 36.0% (chance adjusted, SD = 10%). Out of all presented Mooney images, participants correctly solved and subsequently correctly recalled having solved them in 32.7% (chance adjusted, SD = 11%). Finally, out of all presented Mooney images, participants correctly solved and subsequently correctly identified them by name in 29.2% (chance adjusted, SD = 11%). Median response time was 2.57 s (SD = 0.97) for all old Mooney images during the subsequent memory test. Images correctly identified via HI-I in the scanner were recognized in 1.93 s (SD = 0.77) and via LO-I in 2.39 s (SD = 0.86). Median response time was 3.73 s (SD = 2.0) for all new Mooney images.

To investigate behavioral insight-related better memory, we estimated the association between insight and subsequent memory using two binomial linear mixed model comparisons with a respective full model including insight and run order as a predictor along with random item and subject intercepts and a corresponding baseline model without insight. For the first model comparison, the full model predicted subsequent memory including a categorical insight factor consisting of not solved trials, correctly solved HI-I and LO-I trials. The full model was significantly better than the baseline model indicating that the insight factor significantly predicted variance in subsequent memory ($Chi^2$(2) = 666.49, $p < 0.001$, see Fig. 2A, B). Post hoc analyses revealed that HI-I (OR [odds ratio] = 7.78, 95% CI [6.42, 9.53]) and LO-I (OR = 3.72, 95% CI [3.12, 4.44]) trials were significantly better

remembered compared to unsolved trials. Critically, HI-I trials were also significantly more likely to be remembered compared to LO-I trials (OR = 2.10, 95% CI [1.80, 2.45]), providing evidence for insight-related better memory. The second model comparison was identical to the first one with the exception that the full model included a continuous (excluding unsolved trials) instead of a binarized insight variable predicting subsequent memory while additionally controlling for solution time. The full model was significantly better than the baseline model indicating that insight still significantly predicted subsequent memory ($Chi^2$(1) = 76.34, $p < 0.001$, OR = 1.47, 95% CI [1.33, 1.61]) (see Fig. 2C). This result was replicated in a preregistered behavioral control experiment (https://aspredicted.org/xx7hv.pdf, 11/28/2023; see Fig. S7 in the Supplementary Methods).

We further investigated whether solving Mooney images reflects a similar underlying creative problem-solving process[47] as the more commonly used verbal insight tasks. To assess this relationship, we computed Spearman's rank correlation between the participants' average continuous insight experiences in Mooney images and an anagram task (see Fig. 2D). The obtained correlation was statistically significant ($\rho = 0.633$, $p < 0.001$, 95% CI [0.39, 0.79]), indicating that the insight experience evoked by Mooney images is comparable to that of another verbal insight task.

### Hypothesis 1: VOTC exhibits visual insight-related RC

During insight-related RC, the content representation of the Mooney image changes suddenly from meaningless to meaningful when the object is identified. Our first hypothesis was that VOTC regions (aFusG, pFusG, iLOC, aITG, mITG, pITG) show this insight-related RC, as reflected in changes of distributed activity patterns from pre to post solution. In particular, we operationalized this change by two consecutive representational similarity analyses (RSA)[20]. First, by correlating the activity patterns pre (0.5 s after Mooney image onset) and

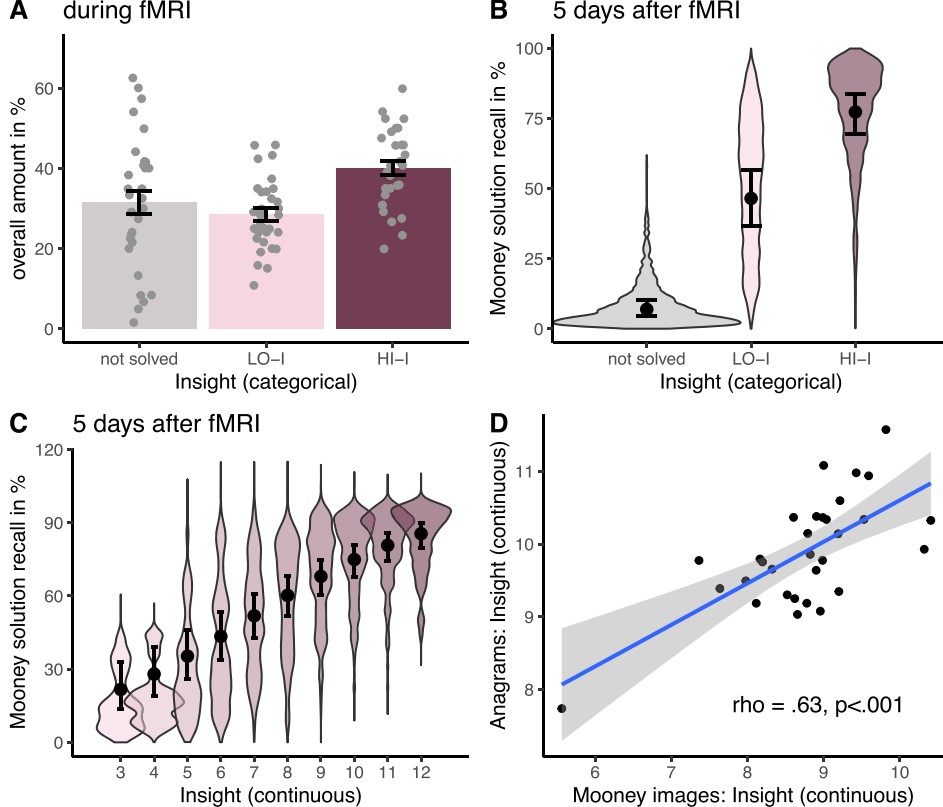

**Fig. 2 | Behavioral insight memory effect. A** Overall proportion of all trials per condition, error bars represent SEM; n = 31 samples. **B** Behavioral insight memory effect (Chi²(2) = 666.49, p < 0.001), based on a two-sided likelihood ratio test comparing a model with and without (categorical) insight; violin plots represent predicted data (n = 2633 samples); violin plots represent predicted data (n = 2633 samples) from single trial analysis split by *HI-I* = correctly solved high insight trials; *LO-I* = correctly solved low insight trials and *not solved* trials, dots are marginal mean percentages to correctly recall the previously identified Mooney image in the respective condition. Error bars represent 95% confidence intervals. **C** Significant relationship between continuous insight (3–12) and memory, controlling for solution time (Chi²(1) = 76.34, p < 0.001, OR = 1.47, 95% CI [1.33, 1.61]), based on a two-sided likelihood ratio test comparing a model with and without (continuous) insight; violin plots represent predicted data (n = 2633 samples) from single trial analysis for each insight value; dots are marginal mean percentages to correctly recall the previously identified Mooney image in the respective condition. Error bars represent 95% confidence intervals. **D** Spearman correlation (subject level) between anagrams and Mooney images. Shaded area represents 95% confidence interval around the fitted regression line (blue) (n = 31 samples).

post solution (button press), and by determining if the correlation decreased more strongly (i.e., greater change) with an increase in correct insight (see Fig. 3A). Second, in areas showing greater multivariate activity pattern change as a necessary condition for correct insight-related RC, we tested whether this change reflects a stronger conceptual reorganization of the identified Mooney object from meaningless to meaningful. For this purpose, we created two model-based representational similarity matrices (RSM) that were derived from two conceptually highly distinct neural networks, AlexNet and Word2Vec[48,49]. AlexNet was trained using large image databases, Word2Vec using large text corpi. The correlation of the distributed activity patterns elicited by the Mooney images and these model-based RSMs reflects the neural representational strength of the Mooney stimulus. We contrasted the representational strength pre and post solution and examined if it increased more from pre to post solution with increasing insight (see Fig. 3B1:4 for further details).

First, we analyzed RC from pre to post solution via multivoxel pattern similarity. To identify ROIs exhibiting the hypothesized RC-related changes in the correlation between pre and post-solution activity patterns, we examined the association between insight in the aforementioned VOTC regions and this correlation, as well as their interaction. This was done using nested model comparisons: the baseline model included random subject and item intercepts, controlling for response time and run order, while two full models further included insight and the interaction between insight and ROI,

respectively. There was no statistically significant main effect for insight (Chi²(1) = 1.85, p = 0.17, ß = 0.01, 95% CI [−0.00, 0.03]) over all regions but an interaction between ROI and insight indicating differences between ROI regarding changes in pre and post solution activity patterns (Chi²(5) = 158.44, p < 0.001) (Fig. 3A). Post hoc analyses revealed that bilateral iLOC (t(19,514.5) = −8.08, p-Bonferroni < 0.001, ß = 0.02, 95% CI [0.01, 0.02]) and bilateral pFusG (t(19,514.5) = −6.09, p-Bonferroni < 0.001, ß = 0.01, 95% CI [0.01, 0.01]) significantly reduced similarity in activity patterns from pre to post solution with increasing insight. All other brain areas showed no statistically significant insight effect in the hypothesized direction (see Fig. 3A).

Second, we analyzed representational strength using AlexNet. We restricted the representational strength analyses to areas (iLOC, pFusG) that showed the hypothesized greater change in pre to post solution activity patterns. To identify areas where the representational strength increased more strongly from pre to post solution for correct insight (see Fig. 3B), we performed a nested model comparison. The baseline model included insight, time (pre vs. post), response time, run order and random subject and item intercepts. The full models further included an insight*time or insight*time*ROI interaction, respectively. The model with the insight*time interaction performed significantly better (Chi²(1) = 15.19, p < 0.001, ß = 0.06, 95% CI [0.03, 0.09]) than the model without this two-way interaction indicating an overall insight*time interaction over both regions. Furthermore, the model including the three-way insight*time*ROI interaction did not explain

## A. RC from pre to post solution: Multivoxel pattern similarity

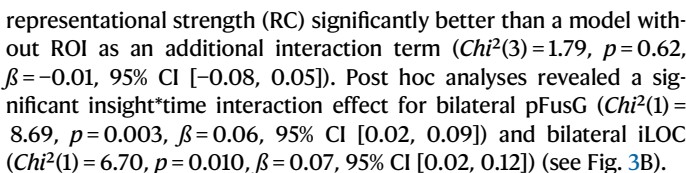

## B. RC from pre to post solution: Representational strength - AlexNet.

representational strength (RC) significantly better than a model without ROI as an additional interaction term ($Chi^2(3) = 1.79$, $p = 0.62$, $ß = -0.01$, 95% CI [−0.08, 0.05]). Post hoc analyses revealed a significant insight*time interaction effect for bilateral pFusG ($Chi^2(1) = 8.69$, $p = 0.003$, $ß = 0.06$, 95% CI [0.02, 0.09]) and bilateral iLOC ($Chi^2(1) = 6.70$, $p = 0.010$, $ß = 0.07$, 95% CI [0.02, 0.12]) (see Fig. 3B).

Finally, we analyzed representational strength using Word2Vec. To ensure that the interaction results obtained via the AlexNet model are robust and generalize to other conceptual models, we also measured RC using a purely verbal conceptual model–Word2Vec[48] and conducted the same nested model comparisons as in the analyses with AlexNet. As displayed in Fig. 3B, the results largely replicated those found with AlexNet. The model with the two-way insight*time interaction ($Chi^2(1) = 16.12$, $p < 0.001$, $ß = 0.06$, 95% CI [0.03, 0.09])

performed significantly better than a model without this interaction indicating an insight effect over both regions. The three-way insight*time*ROI interaction was not significant ($Chi^2(3) = 1.84$, $p = 0.61$). Post hoc analyses revealed a significant condition*time interaction effect for bilateral pFusG ($Chi^2(1) = 12.39$, $p < 0.001$, $ß = 0.07$, 95% CI [0.03, 0.10]) and bilateral iLOC ($Chi^2(1) = 3.92$, $p = 0.048$, $ß = 0.05$, 95% CI [0.00, 0.11]).

In sum, the combination of distributed activity pattern changes and increase in representational strength of the Mooney objects from pre to post solution provide evidence for RC in VOTC, particularly in pFusG and iLOC. These findings support our first hypothesis that VOTC-RC is enhanced by insight. Although these results highlight a neural mechanism of RC, i.e., the cognitive component of insight, they cannot explain the evaluative component of insight, namely the

**Fig. 3 | VOTC regions show representational change (RC).** Insight was analyzed as a continuous variable but for visualization purposes, it was median-split into low, (medium) and high insight values. Asterisk = statistical significance at $p < 0.05$; **A** RC from pre to post solution: multivoxel pattern similarity. Multivoxel patterns per ROI for each time point (pre and post solution) are extracted and subsequently correlated. Pre = 0.5 s after stimulus presentation; post = during solution button press. Those Pre-Post Solution Similarity values ($r$) are subsequently estimated in a linear mixed model as a function of insight and ROI. Bar plots show change (Δ) in Multivoxel Pattern Similarity (MVPS) analysis. Change in MVPS = 1 minus the correlation between the post and pre solution multivoxel pattern in the respective ROI. Violin plots represent predicted data ($n = 19,536$ samples) from single trial analysis for 6 bilateral VOTC regions; points indicate estimated marginal means; error bars represent 95% (between-subject) confidence intervals. Only the estimated slope coefficients (two-sided $t$-test, with Bonferroni correction for multiple comparisons) of bilateral iLOC ($t(19,514.5) = -8.08$, $p < 0.001$, $ß = 0.02$, 95% CI [0.01, 0.02]) and pFusG ($t(19,514.5) = -6.09$, $p < 0.001$, $ß = 0.01$, 95% CI [0.01, 0.01]) showed significant changes in MVPS in the hypothesized direction. **B** RC from pre to post solution: Representational strength—AlexNet. This RSA method employs four steps. (1) A neural activity-derived RSM (Brain RSM, size 120 × 120) is generated for each region-of-interest (ROI) and each time point (pre post solution, see **A**) where

each cell is representing a multivoxel pattern similarity value for each Mooney image pair. (2) A conceptual *Model RSM* (here using AlexNet, size 120 × 120) is generated where each cell is representing a similarity value for each Mooney object pair. (3) For each brain region and each time point, the row of each stimulus (-120) in the *Model RSM* and in the *Brain RSM* are correlated yielding a representational strength measure (i.e., brain-model fit) per region and time point. (4) The representational strength is used as a dependent variable in linear mixed models to investigate which ROIs exhibit an insight-related increase in representational strength from pre to post solution (time). Violin plots depict predicted Representational Strength (Rep-Str) values at single-trial level ($n = 14,652$ samples), defined as the second-order correlation between multivoxel patterns in the respective ROI at pre and post-response time points and conceptual Model RSMs (AlexNet and Word2Vec [W2V]); points indicate estimated marginal means ± SEM. Bilateral iLOC (AlexNet: $Chi^2(1) = 6.70$, $p = 0.010$, $ß = 0.07$, 95% CI [0.02, 0.12]; W2V: $Chi^2(1) = 12.39$, $p < 0.001$, $ß = 0.07$, 95% CI [0.03, 0.10]) and pFusG (AlexNet: $Chi^2(1) = 8.69$, $p = 0.003$, $ß = 0.06$, 95% CI [0.02, 0.09]; W2V: $Chi^2(1) = 3.92$, $p = 0.048$, $ß = 0.05$, 95% CI [0.00, 0.11]) show a significant increase in Rep-Str over time, based on two-sided Likelihood ratio tests (uncorrected for multiple comparison). The brain images were generated using MRIcroGL[116].

positive emotion, certainty, and suddenness associated with the insight experience. In line with previous research, we hypothesized that this evaluative component is mediated by the amygdala and hippocampus.

## Hypothesis 2: amygdala and hippocampus show insight-related activity

To examine our second hypothesis that the evaluative component of insight is mediated by the amygdala and anterior/posterior hippocampus, we estimated whether bilateral amygdala and hippocampal activity during correct Mooney object identification would vary depending on the strength of insight. Note, we used univariate activity because both brain regions are unlikely to represent visual information like VOTC. We conducted the analyses with the data from the single-trial analyses containing a beta estimate of univariate amygdala, anterior and posterior hippocampal activity (ROI) for the solution button press at each trial. To assess whether amygdala and hippocampal activity varies parametrically with the strength of insight, we compared a baseline model including a random participant intercept and controlled for run order and the specific ROI with a full model additionally including insight. Furthermore, we compared this full model with a third model additionally including an interaction term between insight and ROI. For completeness, we also report univariate whole-brain activity (see Supplementary Methods, Fig. S8).

As illustrated by Fig. 4A, the full model including insight performed significantly better than the baseline model without insight ($Chi^2(1) = 47.57$, $p < 0.001$, $ß = 0.09$, 95% CI [0.08, 0.14]) implying a positive relationship between brain activity and insight over all three ROIs. Furthermore, the model with the insight*ROI interaction performed better than the model without the interaction term ($Chi^2(2) = 26.91$, $p < 0.001$) implying differences between the ROIs in this condition. Post hoc analyses revealed that activity in amygdala activity ($z = 7.55$, $p < 0.001$, $ß = 0.18$, 95% CI [0.13, 0.22]) and anterior ($z = 5.57$, $p < 0.001$, $ß = 0.13$, 95% CI [0.08, 0.18]) but not posterior ($z = 1.11$, $p = 0.27$, $ß = 0.03$, 95% CI [−0.02, 0.07]) hippocampus were statistically significantly associated with insight (see Fig. 4A). A control analysis was performed adjusting solution time between HI-I and LO-I trials (i.e., removing fast HI-I and slow LO-I trials until the time taken to solve both types of trials was no longer significantly different [$p > 0.20$]) and repeating the above mentioned analysis. The results did not change significantly (see "Further control analyses to account for the influence of solution time" in the Supplementary Methods).

To exclude potential normalization artifacts due to the small volume of the amygdala and hippocampal ROIs, we additionally conducted another ROI analysis in subject space. Univariate activity during

solution, i.e., onset of the button press, was parametrically modulated by insight and solution time and estimated using GLMs as implemented in SPM[50]. To test whether amygdala and hippocampal activities parametrically vary depending on the strength of insight, we computed two models, the baseline model including a random subject intercept and controlling for the run order and the full model additionally including the respective ROI as predictor.

The intercept of the baseline model was significant suggesting that bilateral amygdalar and hippocampal activity are significantly parametrically modulated by insight during (correct) solution ($t(127.8) = 6.44$, $p < 0.001$, $ß = 0.14$, 95% CI [0.10, 0.18]). The full model performed significantly better than the baseline model suggesting differences between the ROIs ($Chi^2(2) = 28.59$, $p < 0.001$). However, post hoc analyses showed that the intercept in the baseline model was significant regardless of whether amygdala activity ($t(121.7) = 5.21$, $p < 0.001$, $ß = 0.18$, 95% CI [0.11, 0.25]), anterior ($t(121.9) = 3.24$, $p = 0.002$, $ß = 0.10$, 95% CI [0.04, 0.15]), or posterior hippocampus ($t(120.4) = 4.37$, $p < 0.001$, $ß = 0.14$, 95% CI [0.08, 0.21]) was modeled. This suggests that all three ROIs are significantly modulated by insight in a parametric manner.

**Activity in amygdala and hippocampus during solution is predominantly associated with insight dimensions certainty and emotion.** To explore the factors influencing the impact of insight on the hippocampus and amygdala, we delved into the three dimensions of insight. Note, our measurement model revealed correlations among all three dimensions—certainty, suddenness, and emotion. However, our focus was on identifying which dimension contributes unique variance beyond the shared variance among the three (note, these insight dimensions have only four values and are less sensitive than the insight sum measure). For this, we repeated the above reported mixed effects single-trial analysis (standard space), substituting the insight measure with certainty, suddenness, and emotion and report the $t$-test performed on their individual regression weights. The results showed that for bilateral amygdala, both emotion ($t(2779.8) = 2.02$, $p = 0.044$, $ß = 0.05$, 95% CI [0.00, 0.09]) and certainty ($t(2959.2) = 5.39$, $p < 0.001$, $ß = 0.13$, 95% CI [0.08, 0.18]) individually explained variance in the BOLD signal. For the anterior hippocampus ($t(2715.5) = 6.77$, $p < 0.001$, $ß = 0.17$, 95% CI [0.12, 0.21]) and posterior hippocampus ($t(2753.8) = 4.07$, $p < 0.001$, $ß = 0.10$, 95% CI [0.05, 0.15]), only certainty explained individual variance beyond the influence of suddenness and emotion. Hence, the BOLD effects associated with the evaluative component of insight appear to be primarily influenced by the level of certainty regarding the correctness of the solution and (to a lesser degree) positive emotions.

## A. Univariate Activity

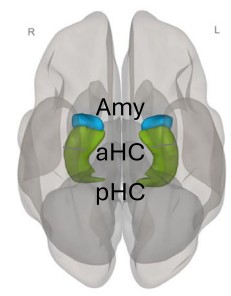

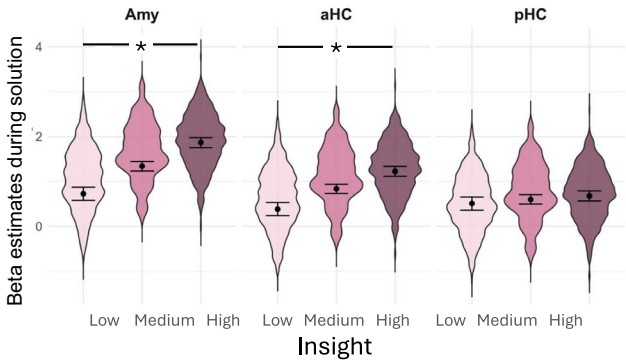

## B. Functional connectivity

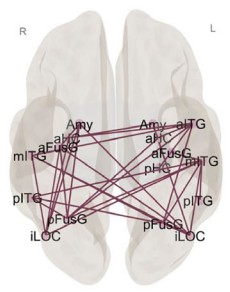

Solution NW: HI-I>LO-I

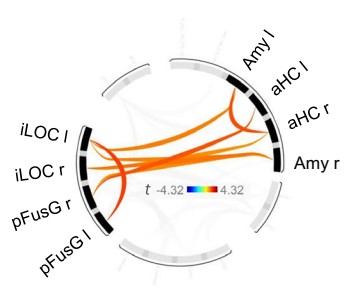

**Fig. 4 | Enhanced insight-related amygdala and hippocampal activity and information integration in the solution network.** Asterisks denote statistically significant differences at *p* < 0.05. **A** Amy = amygdala, aHC = anterior hippocampus, pHC = posterior hippocampus; Violin plots illustrate predicted beta estimates at the solution time point (*n* = 9432 samples) from single trial analysis in standard space, with dots representing the estimated marginal means ± SEM. Insight was analyzed as a continuous variable but for visualization purposes, it was split into low, medium and high insight values. The estimated slope coefficients (two-sided *t*-test) representing the strength of insight were significantly associated with amygdalar (Amy, *t*(9238.4) = 7.54, *ß* = 0.18, *p* < 0.001, 95% CI [0.13, 0.22]) and anterior hippocampal (aHC, *t*(9238.4) = 5.56, *ß* = 0.13, *p* < 0.001, 95% CI [0.08, 0.18])) activity. **B** NW network; network measures stem from contrast: *HI-I > LO-I. HI-I* = trials solved with high accompanied insight; *LO-I* = trials solved with low accompanied

insight. Insight-related increased functional connectivity (*p-FWE* = 0.009, one-sided) for *solution network* calculated from a ROI-to-ROI connectivity matrix during 10 s stimulus presentation averaged across 31 participants including VOTC regions: bilateral inferior Lateral Occipital Cortex (iLOC), anterior (aFusG) and posterior (pFusG) Fusiform Gyrus, anterior (aITG), middle (mITG) and posterior (pITG) Inferior Temporal Gyrus as well as Amygdala (Amy), anterior (aHC) and posterior (pHC) Hippocampus. Violin plots represent effect size (averaged ROI-to-ROI Fisher z-transformed correlation coefficients) for estimated connection strength within the *solution network*, separated by condition (HI-I, LO-I) across participants. Dots are means and error bars represent 95% confidence intervals. Connectivity matrix depicts significant functional connectivity between RC-VOTC and amygdalar/hippocampal ROIs (*p* values are provided in Table S1). The brain image was generated using CONN[53] [RRID:SCR_009550].

## Hypothesis 3: the VOTC, amygdala, and hippocampus network becomes more integrated during insight

Next, we examined the assumption that the stronger conceptual RC is efficiently integrated into the solution process. For this, we created a *solution network* consisting of all bilateral VOTC ROIs (aITG, mITG, pITG, aFusG, pFusG, iLOC) relevant for object identification[42]. The *solution network* (18 ROIs in total) further included amygdala and anterior as well as posterior hippocampus because they are likely involved in the evaluation of the solution content. Efficient information integration was quantified via increased functional connectivity within this network during the solution process, i.e., stimulus presentation (ten seconds block while participants tried to identify the Mooney images). Efficient information integration was additionally quantified via graph measures representing functional segregation/ integration and network centrality[51] based on the functional connectivity values from our predefined *solution network* during stimulus presentation. We assumed increased network centrality (degree), network functional integration (global efficiency, [reduced] average path length) as well as functional segregation (clustering coefficient, local efficiency) to represent efficient information integration.

**Increased functional connectivity of the solution network during insight.** Functional connectivity (FC) was computed between each

pair of the ROIs in the *solution network* resulting in a sum FC score for each ROI (*Mass* = sum of *t*-squared statistics over all connections per ROI)[52]. The resulting connectivity values were predicted by the insight factor (correct *HI-I > LO-I*) in CONN[53] (note, using a continuous insight measure for graph analyses was not applicable, therefore we median-split insight into high (HI-I) and low (LO-I) insight trials in this context). Due to hemispheric asymmetries in FC[54,55], we split the ROIs in the left and right hemisphere but FWE-controlled for multiple comparison. The *solution network* showed significantly increased functional connectivity (*Mass* = 364.03, *p-FWE* = 0.009) for *HI-I > LO-I* (see Fig. 4B). The VOTC-RC ROIs, pFusG and iLOC, were strongly connected to other ROIs in the network particularly to amygdala and hippocampus (see Fig. 4B) suggesting that VOTC-RC ROIs and amygdala/hippocampus are directly connected during insight. Note, the posterior hippocampus did not show any significant connections with VOTC-RC areas. The statistics with all individual connections in this network are listed in the Supplementary Methods, Table S1.

**Increased functional segregation and integration of the solution network during insight.** Insight-related increased functional connectivity in the *solution network* was consistent with results from different graph theory measures.

**Network centrality.** Degree ($t(30) = 3.17$, $p = 0.003$, $ß = 0.41$) was increased for the contrast *HI-I > LO-I* indicating an insight-related stronger overall connectedness of ROIs in the *solution network*. Individual ROIs that showed more insight-related degree within the *solution network* were bilateral pFusG (right: $t(30) = 3.55$, *p-FDR* = 0.011, $ß = 0.60$; left: $t(30) = 2.31$, *p-FDR* = 0.040, $ß = 0.47$), bilateral iLOC (right: $t(30) = 2.53$, *p-FDR* = 0.040, $ß = 0.53$; left: $t(30) = 2.45$ *p-FDR* = 0.040, $ß = 0.57$) as well as left anterior hippocampus ($t(30) = 2.26$, *p-FDR* = 0.040, $ß = 0.53$) and right amygdala ($t(30) = 2.57$, *p-FDR* = 0.040, $ß = 0.72$) and left mITG ($t(30) = 2.34$, *p-FDR* = 0.040, $ß = 0.61$).

**Functional segregation.** The *solution network* also exhibited increased local efficiency ($t(22) = 2.85$, $p = 0.005$, $ß = 0.05$) as well as an increased clustering coefficient ($t(22) = 3.08$, $p = 0.003$, $ß = 0.05$) for *HI-I > LO-I*. No individual ROI survived multiple comparison correction for either of these two graph measures. However, right iLOC showed an increased clustering coefficient ($t(17) = 1.85$, *p-uncor* = 0.041, $ß = 0.05$) when exploring uncorrected *p*-values. Those results imply that insight-related processing is more specialized as it occurs within more densely interconnected ROIs of the *solution network*[51].

**Functional integration.** Furthermore, the *solution network* exhibited an insight-related higher global efficiency ($t(30) = 2.25$, $p = 0.016$, $ß = 0.02$) and a reduced average pathlength ($t(30) = -2.73$, $p = 0.005$, $ß = -0.06$). For global efficiency, no individual ROI survived the threshold. For average pathlength, bilateral pFusG (left: $t(28) = -2.43$, *p-FDR* = 0.024, $ß = -0.07$; right: $t(28) = -3.02$, *p-FDR* = 0.011, $ß = -0.07$) and iLOC (left: $t(29) = -3.38$, *p-FDR* = 0.009, $ß = -0.08$; right: $t(27) = -2.89$, *p-FDR* = 0.011, $ß = -0.07$) showed significantly less average pathlength for *HI-I > LO-I*. Those results imply that information is rapidly integrated from different brain regions of the *solution network*. Specifically, those VOTC regions identified to be involved in RC (pFusG & iLOC) exhibit hub properties within the *solution network* including increased functional integration.

## Hypothesis 4: VOTC-RC and hippocampal activity predict insight-related better subsequent memory

Having linked RC in VOTC and amygdala as well as hippocampal activity to the visual insight process, we examined whether the underlying neural correlates of both insight components also predicted better subsequent memory. To enhance statistical robustness, *p* values were obtained through permutation tests ($n = 999$)[56].

First, we tested whether insight-related better memory for correctly solved trials is associated with our two consecutive measures of RC, i.e., (1) the changes of multivoxel activity patterns (i.e., decrease in multivoxel pattern similarity) and (2) an increase in representational strength of the object from pre to post solution in those regions that elicit RC (pFusG, iLOC). To test insight-related better memory for (1), we compared a series of nested models that included insight (continuous variable), memory (remembered vs. forgotten), and ROI, along with interactions among these predictors. For (2), we performed a similar series of nested model comparisons as in (1), but also incorporated time (pre vs. post-solution) and its interactions with the other predictors of interest. All models additionally included random subject and item intercepts controlling for run order and solution time (see "Methods" section for details).

First, the model predicting multivoxel pattern similarity that included memory, controlling for insight, showed a significantly better fit than the model without memory indicating an overall memory effect ($Chi^2(2) = 14.40$, $p = 0.001$, $ß = -0.10$, 95% CI [−0.14, −0.05]). Additionally, the model with a two-way interaction between insight and memory for pre to post solution activity pattern similarity across both RC ROIs, performed significantly better than the model without this interaction ($Chi^2(1) = 9.27$, $p = 0.003$, $ß = -0.07$, 95% CI [−0.11, −0.02]). Finally, the model including a three-way interaction between insight, memory and ROI provided a significantly better fit than the

model excluding it ($Chi^2(3) = 11.49$, $p = 0.011$) indicating differences in insight-related better memory across the two ROIs. Post hoc analyses revealed that pFusG showed a main effect for memory ($Chi^2(1) = 3.75$, $p = 0.048$, $ß = -0.07$, 95% CI [−0.13, −0.01]) when controlling for insight and a significant insight-memory interaction ($Chi^2(1) = 5.84$, $p = 0.014$, $ß = -0.07$, 95% CI [−0.12, −0.01]) (see Fig. 5A). Similarly, iLOC showed a memory effect ($Chi^2(1) = 8.74$, $p = 0.005$, $ß = -0.12$, 95% CI [−0.20, −0.05]) and also an insight-memory interaction effect ($Chi^2(1) = 4.08$, $p = 0.038$, $ß = -0.07$, 95% CI [−0.14, −0.00]) (see Fig. 5A).

2a) When predicting representational strength computed via AlexNet, the model including memory as predictor did not fit statistically significantly better than the model without memory ($Chi^2(1) = 1.35$, $p = 0.231$, $ß = -0.02$, 95% CI [−0.06, 0.02]). However, the model with a memory*time interaction ($Chi^2(1) = 50.84$, $p = 0.001$, $ß = 0.24$, 95% CI [0.17, 0.30]), controlling for insight, performed significantly better than the model without this interaction. There was no statistically significant evidence, though, for an insight*memory*time interaction ($Chi^2(1) = 0.71$, $p = 0.401$, $ß = -0.03$, 95% CI [−0.10, 0.04]) (see Fig. 5B). Furthermore, the four-way interaction with ROI was also not significant ($Chi^2(7) = 4.04$, $p = 0.79$).

2b) The Word2Vec-based model for representational strength, which included memory, fits significantly better than the model without memory, indicating an overall memory effect over both time points when controlling for insight ($Chi^2(1) = 10.33$, $p = 0.002$, $ß = 0.07$, 95% CI [0.03, 0.11]). Additionally, there was a time*memory interaction ($Chi^2(1) = 9.82$, $p = 0.004$, $ß = 0.10$, 95% CI [0.04, 0.17]). Importantly, the model with the three-way insight*memory*time interaction also showed a significantly better fit than the model without this three-way interaction ($Chi^2(2) = 10.46$, $p = 0.004$, $ß = 0.09$, 95% CI [0.02, 0.15]) (see Fig. 5D). Furthermore, the four-way interaction with ROI was not significant ($Chi^2(7) = 6.63$, $p = 0.46$).

Second, we tested whether insight-related better memory is associated with amygdala and hippocampal activity. For this, we used the data from the single-trial analyses containing a beta estimate of univariate amygdala, anterior and posterior hippocampal activity (ROI) for the solution button press at each correctly solved trial. A series of nested models were compared, regressing the beta estimates onto insight (continuous), memory (*Remembered, Forgotten*), ROI as well as an interaction between variables, while also controlling for response time, run order and random subject and item intercepts (see "Methods" section for details).

The model including memory did not significantly outperform the model without it ($Chi^2(1) = 0.17$, $p = 0.67$, $ß = 0.02$, 95% CI [−0.04, 0.06]). However, the model which included an insight*memory interaction provided a significantly better fit than the model without this interaction ($Chi^2(1) = 8.06$, $p = 0.006$, $ß = 0.06$, 95% CI [0.02, 0.11]) across all three regions. Additionally, the better fit of the model with the three-way insight*memory*ROI interaction ($Chi^2(6) = 31.08$, $p < 0.001$), compared to the two-way insight*memory interaction suggests that this interaction varied by ROI. Post hoc analyses indicated statistically significant evidence for an insight*memory interaction only in anterior hippocampus ($Chi^2(1) = 5.68$, $p = 0.015$, $ß = 0.09$, 95% CI [0.02, 0.17]), but not in the posterior hippocampus ($Chi^2(1) = 3.07$, $p = 0.087$, $ß = 0.07$, 95% CI [−0.01, 0.15]) nor in the amygdala ($Chi^2(1) = 1.00$, $p = 0.334$, $ß = 0.04$, 95% CI [−0.04, 0.11]) (see Fig. 5C).

For exploratory purposes, we investigated which insight dimension drives this insight*memory interaction in anterior hippocampus. For this, we repeated the above mentioned mixed effects single trial analysis with the insight*memory interaction, substituting the insight measure with either certainty, suddenness or emotion. Only the regression weight of the interaction between suddenness and memory ($t(2962.2) = 2.69$, *p-Bonferroni* = 0.028, $ß = 0.10$, 95% CI [0.03, 0.18]) predicted significant variance in anterior hippocampus while no statistically significant evidence for an interaction between

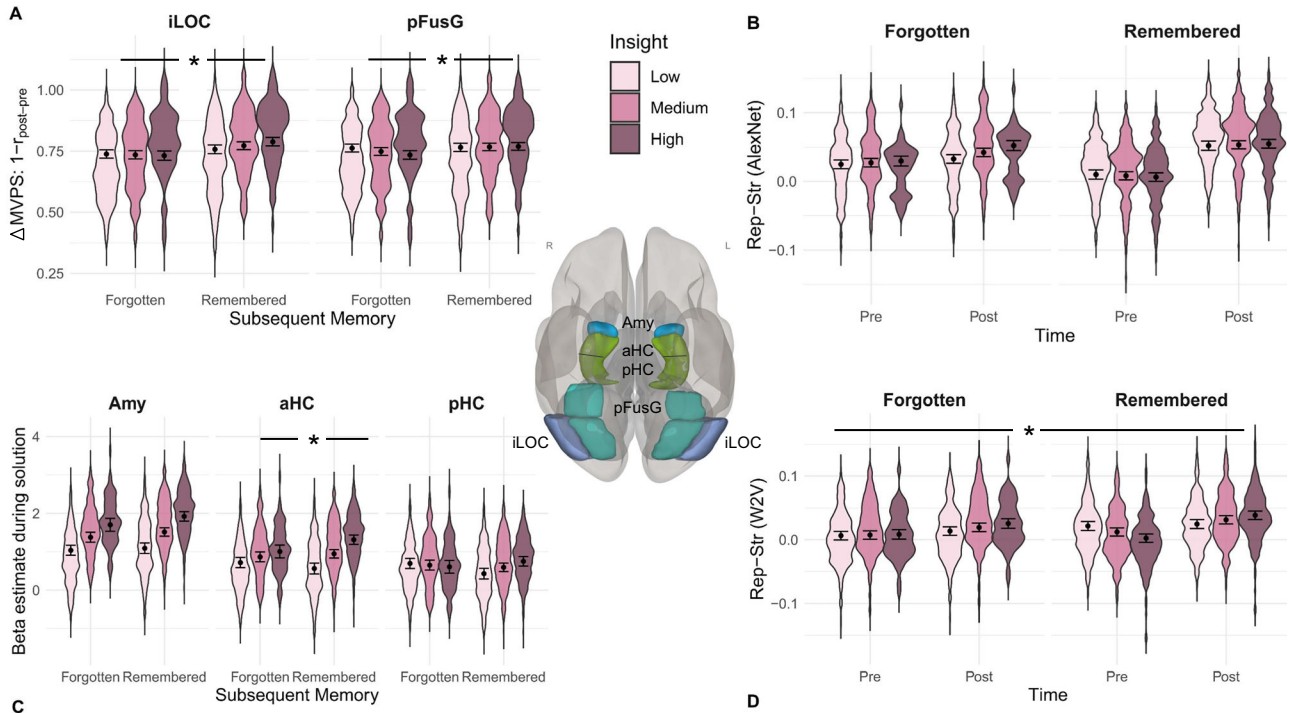

**Fig. 5 | Representational change in VOTC and hippocampal activity during solution are associated with insight-related better subsequent memory.** Amy Amygdala, aHC anterior hippocampus, pHC posterior hippocampus, pFusG posterior Fusiform Gyrus, iLOC inferior Lateral Occipital Lobe. Violin plots illustrate the predicted estimates at a single-trial level, with dots representing the estimated marginal means ± SEM (likelihood ratio tests, uncorrected for multiple comparison). Insight was analyzed as a continuous variable but for visualization purposes, it was split into low, medium and high insight values. **A** Change (Δ) in Multivoxel Pattern Similarity (MVPS). Change in MVPS = 1 minus the correlation between the post and pre solution multivoxel pattern in the respective ROI. Asterisk represents Insight*Memory interaction for iLOC ($Chi^2(1) = 4.08$, $p = 0.038$, $ß = -0.07$, 95% CI [−0.14, −0.00], $n = 2310$ samples, tested two-sided) and pFusG ($Chi^2(1) = 5.84$,

$p = 0.014$, $ß = -0.07$, 95% CI [−0.12, −0.01]), $n = 4620$ samples, tested two-sided). **B** Results display combined representational strength of solution object for iLOC & pFusG measured via a conceptual model created out of the penultimate layer of AlexNet. There was no statistically significant Insight*Memory*Time effect ($p > 0.40$; $n = 13,860$ samples). **C** Violin plots represent beta estimates of BOLD activity at post solution divided by insight memory conditions. Asterisk represents Insight*Memory interaction for aHC ($Chi^2(1) = 5.68$, $p = 0.015$, $ß = 0.09$, 95% CI [0.02, 0.17], $n = 2976$ samples). **D** Results display combined representational strength of solution object for iLOC & pFusG measured via a conceptual Word2Vec [W2V] model. Asterisk represents Insight*Memory*Time interaction at $p < 0.05$ ($Chi^2(2) = 10.46$, $p = 0.004$, $ß = 0.09$, 95% CI [0.02, 0.15], $n = 13,860$ samples). The brain image was generated using CONN[53] [RRID:SCR_009550].

memory and emotion ($p > 0.268$) or memory and certainty ($p > 0.096$) was found.

For reasons of completeness, we also explored whether FC and graph theoretical measures also predicted insight-related better memory. However, these results provided no consistent evidence (all analyses are detailed in "Relationship between insight-related efficient information integration and subsequent memory" in the Supplementary Methods).

## Discussion

Despite two decades of research, little is known about the neural mechanisms of insight, particularly regarding the underlying representational change and its impact on subsequent memory. To investigate these issues, we scanned participants with fMRI while they identified Mooney images, and tested their memory for the solutions 5 days later. Identifying Mooney images requires rechunking and integrating their elements into a coherent object (RC, cognitive component of insight), which often elicits positive emotions and a feeling of suddenness and certainty (evaluative component of insight). We found that insight enhanced (1) RC in ventral occipito-temporal cortex (VOTC), (2) activity in amygdala and hippocampus (3) efficient information integration within a network of solution-processing brain regions, and (4) subsequent memory involving VOTC RC and hippocampal activity. Taken together, these findings represent coherent evidence of an integrated neural mechanism for insight and its association with memory. Its generalizability to other, more complex

creative problem-solving tasks requires further assessment. The four findings are discussed in separate sections below.

### Cognitive component of insight: representational change in VOTC

While insight-related RC has been examined behaviorally[13,19] and through univariate BOLD activity[18,57], this study presents multivariate evidence. Using a visual Mooney identification task, we found (visual) insight-related RC in middle VOTC regions (pFusG and iLOC), which are areas that process complex visual features[58,59]. From pre to post solution, the conceptual representation of the target objects increased in pFusG and iLOC, suggesting a change from representations coding for meaningless black and white Mooney image blobs to representations coding for meaningful objects. These results align with prior research, reporting the presence of bilateral ERP components (N[cl]) in LOC during the recognition of objects in fragmented images, similar to Mooney images[60,61]. Note, there is limited knowledge whether the process that leads up to RC, e.g., the transition from meaningless to meaningful objects, occurs gradually or discretely and is only perceived as sudden. However, previous research suggests a linear correlation between BOLD activity in the fusiform gyrus as well as occipital temporal sulcus and awareness of object identity[41], indicating a more continuous process.

However, we did not find statistically significant evidence for RC in anterior VOTC (aFusG and aITG), which processes more abstract concepts and semantics[58,59]. This could reflect the typically lower

signal-to-noise ratios (SNR) in the MRI of this region[62] or the visual nature of the Mooney identification task. Verbal restructuring likely involves regrouping conceptual relationships in verbal tasks[16,17]. Hence, future research could examine if insight tasks that require more abstract semantic processing (e.g., verbal restructuring) yields RC in anterior VOTC regions.

## Evaluative component of insight: amygdala and hippocampal activity

Our second finding was that insight—in particular its dimensions certainty and emotion—boosted activity in the amygdala and hippocampus[24,26,27]. In previous insight studies, amygdala activity has been attributed to positive emotions[10,25], aligning with the broader role of this region in emotional arousal[63], and hippocampal activity to sudden solution awareness[7,8,64], in line with its role as a novelty or "mismatch" detector[29–31]. This function has been linked to a prediction error, when there is a mismatch between a prior expectation and the outcome, which leads to the updating of these expectations and memory storage[65–67]. In our task, as a result of RC in visual cortex, the originally meaningless stimulus becomes suddenly meaningful, yielding a prediction error, which triggers hippocampal activity and the storage of the salient event[8]. This aligns with our finding that the insight memory effect in the anterior hippocampus is specifically driven by the suddenness dimension of insight.

## Insight-related integration of the solution network

Our third finding was that insight boosted functional connectivity within the *solution network* (VOTC, amygdala, hippocampus). As shown by functional connectivity and graph measures, insight was associated with greater functional integration and segregation of the *solution network*. Those results suggest that during insight, solution-relevant information was more efficiently processed, as has been proposed before for the creative process[68,69] and particularly for insight[12,16,70,71]. This provides neuroscientific evidence for these ideas in the context of insight. Importantly, during insight, both VOTC-RC regions, bilateral pFusG and iLOC, increased their network communication (increased degree and decreased average path length). This result suggests that these regions flexibly increased their local connectedness and ability to combine visual information, which could allow them to more efficiently integrate the reorganized internal representations within the ventral stream object processing pathway, including the hippocampus and the amygdala.

Note however, functional connectivity, as measured here, cannot determine the precise timing nor the direction of the effects within the *solution network*. In a feedforward direction, RC in VOTC could lead to evaluation responses in hippocampus and amygdala, whereas in a feedback direction, hippocampal and amygdala responses could guide or enhance RC in VOTC areas. For example, a Mooney image could trigger the automatic memory retrieval of a cat image, which then guides the rechunking and integration of the Mooney image (RC) in VOTC[72]. Moreover, emotions triggered by the sudden solution may trigger the amygdala to enhance solution-relevant sensory representations thanks to its ability to project to all levels of the ventral stream[28,73,74].

## Insight and memory

Finally, this study offers evidence for the neural mechanism underlying insight's enhancement of memory. Specifically, RC in VOTC (iLOC) and anterior hippocampal activity predicted subsequent memory effects with increasing insight to a greater extent for trials rated with greater insight. However, we did not observe a statistically significant interaction between insight and memory in relation to amygdala activity. A couple of studies linked amygdala activity to improved subsequent memory during problem solving[10,34] but without considering the critical insight*memory interaction[75]. Note, however, the high variability

between insight and memory (few forgotten trials with higher insight), led to generally low statistical power to detect an insight*memory interaction.

The association between VOTC-RC (iLOC) and subsequent memory aligns with multivariate fMRI studies showing that representational strength during the encoding of everyday objects predicts their later recall[37,76]. Whereas in these studies subsequent memory was predicted by absolute representational strength, in our study, later memory was specifically predicted by the insight-related *change* in strength from pre to post solution. This is further consistent with a "prediction error"-memory mechanism, in which memories are strengthened proportional to the extent to which the outcome (e.g., suddenly identifying a meaningful object) deviates from prior expectations (e.g., solution object was unexpected)[63,77,78]. Note, the RSA analyses using AlexNet did not show a significant correlation with insight-related better memory, whereas Word2Vec did. This difference may be attributed either to issues of statistical power or to the observation that semantic representations, linked to better subsequent memory[79], may be more effectively captured by the Word2Vec model than by the vision-based AlexNet model.

The observation that hippocampal activity can predict insight-related better memory aligns with its pivotal function in converting transient percepts into durable memory traces[38,80]. Note, consistent with previous accounts, we proposed a double role for hippocampus, as a novelty or "mismatch" detector and as an encoder of this novel event[7,8,29,67]. Hence, in the context of insight, the hippocampus may detect novel, unexpected internal events (e.g., enhanced conceptual RC of solution object in VOTC) by eliciting a prediction error signal (surprise) and subsequently encodes these events, proportional to the magnitude of this conceptual RC[8]. This dual function is further supported by hippocampal activity showing an insight*memory interaction *and* a main effect for insight, as well as by increased functional connectivity between VOTC-RC regions (pFusG, iLOC) and the hippocampus. Note however, no FC evidence predicts subsequent memory, possibly due to limited power, independent contributions of both regions to insight-related memory enhancement or the possibility that the chosen time window (10 s) is too extensive to capture the transient encoding advantage associated with insight.

## Limitations

Three key limitations should be addressed. Firstly, the accuracy metric is not exact as participants could not enter specific names of identified objects during problem-solving due to technical constraints in the scanner. Nevertheless, we assessed the likelihood of incorrectly classified correct responses through a control experiment (18.7%, refer to "Further control analyses to account for the influence of solution time" in the Supplementary Methods) and then adjusted the accuracy and subsequent memory measures accordingly for this proportion. Secondly, the limited number of trials in the memory condition, particularly for forgotten trials, may reduce statistical power of our results and increase interindividual variability. To mitigate this, we conducted permutation tests for all brain-derived insight-memory analyses, except for connectivity/graph analyses, where this method was not available. Thirdly, we observe a consistent correlation between solution time and our insight measure, a phenomenon documented in previous studies[46], which, however, can induce time-on-task effects on the size of the BOLD signal and therefore bias the results[81]. To address this issue, we included solution time as a covariate in all reported analyses at the single-trial level. Additionally, we conducted further control analyses to account for the influence of solution time (see "Further control analyses to account for the influence of solution time" in the Supplementary Methods), such as adjusting solution time between *HI-I* and *LO-I* trials. However, all control analyses consistently

demonstrated a systematic relationship between insight/memory and the corresponding brain measure.

This work enhances our mechanistic understanding of the core components of insight, a central aspect of creativity, and its connection to memory formation. We presented consistent neural evidence indicating that insight in problem solving likely reflects an enhanced conceptual RC of solution representations efficiently integrated into a coherent whole. This RC generates solution awareness alongside an emotional and suddenness response in the amygdala and hippocampus, boosting solution encoding and subsequent memory. These findings illuminate memory formation mechanisms and may possess educational implications, highlighting the importance of fostering insight-driven learning environments as effective avenues for problem-solving and knowledge retention.

## Methods

### Participant details

The study adheres to all ethical regulations and received approval from the local ethics committee of Humboldt University Berlin (proposal number 2020-48). No statistical method was used to predetermine sample size due to the absence of prior multivariate results on insight. Based on previous research[6,10], 20 participants would suffice for behavioral and univariate fMRI effects. To account for the lower power of multivariate fMRI, we aimed for a final sample size of approximately 30. The final sample consisted of $N = 31$ [age (in year): range = 19–33, 20 females: $M = 25.13$; 11 males: $M = 26.27$]. Initially, the 38 participants [age (in years): range = 20–34, 23 females: $M = 26.6$; 15 males: $M = 25.7$] were recruited via an online student platform in Berlin. The study was conducted in German. Inclusion criteria were German as mother language, normal or corrected to normal vision, between 18–35 years, no prior neurological or psychiatric diseases and MRI compatibility. Participants provided informed consent prior to study enrollment. All participants received monetary compensation according to their time on task. We excluded 7 participants from data analysis due to technical issues at the scanner ($N = 1$), excessive head movement in the scanner ($N = 3$), pathological findings in brain anatomy ($N = 2$) and too low performance (0% correct recall) in the subsequent memory test ($N = 1$). In all analyses, we pooled data from both sexes since we did not have a specific research question related to sex differences and did not have enough participants to separate the sample by sex.

### Stimulus material

To induce perceptual insight in the MR scanner, we used high-contrast, black and white (henceforth Mooney) images (see Fig. 1A), which have been successfully used to elicit insight[6,10,40]. Those images depict a clearly identifiable object but, due to their high-contrast thresholding, they are usually hard to identify at first, with response times taking several seconds[40]. Visually rechunking the abstracted shapes can lead to sudden object identification, often involving an insight[6]. Examples of Mooney images used in this study and their real-world pictures are shown in Fig. S1 (Supplementary Methods). The materials consisted of 120 Mooney images used for the problem solving task in the scanner and later included as "old" items in the memory test plus 60 additional Mooney images included in the memory test as "new" (distractor) items (180 in total). A fraction of these images ($N = 34$) were selected from Imamoglu et al.[40] and the rest were created using a custom software (available here https://github.com/MaxiBecker/Insight_Memory_Effect). The final set of images was selected across several pilot studies to ensure a solution accuracy of at least 30%, a response time in the 2–13 s range, and sufficiently high rate and variation of self-rated insight. The final set had a mean accuracy of 56% (SD = 18.2%), mean response time 6.23 s (SD = 3.11 s) and probability of self-rated insight of 62% (SD = 17%, range [20%–100%]).

For a different research question, we also presented 120 anagrams in an interleaved fashion with the Mooney images. Those anagrams consisted of six to seven letter frequent words of concrete objects in German (e.g., "BAGGER"). The word's letters were scrambled and for each anagram, participants had to rearrange a set of letters to form a meaningful word (e.g., "RGEAGB" can make "BAGGER"). Mean accuracy was 61.4% (SD = 17.8%).

### Behavioral data acquisition

The behavioral data including the aggregated univariate and multivoxel pattern fMRI data, as well as the stimulus material have been made publicly available[82].

**Assessing insight.** Originally, insight was only binarized as present or absent[24,83,84]. However, some evidence suggests that the overall subjective insight (AHA) experience is more continuous and can be decomposed into three different main dimensions: (1) positive emotional response or internal reward upon finding the solution, (2) experienced suddenness of the solution and (3) certainty about the solution[1,22]. To have better control over these three dimensions, we assessed them separately (see section: "Mooney image paradigm"), and then combined them into a compound insight measure by adding up the three ratings into a continuous scale (3–12). The individual dimensions were only employed for exploratory investigations to determine which ones influence univariate activity in amygdala and hippocampus during both insight and insight-related memory processes. Analyzing the data with the continuous insight variable for connectivity and graph analyses was not applicable. Hence, for those analyses, the continuous insight measure was median split into high (*HI-I*) and low (*LO-I*) insight trials.

**Mooney image paradigm.** Participants had two MRI sessions on two consecutive days and in each session they completed two runs with 30 trials each. The presented images per run and session were counterbalanced between participants to avoid order effects. Before entering the MRI scanner, participants were instructed orally and received three test trials to assure they understood the task and the concept of the Aha experience with its individual scales: "When solving problems including identifying Mooney images, you may experience the solution in different ways. Sometimes, the solution comes to you in a sudden manner, with a strong sense of certainty and a strong positive emotion −this is what we commonly refer to as an Aha experience. However, the solution can also emerge more gradually, accompanied by little or no positive emotion and less certainty about its correctness. It's important to note that these three aspects−suddenness, emotion, and certainty−may vary independently or together. So after identifying each Mooney image, you are asked about the solution and subsequently HOW you experienced finding the solution: You will first be asked to rate the degree of suddenness with which you became aware of the solution, using a scale from 1 (indicating a gradual solution awareness) to 4 (indicating a sudden solution awareness). Next, you will be asked to assess your emotional response upon finding the solution, on a scale from 1 (no positive emotion) to 4 (strong positive emotion). Finally, you will rate your certainty about the correctness of the solution, on a scale from 1 (very uncertain) to 4 (very certain)."

All these assessments will be made using a 4-point Likert scale. The paradigm was presented on a black screen (resolution 1280*960 pixel) using Matlab (2016a) and Psychtoolbox-3 (v3.0.17)[85]. The black/white Mooney images had a size of 800 × 800 pixels. During each trial a Mooney image was presented for 10 s after a jittered fixation cross (mean 3 s ±1 s) (see Fig. 1B). The participants had a four-button response box in their right hand and were instructed to press the "solution" button (right index finger) as soon as they found a solution even though the Mooney image would stay on the screen for the remaining time. The Mooney images were presented as an invariant 10 s block to be able to compare time series of equal length for subsequent functional connectivity analyses. If the participants did not

press the "solution" button during the 10 s stimulus presentation, a new trial would start, and the participants were instructed to press a "no solution" button (middle finger) as soon as they saw the fixation cross to keep the amount of button presses constant for each trial. If the participants did press the "solution" button during stimulus presentation, they would be first presented with a jittered fixation cross (mean 3.5 s ±1.5 s), followed by the four questions regarding the solution and their insight experience (split by their three dimensions: suddenness, positive emotion and certainty). There was no time limit for these 4 questions, and subsequently, the new trial would start. Subsequently, four out of 13 broad categories, e.g., "mammal", "human being", "body & health" or "objects in the house" (see "Categories as response options for solution in the MRI scanner" in the Supplementary Methods) were shown and the participants had to indicate the category of their solution. They had max. 5 s to choose one response category and on average, they needed 3.13 s (SD = 2.5 s) to make a response. This way of assessing the solution was used to test whether participants correctly identified the Mooney object, because they could not type their response in the scanner. Note, the 30 Mooney images per run were presented interleaved together with 30 anagram riddles which are not directly part of the current study.

**Subsequent memory test.** Five days after the second fMRI session (4.62 days, SD = 0.49), participants completed the subsequent memory test online at home. All 120 Mooney images they had seen in the scanner were randomly presented mixed with the 60 new Mooney images they had not seen before. Participants were explicitly instructed not to identify the Mooney objects again but only rely on their memory. In each trial the respective Mooney image was shown after a fixation cross (1 s) for maximally eight seconds. During this time, participants had to decide whether they had seen this Mooney image before during the MRI sessions, i.e., whether they recognized the high-contrast black and white picture. In particular, they were asked to rate their confidence on a 5-point scale (definitely old, probably old, don't know, probably new, definitely new). If they responded 'don't know', 'probably new' or 'definitely new', the next trial would start. If they responded old (definitely old, probably old), they were asked whether they had identified this Mooney object before in the scanner again using a 5-point confidence scale without a time limit (definitely solved, probably solved, don't know, probably not solved, definitely not solved). If they responded 'solved' (definitely solved or probably solved), they were finally asked to enter the name of the Mooney object (no time limit). Subsequently the next trial would start.

## MRI data acquisition

Task-based functional (henceforth fMRI) and structural images were collected on a Siemens Magnetom Prisma 3 T scanner (Erlangen, Germany) and a standard 64-channel head coil was used. All sequences were adapted according to the Human Connective Project[86]. Multiband functional images were collected using a T2*-weighted echo planar imaging (EPI) sequence sensitive to blood oxygen level dependent (BOLD) contrast (TR = 0.8 ms; TE = 37 ms, voxel size = 2.0 mm³, flip angle = 52°, FoV = 208 mm, acquisition matrix = 208 × 208 × 144; 72 slices). The structural images were obtained using a three-dimensional T1-weighted magnetization prepared gradient-echo sequence (MPRAGE) (TR = 2500 ms; TE = 2.22 ms; TI = 1000 ms, acquisition matrix = 240 × 256 × 167, FoV = 256 mm, flip angle = 8°; 0.8 mm³ voxel size, 208 slices). Additionally a spin echo field map was acquired to account for the B0 inhomogeneities (TR = 8000 ms; TE = 66 ms; flip angle = 90°, 72 slices; FoV = 208 mm; 2 mm³ voxel size, acquisition matrix = 208 × 208 × 144).

## Behavioral data analysis

To eliminate trials that might solely involve object recognition without insight components, we excluded all trials with solution times faster than 1.5 s from all behavioral and fMRI analyses. Additional details regarding the estimation of this threshold can be found in "Estimating threshold for insight trials based on object recognition time" and Fig. S3 (Supplementary Methods). To further address a potential confound associated with the offset of the Mooney image (a concern specific to fMRI analyses but to maintain this consistently across all analyses), we also excluded any trials that were solved after 9.5 s. The average amount of remaining trials per condition is illustrated in Fig. S5.

**Accuracy adjustment.** As participants selected one of four response categories in the Mooney identification task, some incorrect responses were classified as correct just by chance. To assess the proportion of falsely classified correct responses, a behavioral control experiment was conducted. The Mooney identification task in this control experiment mirrored the task in the fMRI sample, except that participants typed the solution word instead of choosing one out of four response categories, eliminating ambiguity. The proportion of falsely classified correct responses in the fMRI experiment was estimated by comparing the solution words from the Mooney identification task in the control experiment with the response categories of the same task in the fMRI sample. The incorrect solution words from the control experiment were then categorized based on whether they (1) aligned with the correct response category (i.e., falsely classified as a correct response), (2) aligned with one of the other response categories (i.e., correctly classified as an incorrect response), or (3) did not align with any specified response category (i.e., random guess and therefore 25% chance that response is falsely classified as a correct response). The proportion of falsely classified correct responses of all correct responses in the control experiment was calculated from this categorization and amounted to 18.7% (SD = 10%, 7% came from random guessing and 11.7% aligned with the correct response category). The amount of correctly identified and subsequently remembered Mooney images in the fMRI experiment was finally adjusted by this proportion.

**Subsequent memory measure.** We were more interested in recall (naming the correct Mooney object) than just recognition (remembering to have seen the black/white Mooney dots) because in the context of problem solving the more relevant memory measure is the ability to recall the content of the solution than just to recognize the problem, i.e., Mooney object. To reduce the likelihood that participants just identify the Mooney image again instead of remembering the depicted object, we used all three questions as a compound memory measure. Therefore, the identification of the Mooney object was considered as remembered, when it was correctly identified in the MR scanner and (1) the Mooney image was correctly recognized as seen in the scanner, (2) the Mooney image was correctly recognized as having been identified in the scanner and finally (3) the hidden Mooney object was correctly named in the Subsequent Memory Test (see section "Subsequent memory test"). Insight-related better memory for all fMRI analyses was measured as an interaction between insight (continuous) and memory (Remembered vs. Forgotten). Forgotten trials were defined as trials that were initially correctly identified, but then subsequently not or not correctly remembered. On average, 53% were not remembered and 47% were incorrectly remembered from all *Forgotten* trials (SD = 17%). The false alarm rate was calculated as the percentage of incorrectly recognized Mooney images of all new images (*n* = 60) per participant.

All behavioral and fMRI 2nd level analyses (except for functional connectivity analysis) were conducted in R (v4.4.0)[87]. To control for random variance between subjects and the different items, we calculated random mixed effects models using the lmer-function (v1.1-35.3). Residual plots for those results were visually examined and showed no clear signs of deviations from homoscedasticity or normality. *p* values for the nested mixed effects models were calculated via likelihood-ratio tests. Post hoc tests to estimate the slope of insight per ROI were

estimated using the ggeffects (v1.6.0) and model_based (v.0.8.8) packages. Unless otherwise specified, p values are always reported as two-tailed.

**Insight memory effect**. First, two binomial mixed effect models were estimated with correct subsequent memory (binary) as dependent measure and a factor that included all trials differentiating between three conditions—not solved trials, correctly solved trials with high (HI-I) as well as low insight (LO-I)—as independent measure controlling for the run order (1–4) and including two random intercepts for subject and item (see Eq. (2) below). The baseline model excluded the independent measure.

$$\text{Subsequent memory} \sim \text{run} + (1|\text{ID}) + (1|\text{item}) + \mathcal{E} \quad (1)$$

$$\text{Subsequent memory} \sim \text{run} + \text{insight} + (1|\text{ID}) + (1|\text{item}) + \mathcal{E} \quad (2)$$

Second, this analysis was repeated, incorporating response time as a covariate of no interest in both models and exchanging the categorical with the continuous insight variable (this excluded unsolved trials). Due to the relatively lower trial count for the insight-memory analyses (see Fig. S5), we used a non-parametric bootstrapping approach with 1000 resamples to estimate the odds ratios and their 95% confidence intervals for the effects of interest. This was implemented using the boot function (v.1.3-30) in R, where the model was refitted to each resampled dataset providing robust estimates of effect size and confidence intervals.

To eliminate potential confounds related to encoding time differences (e.g., HI-I trials were solved faster, giving participants more time to encode their solution), we conducted a preregistered control experiment (https://aspredicted.org/xx7hv.pdf, 11/28/2023). In this experiment, participants followed the same procedures as the fMRI study using the same 120 Mooney images, but the stimulus disappeared when the solution button was pressed, eliminating encoding time differences. Details to this experiment can be found in "Further control analyses to account for the influence of solution time" in the Supplementary Methods and Fig. S7.

**fMRI data preprocessing**

Functional as well as structural images were preprocessed with fMRI-Prep version 20.2.5 using default parameters[88]. Each T1w (T1-weighted) volume was corrected for INU (intensity non-uniformity) using N4BiasFieldCorrection v2.1.0[89] and skull-stripped using antsBrainExtraction.sh v2.1.0 (using the OASIS template). Spatial normalization to the ICBM 152 Nonlinear Asymmetrical template version 2009c [ref. 90; RRID:SCR_008796] was performed through nonlinear registration with the antsRegistration tool of ANTs v2.1.0 [RRID:SCR_004757], using brain-extracted versions of both T1w volume and template. Brain tissue segmentation of cerebrospinal fluid (CSF), white-matter (WM) and gray-matter (GM) was performed on the brain-extracted T1w using fast [FSL v5.0.9, RRID:SCR_002823].

Functional data was motion corrected using MCFLIRT (FSL v5.0.9)[91]. Distortion correction was performed using an implementation of the TOPUP technique[92] using 3dQwarp [AFNI v16.2.07]. This was followed by co-registration to the corresponding T1w using boundary-based registration[93] with six degrees of freedom, using bbregister (FreeSurfer v6.0.1). Motion correcting transformations, field distortion correcting warp, BOLD-to-T1w transformation and T1w-to-template (MNI) warp were concatenated and applied in a single step using antsApplyTransforms (ANTs v2.1.0) using Lanczos interpolation.

Physiological noise regressors were extracted by applying anatomical CompCor (aCompCor)[94]. Here, six components were calculated within the intersection of the subcortical mask and the union of CSF and WM masks calculated in T1w space, after their projection to the native space of each functional run. Framewise displacement[95] was calculated for each functional run using the implementation of Nipype.

To increase the signal to noise ratio, the preprocessed functional data in standard as well as in native space were smoothed with an 2 mm full width half maximum Gaussian kernel (FWHM) for the ROI-based analysis and with an 8 mm FWHM for the exploratory whole-brain analysis using SPM12 (Welcome Department of Cognitive Neurology, London, UK).

**Multivariate fMRI data analysis—representational similarity analyses**

We hypothesized that (1) RC should be associated with changes in distributed, i.e., multivoxel, patterns from pre to post solution, (2) those patterns should start representing the meaningful solution object after solution and (3) visual RC in Mooney images (perceptual regrouping of black/white contours leading to sudden object recognition) should occur in regions involved in visual object recognition along the visual ventral pathway such as the temporal-occipital [pITG], anterior [aITG] and posterior [mITG] Inferior Temporal Gyrus; inferior Lateral Occipital Cortex [iLOC]; anterior, posterior, occipito-temporal [toFusG] and occipital [oFusG] Fusiform Gyrus[42,96,97]. The masks for those ROIs were extracted from the FSL Harvard-Oxford (HO) Atlas[98]. Note, to reduce the number of Fusiform Gyrus ROIs but respect functional differences[99], we merged this region into only two ROIs: anterior [aFusG] and posterior [pFusG] Fusiform Gyrus. To create the aFusG, the anterior and posterior Fusiform Gyrus ROIs from the HO Atlas were merged into one. To create the pFusG, the toFusG and oFusG ROIs were merged into one.

We used RSA to test whether and which ROIs along the visual ventral pathway (VOTC areas) exhibit those properties during insightful problem solution. We first extracted the multivoxel pattern in the above named ROIs per participant, per Mooney image and for a pre as well as post-solution time point. Subsequently, we conducted two different types of representational similarity analysis: (1) pre to post-solution similarity to identify ROIs showing a stronger decrease in correlation for HI-I than LO-I trials and (2) a model-based RSA[20,37] where distributed activity patterns were correlated with the output of two very distinct neural network models, AlexNet and Word2Vec.

Extraction of ROI-wise multivoxel patterns: to obtain multivoxel patterns, we first estimated beta values in each voxel for each pre and post solution event. Due to the short response time in particular for HI-I events, the pre and post solution regressors would be highly collinear, i.e., explain shared variance of the time-series, which results in unreliable parameter estimation and an hence an underestimation of the similarity of activity patterns[100]. Therefore, we conducted two separate first-level models—one for the pre and one for the post solution time points in which each event was represented by an individual regressor (its onset convolved with the HRF). Pre-solution onsets were 0.5 s after onset of the Mooney images and post-solution onsets were the time point of the button presses; both events were modeled with a duration of 1 s. Because there was no button press for unsolved trials during the stimulus presentation, we created a jittered onset regressor for the "post-solution stage" of the unsolved trials that on average matched the response time of the solved trials (~3.7 s). These two univariate single-trial models were conducted as general linear models as implemented in SPM. Due to the sluggish HRF, one would expect a higher intrinsic auto-correlation of the activity patterns of pre and post events when they occur close in time which is on average more often the case for HI-I than LO-I events. Importantly though, this auto-correlation rather counteracts the hypothesized greater decrease in similarity in HI-I trials because they were solved faster and therefore do not confound the results. We added an additional nuisance regressor for all remaining button presses (related to the insight ratings and selecting a solution category) for one second starting with the respective button press. Additionally, we separately modeled six

motion parameters and separately modeled the mean for each of the four runs. The time series were corrected for baseline drifts by applying a high-pass filter (128 s) and for serial dependency by an AR(1) autocorrelation model.

Subsequently, the corresponding pre and post solution activation patterns for each Mooney image from the corresponding beta images per ROI (iLOC, aFusG, pFusG, aITG mITG, pITG) were extracted and vectorized. Those ROI-wise multivoxel patterns were used for (1) Pre to post solution RSA and (2) Pre to post solution model-based RSA. All analyses were conducted using in-house MATLAB scripts.

**Pre to post solution RSA.** To compare the trial-wise change in multivoxel pattern from pre to post solution, we first matched the extracted pre and post solution activation patterns for each Mooney image from the corresponding beta images per ROI into a pre to post solution vector pair and subsequently correlated it (Spearman's Rho) (see Fig. 3A).

For statistical analysis, a linear mixed effects model was applied estimating the multivoxel pattern similarity between the pre and post solution phase (dependent variable) as a function of the insight (continuous) for correctly solved trials and respective ROI. Additionally, response time and the run order (1–4) were added as covariate of no interest, and we included only trials where pre and post solution phases were at least 2 s apart to reduce overlap of the hemodynamic response. Item and subject variables were estimated as random intercepts. To estimate differences in multivoxel pattern similarity between ROIs as a function of insight, another model was estimated including an interaction term between ROI and insight (see Eqs. (3)–(5) below):

$$PPSS \sim RT + run + ROI + (1|ID) + (1|item) + \mathcal{E} \qquad (3)$$

$$PPSS \sim RT + run + ROI + insight + (1|ID) + (1|item) + \mathcal{E} \qquad (4)$$

$$PPSS \sim RT + run + ROI*insight + (1|ID) + (1|item) + \mathcal{E} \qquad (5)$$

Note. PPSS = Pre-Post Solution Similarity; RT = response time.

For the resulting $p$ values, we used Bonferroni-corrected alpha levels because we did not have a hypothesis pertaining to which ROI may exhibit insight-related changes in multivoxel pattern similarity.

**Pre to post solution model-based RSA.** Here, representational similarity between the activity pattern elicited by the stimuli is compared to the computationally model-based representational similarity between the same stimuli. This analysis involves four steps: (1) creating representational similarity matrices for each ROI based on the correlation of the multivoxel activity patterns elicited by the stimuli, (2) creating a model-based similarity matrix based on the correlation of the output of the computational model for each stimulus (3) correlating the model-based similarity matrix with the ROI-specific representational similarity matrices for each time point (4) associating the brain-model correlation for each trial with insight or insight*memory. Those steps are further detailed below.

The first step involved creating a *Neural activity-derived RSM* (120×120) for each participant, ROI (iLOC, aFusG, pFusG, aITG, mITG, pITG) and time point (pre and post solution), representing the similarity in the multivoxel activity pattern across stimuli. That is to say, each cell in a matrix represents the Spearman correlation between the vectorized multivoxel patterns of two Mooney images (see Fig. 3B1).

The second step involved creating a *Model RSM* consisting of pairwise *conceptual* similarity values for each Mooney image pair represented in a 120 × 120 matrix. Here, we assumed that *conceptual* similarities between the Mooney objects increase after problem solution (=object identification). To assure robustness of the results, we used two different methods to quantify conceptual similarity between the Mooney images–a deep convolutional network (DNN) AlexNet[49,101,102] and a Word2Vec model[45,48,103] resulting in two different conceptual *Model RSMs*. Conceptual similarity via the DNN AlexNet *Model RSM* is based on categorical features[49]. DNNs consist of layers of convolutional filters that are trained to classify images into categories with a high level of accuracy. While filters from early layers predominantly detect simple visual features, later layers organize items by their categorical features[104]. According to previous research, DNNs like AlexNet model visual representations along the ventral visual pathway[101,102] and outperform traditional theoretical models of the ventral visual pathway (e.g., HMAX, object-based models)[105,106] in their capacity to identify specific objects with the appropriate category-level label. They may therefore be particularly well suited to investigate RC of visual objects using Mooney images. Here, we used the pre-trained 8-layer DNN, the AlexNet, which was successfully trained to classify 1.2 million images into 1000 categories[49]. AlexNet consists of eight layers, the first five are convolutional (some are followed by max-pooling layers) and the last three are fully connected layers. Conceptual similarity between the Mooney objects was quantified as follows: First, the actual non-abstracted, colored object images (not their abstracted Mooney counterpart) were entered into the pre-trained AlexNet and the activations of its penultimate layer for each image were extracted and converted into one activation vector (note, activation from this layer can be regarded as conceptual because it is the last layer before the images are explicitly categorized into the trained categories)[106,107]. Subsequently, for each pair of images, the similarity (Spearman correlation) was computed between the activation vectors resulting in a 120 × 120 matrix (=*Model RSM*). For a schematic visualization of this process, see Fig. 3B2).

In contrast, in the Word2Vec model, pair-wise stimuli similarity is based on cosine similarity derived from statistical co-occurrences in text data via a pre-existing word embedding model as previously published (refs. 45,108 for more details). Word embeddings represent words as dense numerical vectors derived from a neural network being trained with a huge text corpus using a word2vec algorithm (see ref. 48. Cosine similarity is defined as the angle between two 300 dimensional vectors that represent both stimuli (concepts, e.g., "tiger" and "dog") in this vector space. The *Model RSM* that was created with Word2Vec consisted of pair-wise conceptual similarity values for each Mooney concept (word) represented in a 120 × 120 matrix.

For the third step, the item-wise rows (one item = one Mooney image object) from the *Neural-activity-derived RSM* for each participant, each time point and each ROI were partially correlated (Pearson's r) with the same item-wise rows of each *Model RSM* (AlexNet and Word2Vec). Note, to control for the confound of having presented the Mooney images in different scans, i.e., blocks and sessions, those item-wise row correlations between the brain and model data were partialled out by a variable indicating the respective run order (1–4). Such an item-wise approach differs from the classical method of assessing such second-order correlations between brain and model activation patterns[109], which typically relate the entire item × item matrix at once (note, this item-wise approach has been successfully applied elsewhere[37]. Furthermore, the item-wise row correlations were further filtered into solved and unsolved Mooney images because we assumed that only solved Mooney images (successful object identification) should exhibit conceptual representational similarity among each other. The resulting representational strength ("2nd order correlation") identifies brain regions that process and/or store conceptual representations emphasizing either categorical visual features (AlexNet) or similar meaning based on co-occurrences (Word2Vec). Hence, higher representational strength values in a brain region indicate higher conceptual similarity (see Fig. 3B3).

The fourth step involved conducting a statistical analysis similar to the *pre to post solution similarity analysis*, using mixed effects

models in a nested way to estimate the effect of changes in representational similarity (i.e., representational strength or brain-model fit) as a function of insight after correct solution, ROI and time (see equations below). Time was a factor that indicated the pre (0.5 s after stimulus presentation) or the post solution (solution button press) phase. Note, the ROI factor only contained pFusG and iLOC because only those visual brain areas exhibited changes in pre to post solution similarity (see 1.). Additionally, response time (RT) and run order were added as covariate of no interest and we included only trials where pre and post-solution phases were at least 2 s apart to minimize overlap of the hemodynamic response. Next to the ROI*insight interaction, we additionally modeled a 3-way interaction (ROI*insight*time) to test which ROI exhibits insight-related changes in representational similarity from pre to post-solution (IV., see equations below):

$$\text{Rep} \cdot \text{Strength} \sim \text{RT} + \text{run} + \text{time} + \text{ROI} + (1|\text{ID}) + (1|\text{item}) + \mathcal{E} \quad (6)$$

$$\text{Rep} \cdot \text{Strength} \sim \text{RT} + \text{run} + \text{time} + \text{ROI} + \text{insight} + (1|\text{ID}) + (1|\text{item}) + \mathcal{E} \quad (7)$$

$$\text{Rep} \cdot \text{Strength} \sim \text{RT} + \text{run} + \text{time} + \text{ROI*insight} + (1|\text{ID}) + (1|\text{item}) + \mathcal{E} \quad (8)$$

$$\text{Rep} \cdot \text{Strength} \sim \text{RT} + \text{run} + \text{time*ROI*insight} + (1|\text{ID}) + (1|\text{item}) + \mathcal{E} \quad (9)$$

Note. Rep.Strength = Representational Strength, i.e., brain-model fit.

Finally, we tested whether (a) differences in pre to post solution multivoxel pattern similarity as well as (b) the representational strength of the solution object after correct solution in those ROIs that show all described properties of RC (pFusG, iLOC) are related to insight-related better memory. To test this, we performed another set of nested model comparisons with (a) pre to post solution similarity (PPSS) and (b) the changes in representational strength (Rep.Strength: AlexNet or Word2Vec) from pre to post solution as dependent variables and ROI as well as an interaction between (time and) insight and memory as independent factors (see equations below). Note, only pFusG and iLOC exhibited all properties of RC, for this reason we only estimated the described models for those two ROIs. Response time and the run order were entered as covariates of no interest, and items and subject were estimated as random intercepts. To test for differences in insight-related memory between ROIs, another model with ROI*insight*memory interaction term was additionally modeled ((13), see equations below). To enhance statistical robustness for the insight-memory analyses (see Fig. S5), we implemented permutation tests to derive p-values from comparing nested random effects models (*permlmer*) using the *predictmeans* package (v.1.1.0) in R, with 999 permutations[56]:

$$\text{PPSS} \sim \text{RT} + \text{run} + \text{ROI} + \text{insight} + (1|\text{ID}) + (1|\text{item}) + \mathcal{E} \quad (10)$$

$$\text{PPSS} \sim \text{RT} + \text{run} + \text{ROI} + \text{insight} + \text{memory} + (1|\text{ID}) + (1|\text{item}) + \mathcal{E} \quad (11)$$

$$\text{PPSS} \sim \text{RT} + \text{run} + \text{ROI} + \text{insight*memory} + (1|\text{ID}) + (1|\text{item}) + \mathcal{E} \quad (12)$$

$$\text{PPSS} \sim \text{RT} + \text{run} + \text{ROI*insight*memory} + (1|\text{ID}) + (1|\text{item}) + \mathcal{E} \quad (13)$$

Note. PPSS = Pre-Post Solution Similarity; RT = solution time.

$$\text{Rep} \cdot \text{Strength} \sim \text{RT} + \text{run} + \text{ROI} + \text{time} + \text{insight} + (1|\text{ID}) + (1|\text{item}) + \mathcal{E} \quad (14)$$

$$\text{Rep} \cdot \text{Strength} \sim \text{RT} + \text{run} + \text{ROI} + \text{time} + \text{insight} + \text{memory} + (1|\text{ID}) + (1|\text{item}) + \mathcal{E} \quad (15)$$

$$\text{Rep} \cdot \text{Strength} \sim \text{RT} + \text{run} + \text{ROI} + \text{time} + \text{insight*memory} + (1|\text{ID}) + (1|\text{item}) + \mathcal{E} \quad (16)$$

$$\text{Rep} \cdot \text{Strength} \sim \text{RT} + \text{run} + \text{ROI} + \text{time*insight*memory} + (1|\text{ID}) + (1|\text{item}) + \mathcal{E} \quad (17)$$

$$\text{Rep} \cdot \text{Strength} \sim \text{RT} + \text{run} + \text{ROI*time*insight*memory} + (1|\text{ID}) + (1|\text{item}) + \mathcal{E} \quad (18)$$

Note. Rep. Strength = Representational strength of solution object at post solution (button press); RT = solution time.

## Univariate fMRI data analysis−single trial analysis

The amygdala and hippocampus have been previously suggested to be involved in the evaluative component of insight[25,27,64,110]. Therefore, we used those brain areas as ROIs for the univariate analysis. Statistical analyses were conducted with the data from the single-trial analyses containing a beta estimate of univariate bilateral amygdala and anterior as well as posterior hippocampus activity for the solution button press at each trial (see "Method" section, Multivariate fMRI data analysis−multivoxel pattern similarity analysis: extraction of ROI-wise multivoxel patterns).

The left and right ROI masks for amygdala and hippocampus were extracted from the FSL Harvard-Oxford (HO) Atlas[98]. Furthermore due to functional differences, the hippocampus mask was divided into an anterior and posterior part based on the conventional landmark of the uncal apex ($y = -21/22$ in MNI space)[111].

To statistically compare insight-related mean activity differences in amygdala and hippocampus (ROI variable) during solution (time point of solution button press), we estimated three nested mixed effect models (see Eqs. (19)−(21)) including the averaged amygdala, anterior and posterior hippocampus beta value per trial as dependent variable and subjects as well as items as random intercepts. To control for potential differences between the sessions and blocks, response time (RT) and a factor specifying the order of the respective run order (1–4) were added as covariates of no interest. Insight (continuous) for correctly solved trials served as the independent measure. To calculate insight-related better memory on amygdala and hippocampal activity during solution, we additionally estimated memory for the correctly solved trials and their interaction with insight and ROI (see Eqs. (22)−(24) below):

$$\text{Beta values} \sim \text{run} + \text{RT} + \text{ROI} + (1|\text{ID}) + (1|\text{item}) + \mathcal{E} \quad (19)$$

$$\text{Beta values} \sim \text{run} + \text{RT} + \text{ROI} + \text{insight} + (1|\text{ID}) + (1|\text{item}) + \mathcal{E} \quad (20)$$

$$\text{Beta values} \sim \text{run} + \text{RT} + \text{ROI*insight} + (1|\text{ID}) + (1|\text{item}) + \mathcal{E} \quad (21)$$

$$\text{Beta values} \sim \text{run} + \text{RT} + \text{ROI} + \text{insight} + \text{memory} + (1|\text{ID}) + (1|\text{item}) + \mathcal{E} \quad (22)$$

$$\text{Beta values} \sim \text{run} + \text{RT} + \text{ROI} + \text{insight*memory} + (1|\text{ID}) + (1|\text{item}) + \mathcal{E} \quad (23)$$

$$\text{Beta values} \sim \text{run} + \text{RT} + \text{ROI*insight*memory} + (1|\text{ID}) + (1|\text{item}) + \mathcal{E} \quad (24)$$

Note. RT = response time.

The association between insight and amygdala-hippocampal brain activity was tested by comparing the nested models I.–IV. via log-likelihood tests. To enhance statistical robustness for insight-related memory on brain activity, *p* values from comparing nested models IV–VI were obtained from permutation tests (*n* = 999) using the *predictmeans* package (v.1.1.0)[56].

### Univariate fMRI data analysis in standard space—control analysis

To exclude potential normalization artifacts due to the small volume of the amygdala and hippocampal ROIs, we additionally conducted another ROI analysis in subject space. Functional and structural data were preprocessed using fMRIprep in the same way as for the single trial analysis in standard space (see further above*)*, with the exception that data were resampled into the individual's anatomical reference generated with the T1 weighted images.

**First-level analysis.** To evaluate the relationship between insight and amygdala-hippocampal activity, we examined if brain activity shows parametric modulation based on the level of insight (continuous). This investigation was carried out within the context of general linear models (GLMs), employing a mass univariate approach as implemented in SPM with standard parameter settings. A total of 25 separate regressors were created (the numbers in brackets [1–25] indicate the number of the respective regressors including first and second temporal derivative): We modeled the (correct) solution button presses for the Mooney images [1:3]. The Mooney solution button presses were parametrically modulated with the scaled and mean centered insight (continuous) variable [4:6] as well as a scaled and mean centered solution time variable [7:9]. To reduce variance from the implicit baseline, we added two nuisance regressors; one for all solution button presses of the anagram trials [10:12] (which are not reported here) and another one for all remaining button presses [13:15] (related to the insight ratings and selecting a solution category). All those events were modeled for one second starting from the button press and convolved with the canonical hemodynamic response function (HRF). Additionally, we separately modeled six motion parameters [16:21] and separately modeled the mean for each of the four runs [22:25]. The time series were corrected for baseline drifts by applying a high-pass filter (128 s) and for serial dependency by an AR(1) autocorrelation model.

Bilateral amygdala and the anterior and posterior hippocampus ROIs were first resampled into subject space (i.e., creating individual subject anatomical-based ROIs) using the ANTs function *antsApplyTransforms*[112]. Subsequently, we used the *marsbar* toolbox using its default parameter settings to extract the averaged beta values representing the respective onset regressors from the ROIs per session and block[113].

**Second-level analysis.** To investigate the insight-related parametric modulation of the beta values in bilateral amygdala and hippocampal brain activity, we estimated two nested mixed effect models as depicted ((25)–(26)) below including the averaged beta value per ROI as dependent variable and participants as random intercepts. To control for potential differences between the sessions and blocks, a factor specifying the order of the respective runs (1–4) was always added as a covariate of no interest. The respective ROI served as the independent measure.

$$\text{Beta values} \sim \text{run} + (1|\text{ID}) + \mathcal{E} \qquad (25)$$

$$\text{Beta values} \sim \text{ROI} + \text{run} + (1|\text{ID}) + \mathcal{E} \qquad (26)$$

### Multivariate fMRI data analysis—functional connectivity and graph analysis

We expected insight-related increased functional connectivity between those areas that process the solution object (VOTC) and those that likely evaluate it such as the amygdala and hippocampus. To estimate functional connectivity (FC) between those areas, we used the fMRIprep preprocessed fMRI data (see above) and first denoised and subsequently analyzed it in CONN [RRID:SCR_009550]—an open source connectivity toolbox (version 21.a)[53].

**Denoising.** To remove confounding effects to the estimated BOLD signal for each participant in each voxel and at each time point, i.e., session and block, CONN's implemented anatomical component-based noise correction procedure (CompCor) was used. Those confounding effects involve six subject-motion parameters including their first-order derivatives, identified outlier scans, constant or linear condition effects as well as noise components from areas of cerebral white matter and cerebrospinal fluid. Note, to avoid the risk of artificially introducing anticorrelations (negative connectivity) into the FC estimates, no global signal regression was applied[114]. Finally, the resulting time series were band-pass filtered to 0.008–0.09 Hz.

**First-level analysis.** FC during high versus low insight conditions (*HI-I > LO-I*) was estimated. We were only interested in ROIs processing the solution object and therefore constructed a specific *solution network* comprising left and right aFusG, pFusG, aITG, mITG, pITG, iLOC including left and right amygdala and anterior as well as posterior hippocampus (18 ROIs in total). Amygdala and hippocampus were included because we were additionally interested in whether those brain areas that showed insight-related differences in univariate BOLD activity are also part of this *solution network*. All ROIs taken from the FSL Harvard Oxford Atlas were identical to the ones already used for univariate and the other multivariate analyses but analyzed separately by hemisphere[54,55]. The time series for each ROI were acquired by averaging the BOLD time series of each voxel belonging to the respective ROI. ROI-to-ROI connectivity matrices (*solution network*: 18 × 18) were computed for each participant (31), for each condition (1) *HI-I vs. LO-I*, each session (2) and each block (2) separately. Each element in each connectivity matrix represents the strength of FC between a pair of ROIs and was defined as the Fisher-transformed bivariate correlation coefficient between the preprocessed and denoised BOLD time series of two ROIs.

**Second-level analysis.** To estimate FC for the contrast *HI-I > LO-I* in the *solution network*, the within-subjects contrast in the GLM was specified as follows *y* - 1*HI-I − 1*LO-I + *ε*. For statistical inference, the 18 × 18 ROI connectivity matrix was thresholded at the connection-level at *p* < 0.05 (p-uncorrected, one-sided [we only expected positive connectivity]) and at the network-level at *p* < 0.05 (ROI-p, FWE corrected). Note, for cluster-level inference we adopted the Network Based Statistics (NBS) approach[115].

To further explore whether FC in the *solution network* also predicts insight-related better memory, we estimated another first level analysis identical to the one described above with the only difference that four different conditions were estimated: (1) high insight—remembered trials [*HI-I-Rem*], (2) high insight—forgotten trials [*HI-I-Forg*], (3) low insight—remembered trials [*LO-I-Rem*] and (4) low insight—forgotten trials [*LO-I-Forg*]. For the second level analysis, we specified the following within-subjects contrast in the GLM: *y* - 1*HI-I-Rem − 1*HI-I-Forg − 1*LO-I-Rem + 1*LO-I-Forg + *ε*.

**Graph-theoretical measures.** To further characterize efficient information integration during insight we computed several network measures quantifying functional integration (increased global efficiency, reduced average path length), segregation (increased local

efficiency and clustering coefficient) and centrality (degree) of the *solution network* and its individual ROIs with the contrast *HI-I > LO-I* as implemented in CONN[53]. Note, the description and mathematical formulas for each graph measure is in Table S2.

All ROI-graph measures were based on functional connectivity values of our pre-defined non-directed *solution network* (18 × 18 Fisher transformed correlation matrix). For each participant and condition (*HI-I, LO-I*) a series of adjacency matrices are then computed by thresholding the respective Fisher transformed correlation matrix by different correlation coefficients ranging from $r = 0.10$ to $r = 0.8$ in steps of 0.1. Note, only positive correlations were included for better interpretation of the subsequent graph measures[52]. From those resulting adjacency matrices, several graph-theoretical network measures of interest were subsequently computed for the contrast *HI-I > LO-I* addressing topological properties of each ROI within the matrix but also of the entire *solution network*. Importantly, to demonstrate the robustness of the results rather than just picking an arbitrary threshold, we aggregated across the different adjacency matrices (acquired via those different correlation coefficient thresholds) to compute the respective graph measure of interest. Note, we tested one-sided due to directed hypotheses.

During the preparation of this work the authors used chatgpt 3.5 in order to improve the language and readability. After using this tool, the authors reviewed and edited the content as needed and take full responsibility for the content of the publication.

### Reporting summary
Further information on research design is available in the Nature Portfolio Reporting Summary linked to this article.

## Data availability
The stimulus material as well as raw behavioral and aggregated uni-variate and multivariate fMRI data generated in this study have been deposited in the Zenodo database (https://doi.org/10.5281/zenodo.14743957). The raw fMRI data are not publicly available due to data privacy regulations. However, de-identified pre-processed fMRI data can be made available upon request from the corresponding author, subject to institutional and ethical approvals.

## Code availability
The analysis code is publicly available and can be accessed in the Zenodo database https://doi.org/10.5281/zenodo.14743957.

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

## Acknowledgements

M.B. and R.C. were funded by the Einstein Foundation Berlin (EPP-2017-423, R.C.), National Institute of Health (NIA 1RF1AG066901, R.C.) and M.B. was additionally funded by the Sonophilia Foundation. We would like to express our deepest gratitude to Luise Oehler and Teuta Dzaferi for their substantial assistance in data assessment and John D. Haynes, Simon W. Davis, Cortney Howard, Loris Naspi and Michael Krause for useful discussions on experimental design and data processing analysis.

## Author contributions

M.B. and R.C. conceived of the study. M.B. acquired, analyzed the data and wrote the manuscript. T.S. and R.C. supervised data analysis. M.B., T.S. and R.C. contributed to revising the manuscript.

## Funding

## Competing interests

The authors declare no competing interests.
