## [Transparent Peer Review File · Nature Communications]

Insight Predicts Subsequent Memory via Cortical Representational Change and Hippocampal Activity

Corresponding Author: Dr Maxi Becker

Version 0:

Reviewer comments:

Reviewer #1

(Remarks to the Author)

General Comments

This study investigates the neural mechanisms underlying insight problem solving and its impact on memory. The authors employ multivariate pattern analyses to examine representational change in visual areas, going beyond traditional univariate approaches. Moreover, they use Representational Similarity Analysis (RSA) to track pattern similarity changes, as well as model-based RSA with AlexNet and Word2Vec to assess representational shifts associated with conceptual reorganization.

The results support key hypotheses about representational change (RC) in the visual cortex during insight and its connection to memory encoding. Notably, evidence points to robust RC in specific ventral visual stream regions during high insight solutions, with changes in patterns and enhanced conceptual representation confirmed by model-based RSA results. Insight was also found to modulate amygdala and hippocampal activity, with effects holding when controlling for potential confounding factors. Furthermore, functional connectivity and graph theory findings indicate that visual, hippocampal, and amygdala regions form an interconnected network during insight.

Overall, the experimental design and multivariate fMRI analyses provide an excellent basis to elucidate the neural mechanisms underlying insight and its impact on memory formation. A key strength of this work is testing of specific neural hypotheses. I commend the authors for approaching their analyses in such a systematic and rigorous way.

I found very little to criticize about the methods and experimental design. In my view, this is among the most rigorously conducted and innovative cognitive neuroscience studies of insight in the field. The paper is very well written.

My main (minor) question concerns the interpretation of the results, and whether the process studied should be called “creative”, or rather a basic perceptual restructuring. Further details are below.

Specific Comments

“The neural mechanisms driving creative problem-solving still remain largely unknown”; creative problem solving has been studied quite a bit, including by the co-authors. Is this really the case?

The mention of “sudden” representation changes in the visual cortex raises a query considering the study employs fMRI, which inherently lacks the temporal resolution to precisely capture “suddenness”; perhaps an alternative descriptor could be more accurate.

I was a little confused by the distinction between “cognitive and evaluative” components. Doesn’t evaluation involve cognition to some extent?

The task paradigm was well reasoned and elegant for invoking insight. But I wonder if the Mooney image task has been

validated as a measure of *creative* problem solving? For example, do people who can more quickly/accurately identify the Mooney image tend to perform better on other creativity tasks? My question here is whether this task really measures an aspect of creativity (or even problem solving), or rather some basic perceptual/restructuring process. Are we really talking about high-level creative problem solving if everything happens at a perceptual level? I'm not entirely sure that this task measures 1) problem solving and 2) creativity. That doesn't say anything about the quality of the work (which is excellent) – it just speaks to how we interpret the results. I don't doubt that insight was measured here. And I recognize that definitions of creativity vary.

One limitation in my view is the restriction to Mooney images and the omission of a verbal task, which is often a more prevalent measure of insight.

Another limitation is, to my knowledge, the absence of psychometric validation of Mooney image solving as a metric for creativity.

Otherwise, I enjoyed reading this paper and congratulate the authors on an outstanding contribution to the field.

Reviewer #2

(Remarks to the Author)

Becker et al. present a single experiment in which participants recognize degraded images (Mooney figures, or camouflage pictures) during an fMRI scan, self-reporting various dimensions contributing to high- or low-insight ratings; then 5(?) days subsequently perform a memory test. fMRI signal during initial object recognition was associated with the insight-related ratings in several interpretable brain regions (e.g., related to object recognition, reward, memory), as well as increased connectivity among these regions for the high-insight trials. The authors reported changes in the relevant areas from initial stimulus presentation to the moment of recognition, and some of these changes were greater on trials for which participants reported (through the 3-dimension rating) high-insight compared to low-insight trials. Moreover, some of the fMRI markers were associated with improved recognition of the stimuli 5 days later.

Overall, this is an ambitious, challenging, but very well-motivated study. Finding neural markers (both areas and connectivity) that relate to insight visual recognition, to (their definition of) restructuring, and to later memory could all be important findings. The overall structure of the paradigm is well-reasoned and mostly well-described. The results and conclusions seem to make sense... but unfortunately the challenges of such a design have not – yet – been overcome.

The main difficulties facing this study are the low number of [true] observations per condition cell and, related but even more challenging, potentially highly impactful confounds across their main conditions of interest. Specifically:

1) There are far fewer true object recognition trials (in the scanner) with low-insight than with high-insight (this difference is compounded at memory). This results in lower power (lower # of trials = lower signal to noise ratio) to detect both fMRI signal and changes across conditions for the low-insight trials. The actual numbers of such trials aren't provided in the report (that I see), and have to be derived indirectly; even then, it's not always clear for various reasons. On top of that, there is the ambiguity of whether participants actually identified objects scored as "correct", since this was only indicated in a 4-choice task (hence 25% chance rate, before correcting for possibility of narrowing the choices due to logical guesses – e.g., if the object appears as all straight lines, it's probably not a mammal).

2) Trials were "solved" much more quickly (mean 2.75sec) when reported to be with high- than with low-insight (5.08sec).

2a. This creates an untenable confound when assessing fMRI activation. Some trials (high-insight) unfold with stimulus presentation and solution highly conflated; on other trials (low-insight), the solution was much less conflated with stim onset, but possibly conflated with the end of the image presentation (which was always at 10sec, if I understand p. 37 correctly). Also, longer time-on-task (effort at recognizing, or early visual processes) could yield bigger fMRI signal effects in some areas (and smaller in others, either in absolute terms or relative to the whole span of the trial).

2b. Another issue here is that VERY quick responses may simply reflect simple object recognition (they immediately see a puppy, in a slightly blobby picture), rather than "solving" in any normal usage; furthermore, these are more likely to get rated as "high-insight" (due to suddenness and certainty, and perhaps surprise that they're getting an easy trial). But are they true insight-solving? Some in the field (including us) have taken to eliminating the rapid responses from analyses (see e.g. Cransford & Moss, 2012 J. Prob Solving).

2c. The time difference in solving also creates a confound for the subsequent memory task. This could be viewed two ways. If participants really only "studied" the image prior to "solving", then they had less study time for high-insight than low-insight trials. This would suggest the better memory for high-insight trials is even better! OTOH, if participants continued to examine the image as long as it remained on the screen (again, a constant 10 sec), then they had longer to examine the OBJECT (10s – 2.75 = 7.25 sec) for high-insight trials than for low-insight trials (10s-5.08 = 4.92sec). I.e., they had more time to think of the object, to see how the image details comprise the object, etc., for high- than for low-insight trials.

What to do? I recognize the very clear challenges of implementing such a paradigm, and the important nature of the questions being asked. It's also possible that the confounds outlined above do not adversely affect results or interpretations, but I don't see an obvious way to be sure – as the paper now stands.

These confounds are very important, and must be dealt with better. I know the authors report that they included reaction time and (sometimes) accuracy as “covariates of no interest”, but I don’t think that is sufficient. One possible path is to re-sample the number of trials. Specifically, drop the fastest “high-insight” trials (which were probably included a fair number of instant recognition trials anyway), and the slowest “low-insight” trials, until you have data sets that are more comparable (I’d love them to have equal means and distributions, but that’s probably not achievable with this dataset). If the same results occur, you have more assurance they are not due to the time confound.

Problem is, there are already so few accurate low-insight trials (their reporting of exactly how much data [trials/sub per condition cell] remains unclear), so it might be hard to examine meaningful subsamples.

Because the paper is potentially important, it could merit publication IF the authors address the issues (with evidence, hard stats on the least-confounded data they can provide) as much as possible, but are also completely transparent and forthright about the potential weaknesses – low # of trials and possible confounds. When I say “forthright” I mean such info is given in passing or in a footnote, but is seriously discussed. (This would be a welcome trend for the whole field, not just this paper – too many papers try to sell conclusions as rock solid when they are actually hints that need further verification through additional papers being published).

Beyond the above issues, there are some minor interpretive ones, easily handled in revisions.

Conceptually, is this type of vis-aha the same as other insights, such as semantic reinterpretation? This isn’t a big issue for me, and there have long been some connections between the phenomena (including by Gestalt psychologists). But it would be worth discussing, and some relevant literature (e.g., Bar et al., 2001, but others I list below) that should probably be noted and discussed in relation to the specific results uncovered (e.g., VOTC, LOC & Fusiform areas). There are certainly similarities between vis-aha! and other insights, and many people have treated it as more or less the same as other insights. But there could be some important differences. E.g., is it the same to go from basic visual features (in this case, blobby black or white components) to object recognition, as it is to go from a standard but incorrect semantic association or problem representation to a less salient one? For these visual stimuli, there isn’t necessarily an initial misinterpretation to be overcome, or a dominant solving path that leads the solver/viewer astray. In fact, this paradigm is similar to straightforward object recognition paradigms (e.g., Bar et al., Neuron, 2001; Doniger et al., JoCN, 2000), although those paradigms gradually increased the amount of visual information contributing to object recognition (but still see some top-down effects, e.g. Bar et al., 2006). Much of the current discussion ignores the object recognition literature or glosses over the distinction between different types of insight. This is not a fundamental problem, nor is it unique to these authors. I’d just like to see both ideas noted in the manuscript (initially framing the paradigm as eliciting “a type of insight”, noting it shares some but not all characteristics of other insights; AND noting the similarity to these object recognition literature like Bar, [etc]. In this paper it matters because some of the key DVs (including fMRI regions) could be pretty closely yoked to object recognition.

The issue of demonstrating restructuring is important, and it’s great that this paper tries to provide some evidence. Many/most studies (including most of ours) have simply assumed or posited that restructuring occurs. There have been some prior attempts to provide evidence, including using graph theory. One had subjects make semantic relatedness judgments and found that subjects who had immediately prior solved problems with related concepts demonstrated different (changed) semantic networks compared to subjects who had not solved the problems (Durso et al., 1994). (The PsyArXiv paper you cite by Bieth, Kennett et al. could also be discussed more, as it’s highly relevant - but I’m not sure it’s published yet).

As noted above it may remain a question whether going from no-object (so, only activation for basic visual features) to object perception is the same as semantically restructuring a problem. But that’s just a conceptual that could be addressed (one could argue it’s perhaps a nomenclature issue); further, while that may explain the increased signal in object recognition areas, it might not as easily explain some of the other effects (other areas & connectivity results).

Additional specific / short (and probably garbled) notes:

It’s not entirely clear whether the authors are only analyzing “correct” trials when examining fMRI during high- vs low-insight recognition. Probably, but definitely should be, and they should be explicit (EVERY time say “on correct high-insight trials” etc). Still, those “correct” trials are contaminated by lucky guesses on the categorization response made in the scanner, and in a confounded manner.

1) What qualifies as “correct” in initial task in scanner is less than ideal: Correctly selecting the category (e.g., mammal or “objects in the house”) from 4 choices. So, participants start with 25% chance of choosing correct category. And, even without having full object recognition, they might be able to narrow down categories based on features – e.g., a picture with more straight lines and sharp angles is less likely to be a “mammal” than an “object”. So, “chance” might be better than 25%. And yet, as far as I can derive from presented data (%ages in different conditions), it appears that subject made “correct” response on only 38.5% of the trials in which they responded but trial was categorized as low-insight.

2) Because of the confound with accuracy, the “low-insight” trials, besides being lower in number, are more contaminated by lucky guesses on the categorization task. (If 25% were lucky guesses, only 13.5% were correct/real identifications – and it could be worse if we factor in strategic guesses). Thus, again, the signal for the already small # of low-insight trials might be

contaminated by a great deal of noise.

OTHER

Claims for insight boosting memory should be a bit more restrained; each of the cited studies has some limitations. Ludmer et al, e.g., didn't compare insight "solving" to non-insight memory, per se (if someone "solved" a stimulus, that trial was excluded from analyses); other older studies on memory (Auble et al, 1979) sometimes had a confound with study time/ time on task, etc etc

RESULTS

P 6:

Would like very clear upfront statements about # of trials in each condition being analyzed - presumably, e.g., 68.5% of 120, but please state, it helps with assessing stability of data measures. Also would be good to give a range for each of these measures. E.g., 68.5% (responded) x 51.8% (correct) x 34.7% (low-insight) x 120 = ~ 14.8 low-insight identifications. Some subs were below average, so you had very few trials from this category to analyze, let alone in the remembered vs non-remembered subdivisions of the category. Did you include subs who actually had very few correct low-insight trials in the scanner? (I know it's very hard to get large numbers, but just need to be transparent about it.)

More challenging: There are several important confounds.

The type of trial (i.e., the participants' response category) is highly confounded with time (2.75s for high-insight, 5.08s for low-insight). Authors try to handle this by including RT and accuracy [I still am not clear whether you're only analyzing CORRECT ID trials] as covariates of no interest in analyses, "if appropriate. I don't believe this is adequate for such an important confound. fMRI is hugely susceptible to time confounds (e.g, the shorter trials strongly conflate stimulus with response, moreso than the longer trials; and activity in a given region is integrated or summated over time- the longer you engage a process/cortical region, the stronger the BOLD signal). AND, the 'high-insight' trials are more likely to contain simple object recognition ("sudden" recognition immediately after presentation)

Did we ever get the #/% correct for high- vs low-insight responses? The way the averages are presented make the actual number of trials in each condition cell unclear. As is often found, accuracy was better when people report high-insight, so you have even bigger differences in # of trials per response type. More importantly, if subsequent memory measures (behavioral and back-sorted fMRI trials) start with ALL responses, then there are different %ages of correctly identified trials in the high- vs low- insight categories. This is still true if you include only trials that were "correct" in the scanner, but that category is (proportionately or in absolute terms) more contaminated by 'lucky guesses' in the low-insight condition.

RC: Fig 3a: of course you get bigger change in category with more correct responses

Are you "saved" by showing roughly equivalent LARGE change in some other areas? Maybe helps.

SUBSEQUENT MEMORY RESULTS

What I (and I think readers) want to know is:

How many trials (#, and % of all 120) were 'correctly' (with big caveat) recognized with high insight in scanner? Of these, how many were 'remembered' (by your criteria) 5 days later?

Then, same two things for trials recognized with low-insight.

Again, I think the starting point should be "of images correctly identified during fMRI..."

As is, I can't tell whether the #s stated (at bottom of p.6 into top of p. 7) are 68.7% of ALL trials elicited "correct" responses, then 74% of those were remembered. I want to know # of images ID'd first time that are then recognized (by all your criteria) 5 days later.

Trials solved via high vs low insight are remembered better

Again: are you starting with all responses, or just correct ones? For these analyses, you should only consider CORRECTLY identified trials. If not, several problems. One would be if you are including more Incorrect trials in one condition (probably the low-insight one).

METHODS

Subjects (not sure why sub-header here is "Experimental model and...")

Good info. I prefer to put the final N as close to the top as possible vs buried at end of paragraph. Could even say first sentence of "The final analyses included 31 participants" (age/gender)... "Initially, 38 participants were recruited...."

[note: typo "andtoo" needs space]

Similarly, I'd prefer an early and explicit note on the number of trials (on average) analyzed per participant. Total of 120 image trials, but avg x correct w/ high-insight and x correct low-insight, etc.; make explicit the actual # of trials in each

condition cell

Behavioral data acquisition

120 trials of Mooney images (30 in each of 2 runs x 2 days)

Pilot studies ensured solution accuracy of at least 30%, mean 56%

Assessing insight:

Probably best to reduce description and text emphasis on the multi-dimensional scoring of insight. You spend time/words convincing the audience it's important, only to later say you have to drop it. (I'm completely in agreement with you – it would be important, IF it could be done – but it would take MANY MANY trials). Trivial side issue: “originally” insight in self-report (but not neuroimaging) paradigms was assessed on a scale of 1-5 (Bowden & Beeman, 2003). But, it becomes difficult to compare across participants that use scales differently, especially with few responses, so a binary form was adopted for most later neuroimaging and related studies. However, typical neuroimaging paradigms – and certainly here – elicit fewer correct responses. So, attempting to use complex scales is difficult. Hence, here they ultimately binarized the response anyway. So, perhaps, instead of emphasizing and justifying the multiple dimensions (each with 4 point scale), might reverse and say you ultimately used binary categorization, despite attempting to use complex scale. (The attempt to use the multiple scales does seem to be the right approach – IF there are enough usable trials and participants to make use of such data, but there aren't enough solutions).

Mooney image paradigm

Good description.

But, need to ultimately give # of trials and the distribution of response times within the conditions (correct hi- and correct low-insight).

Subsequent memory test

Again, need a clear statement of the number of trials being analyzed. E.g., on average subs had [68?] correct identifications of Mooney images during fMRI, so we're only looking at those. Of those, how many (#) were correctly recognized? And (above) problem with lucky guess “correct” trials during initial viewing in scanner.

1st paragraph:

Minor writing: “2) the Mooney image was correctly recognized AS having been....”

2nd paragraph: need to explicitly state that remembered/forgotten trials were those “initially identified, but then subsequently not or not correctly remembered”. OR, were you assessing memory for all trials (including not identified – that would seem to violate the criteria laid out in 1st paragraph of this section).

SUPPLEMENT / METHODS

INFO on Nature HCC form (some issues already noted above)

Anatomical: are you saying individual subject anatomical-based ROIs for amygdala and hippocampus (very good, if so)

Unclear: ...“ROIs masks matched ...” – is there just an extra s after ROI?

“Correction: Where it applies, we USED FDR or Bonferroni corrected ALPHA LEVELS” (or just correction)

Minor writing issues and typos

There are a number of typos or minor grammatical errors, too many to list (but for the most part I do not think these affected my comprehension of their text).

In at least one place (Fig caption) they state that memory test was 7 days after initial fMRI session, most others state 5 days.

Writing issue:

Representational change terminology. I'm OK with using this term as long as there's at least a brief note of how it may differ in this paradigm versus, say, classic insight problems (or even simple word puzzles). But then several different versions are used (“conceptual change” Etc) often followed by the RC acronym, which is just a bit confusing, especially to people less familiar with the area. Even if it gets repetitious, consistency would be preferred.

Figure 3: Label the cortical areas in Figure (could just note the colors per area in the caption, but in-figure labels would be most helpful)

References, other relevant work

Reference for earlier effort to find behavioral evidence of restructuring involved in insight, using graph-theoretic evaluation of semantic relatedness judgments:

Durso, Rea, & Dayton (1994). Graph-theoretic confirmation of restructuring during insight. *Psychological Science*, 5, 94-98.

Sampling of References regarding object recognition /visual closure / visual Aha! that may prove helpful. I strongly recommend citing & discussing the first (Bar 2001), but the others are also potentially related to interpretation (for the most part compatible with current interpretation, but viewed from a different angle, so to speak).

*Bar, M. et al. (2001). Cortical mechanisms specific to explicit object recognition. *Neuron*, 29, 529-535.

Bar, M. et al., (2006). Top-down facilitation of visual recognition. *PNAS*, 103, 449-454

Doniger et al., (2000). Activation timecourse of ventral visual stream object recognition areas: High density electrical mapping of perceptual closure processes. *J. Cog Neuroscience*, 12, 615-621.

Doniger et al. (2021). Visual perceptual learning in human object recognition areas: A repetition priming study using high-density electrical mapping. *NeuroImage*, 13, 305-313.

Sehatpour, P. et al., Spatiotemporal dynamics of human object recognition processing: An integrated high-density electrical mapping and functional imaging study of "closure" processes. *NeuroImage*, 29, 605-618

Reference related to continuous vs binary insight ratings (does not need to be cited – I never ‘require’ anyone to cite any of my papers – but it’s relevant in that we, too, started by trying to use the continuous nature of strength of “aha” ratings, but found it too difficult to work with statistically so switched to binary (insight/non or analytic/insight, etc; sometimes offering “other” as an option, but that also creates complications) in later studies.

Bowden, E.M. & Jung-Beeman, M. (2003). Aha! Insight experience correlates with solution activation in the right hemisphere. *Psychonomic Bulletin & Review*, 10, 730-737. PMID: 14620371

Reviewer #3

(Remarks to the Author)

The paper reports on an exciting study investigating the synergy between creativity and memory, with a particular focus on insight. The authors report on “sudden representational changes” in visual regions, insight-related activity in MTL regions, and the finding that these effects are predictive of later memory for the solutions. Most studies examining creativity directly struggle with the fact that insight is unpredictable and infrequent, rendering it difficult to examine with neuroimaging. However, here the authors use a task that more reliably provides an experience of insight, enabling them to examine the neural processes underlying the phenomenon underlying insight, albeit outside of the typical creative context.

Overall the manuscript is well written, the analyses highly novel (especially the analyses on RC and representational strength), and the results for insight particularly compelling (less so for memory).

(1) General comment: It should be noted somewhere that while insight is most certainly a phenomenon associated with a rapid shift, in the context of creative problem solving the process leading up to this shift might not be as discrete and sudden. For example, if one has thought about the problem for days or months, likely with slower changes in semantic networks, etc., insight might not necessarily reflect “rapid learning and memory formation”.

(2) Behavioral results: It is notable that almost half of the incorrectly identified images were rated as high insight - does this present a concern with using insight ratings as the main variable of interest in this study? Selecting the correct category is not a precise measure of accuracy; are analyses chance-adjusted? This also means that, at test, an incorrect yet category-consistent answer could reflect either a memory failure OR not having generated the correct solution to begin with. Is there any data that can allay this concern?

(3) fMRI analyses – overall comments: (a) A number of analyses compare high vs low insight trials; showing that the effects are not at all evident for the no-solution trials would be even more compelling. e.g., showing that RC is not different from zero for no-solution trials would demonstrate that the shift in representation is specific to trials on which a solution was generated (and then modulated by degree of insight). (b) Analyses using a categorical insight variable, even though the continuous variable is described as more sensitive, with some memory effects only evident using this. Should it be used throughout?

(4) Evaluative aspect of insight: The 3 ratings are summed to form an “insight” variable. However, it would be useful to know what was driving the insight effect. For instance, in the discussion the authors suggest that effects in the amygdala reflected emotion, and in anterior hippocampus both novelty and encoding. The data exists to answer all of these questions, including to determine whether anterior hippocampal effects on subsequent memory hold when controlling for the effect of novelty (e.g., BOLD signal could be residualized for the effect of novelty prior to running subsequent memory analyses, or using a parametric modulation approach). Relatedly, the authors state that “Using covariates to remove unwanted variance is not feasible in the SPM GLM framework” (p.12-13); however this can be achieved using parametric modulation analysis with

insight as the first parameter (= variable of interest) and response time and accuracy as subsequent parameters (= nuisance covariates).

(5) Functional connectivity (FC) analyses: Some of the analyses seemed redundant. The authors start with all ROIs, then re-run on those ROIs with strongest effects, and end up with the subset of regions identified in other analyses (i.e., pFusG, iLOC, amygdala & anterior HC). This also meant it's not always clear what analyses/results reflect the full network or the subnetwork (e.g., graph theory analyses; plot Fig 5B). The FC analyses are described as defining a "solution network" but this isn't accurate given the full 10 second trial is used (not just the time of insight/solution). For this reason, it is also not surprising that the FC effects are not predictive of memory – this should be mentioned in the discussion.

(6) Even though significant, the memory effects are not all that striking. There are effects of insight evident for solutions that were later forgotten, and relative to that, there really is only a small increase for those that were remembered. This should at least be mentioned in the discussion.

Minor comments

Please mention the final N of the sample in the introduction or beginning of results.

Fig 4 and Fig 6: It is extremely difficult to see the different ROIs; using color might help.

Fig 5: define "NW"

VOTC ROIs are always described as bilateral. Is the same true for MTL ROIs?

p. 4: Unlike positive emotion and novelty/suddenness, there's no explanation in the introduction as to why certainty ratings are included as a measure of insight measure

p. 5: "Third, solution-relevant information is more efficiently integrated in a "solution network" consisting of VOTC, amygdala, and hippocampus during insight". This is poorly worded as the information isn't integrated into the network, but by the network

p. 35: If German as first language was an eligibility criteria, was the study conducted in German?

p. 35: "andtoo"

Version 1:

Reviewer comments:

Reviewer #1

(Remarks to the Author)

The authors were responsive to my previous comments, and I have nothing further to add. I would like to congratulate the authors on a significant contribution to the cognitive neuroscience of creativity. I hope to see more clever and rigorous work like this in the future.

(Remarks on code availability)

Reviewer #3

(Remarks to the Author)

The reviewers have extensively revised their manuscript and have addressed all of my comments to a satisfactory standard. I have no further comments and commend the authors on an exciting contribution to the literature.

(Remarks on code availability)

Reviewer #4

(Remarks to the Author)

This study is interesting and well-conceived, and is potentially an important contribution to the neuroscience of insight. However, it has a number of issues that need to be addressed. The following are in no particular order. (For the record, I am not an fMRI specialist.)

The high error rate in the problem-solving part of the study is a concern. (The paper should described how the data were "chance adjusted"?) This, together with the fact that subjects only had to identify the categories of the objects rather than the objects' identities is also of concern.

Instead of the usual insight judgment procedure in which a participant judges a solution as having been derived by insight, analysis, or "don't know/other", the authors tasked their participants with judging the amount of positive emotion, suddenness, and certainty. Unfortunately, no opportunity was given to subjects to respond "don't know/other" which could have led to some guess or random responding. Subsequently, they found that these 3 measures were correlated, so they summed them into a single insight measure anyway, suggesting that the three dimensions are parts of a single mechanism. (Is the distribution of the unitary insight variable bi-modal, which would suggest that the median split is not the right way to divide trials between high and low insight?) And yet, they subsequently break these measures apart to determine whether

different brain regions underlie these dimensions. The logic of this procedure is murky.

On a related note, insights are often described in the literature as conferring a feeling of confidence. However, this is probably not true for all tasks. For example, for arithmetic tasks such as $(147 \times 7,892)$, it is likely that a person would feel more confident in an analytically derived solution rather than a solution derived by insight. Are there any data to support the idea that perceiving Mooney objects with a unitary Aha experience leads to more confidence?

The methods section does not describe the task procedure in sufficient detail. The instructions given to subjects are not described. One concern is how the 4-point scale is described. Is "1" considered zero emotion, suddenness, or certainty? The concern is how the scale maps on to the unitary Aha experience. Does 1-4 refer to 0-100% Aha or 90-100% Aha? For example, does "1" refer to analytic judgements or insights of moderate strength? Do all subjects interpret the scale in the same way? And if the scale just refers to levels of insight and excludes pure analytic judgments, this procedure excludes the possibility of directly contrasting the end points of the scale. For example, what if the whole distribution is non-monotonic?

Another concern is that the authors pick ROIs that mostly make sense, but shouldn't they compare these ROIs to other areas that, from known theory, should not be expected to play a role? And what if there are stronger results in areas other than the amygdala, hippocampus, and VOTC? Not knowing the answer to this question leaves open the possibility that the real action is elsewhere. Furthermore, Oh et al. (2020) showed a putative reward system response associated with insight in orbitofrontal cortex. Why not include OFC rather than, or in addition to, the amygdala? What about the ACC which has been implicated in past insight studies?

Pre and post solution intervals were divided according to the time of the button press. However, we know from the Jung-Beeman et al. (2004) and Oh et al. (2020) EEG studies that an insight solution occurs 300-500 ms before such a button press because, once a solution has been derived, it takes time to program and execute the button press. This, together with the limited temporal resolution of fMRI makes the division between pre- and post-solution potentially blurry.

The low n per subject per condition is concerning. Judging by Figure S5, for some subjects, the n per cell was extremely low, even 0. I found the authors' response to the other reviewer insufficient. I would want a more detailed statistical justification for retaining subjects whose n per cell is low or zero.

Fig. 2D includes an obvious outlier that seems to boost the correlation coefficient. What is the correlation coefficient without the outlier?

Overall, this study is on the right track. If these points can be adequately addressed, then I would support publication.

I've read through Reviewer #2's feedback and the authors' responses. I will address these points according to the numbering in the authors' responses.

1. The low trial counts in particular conditions are a significant concern. They state, "that low trial counts would be more likely to increase type II error rather than type I error, leading to an absence of a consistent memory effect in the connectivity/graph analyses." The phrase "more likely" is not reassuring. I would want something more convincing, such as simulations to show the likely effects. I would also want a statistician to review this issue.
2. Regarding the effects of having subjects classify, rather than identify, the objects in the Mooney images, this was not an optimal choice for a procedure. The authors' control experiment gives some reassurance. However, it does not fully reassure the reader in light of this non-optimal procedure choice.
3. Regarding potential time-on-task effects on the BOLD signal, the authors' fixes help, but may not be sufficient. A low RT cutoff of 1.5 seconds seems somewhat arbitrary and may not be enough. The authors say that "we removed fast high-insight trials and slow low-insight trials until the time taken to solve both high and low insight trials was no longer significantly different ($p > 0.20$)." Here, statistical significance is uninformative. Just because the RT difference between HI and LI is now nonsignificant doesn't mean that there isn't enough of a difference to cause a BOLD time-on-task confound. Simulations could be helpful in clarifying this issue.
4. Rev #2 makes good points about restructuring. In particular, on some trials subjects may be, to coin a term, "structuring" rather than restructuring if they are going from no structure to some structure. Thus, the findings might show a mixture of structuring and restructuring, which would still be fine because it would not negate the authors' general point. I'm OK with the authors' handling of this issue.

The rest of Rev #2's points mostly focus on the number of datapoints per condition. Again, this is a concern, and I would want to see simulations evaluated by a statistician to determine the level of concern.

Overall, considering Reviewer #2's points and my own review, my feeling is that this is an ambitious study that addresses an important point. However, there are many issues, some small and some not so small, that individually may not be deal-killers, but that together give this study a "death of a thousand cuts" feel. In an ideal world, it would be great if the authors could refine the design and procedure and rerun the study with a large enough sample size that they could drop subjects who do not provide enough datapoints per condition, etc. But I recognize that we do not live in a research environment with unlimited resources and time. Therefore, the options are to reject the paper and hope for better in the future or to publish a version of this paper that highlights the necessary caveats. My feeling is that it would be better to publish a version (that addresses my separate review) with caveats as a way to stimulate additional research along these lines.

(Remarks on code availability)

Version 2:

Reviewer comments:

Reviewer #4

(Remarks to the Author)

I have read the authors' replies and revisions in response to the reviewers' comments. I think that the authors have done a fine and meticulous job in addressing the concerns. I am now happy to support publication of this impressive paper.

(Remarks on code availability)

Reviewer #5

(Remarks to the Author)

In their article titled "Neural Mechanisms of Creative Problem Solving – From Representational Change to Memory Formation", Becker et al investigate the knowledge reorganization and integration (i.e., representational change) related to the experience of insight. They used a task in which participants, while undergoing fMRI scanning, saw ambiguous images and had to recognize the image category (among buildings, animals, tools, and plants). Then, they rate 3 aspects of their insight experience (suddenness, emotion and certainty) and indicate which category the image belongs to. Five days later, the participants completed an online task in which they had to indicate for each ambiguous image whether they had seen it before, if they solved it and if they remembered the category.

The results are organized into 5 large sections: first, a behavioral section and then 4 neuroimaging sections, in which they address four different hypotheses. First, they show that in the behavior, trials with insights are related to better performance (with a parametric effect of the strength of insight). Then, (H1), they show that insight is related to a change in representation (in the VOTC and in the conceptual space). For H2, they found activity in the hippocampus and amygdala that was related to insight. For H3, they report a stronger connectivity between VOTC and amygdala/hippocampus during insight. Finally (H4), they show that representational changes were associated with better memory and that activity in the hippocampus during insight was also related to better memory. In the four neuroimaging sections, they successively used RSA (Representational Similarity Analysis), univariate analysis of activity pre-defined regions of interest (and whole-brain in supplementary), graph-theory analyses and mixed-models on the variables identified in H1 and H2.

The results are overall very hard to follow, but they can nevertheless be summarized with a graphical representation (see the attached document).

Overall, this study brings novel and interesting results, that are of interest for a wide community. The methods seem appropriate, including bootstrapping and permutation testing. However, the conducted analyses are particularly difficult to understand, and yet, I am familiar with all of them. To understand the results, I had to carefully read the methods, which were not truly insightful (if I may) and led me to actually have a look at the open scripts used to obtain the results. I can confirm that the scripts are very clear and well organized, and the results obtained with the scripts match the ones reported in the manuscript. In my opinion, a reader should not have to go into deep details of the methods and scripts to actually grasp the results, especially in a journal such as Nature Communication. One main problem in the report of the results is that the authors do not explain which analyses they conducted and report results with confusion between model comparison results (all the χ^2 statistics) and the value of the fixed effect coefficients (odds-ratio). Moreover, some phrasings do not exactly match the results, and some conceptual aspects of the scientific questions are quite vague. I have detailed below each of these aspects that seriously lower the overall quality of the manuscript.

1. Conceptual aspects

- There are not a lot of studies addressing representational changes during insight but the authors should at least the ones that address this question, such as Bieth et al 2024 (<https://www.nature.com/articles/s44271-024-00100-w>).

- In the introduction, the authors state that "Amygdala activation may reflect the positive emotions (Janak & Tye,2015), and the hippocampus activation, the sudden arrival and/or surprising content of the solution.". In my opinion, this is reverse inference and that should be avoided.

- In the introduction, the authors state their hypotheses and mention a "solution process", I don't see to which process they are referring to, which make their hypotheses vague when we read them and try to make sense of them.

- When the fourth hypothesis is announced, cognitive and evaluative components are mentioned but it is difficult to immediately match cognitive with RC and evaluative with hippocampus (maybe the authors could phrase it differently, or at least add between parentheses "cognitive" and "evaluative" after RC and and hippocampal activity respectively.

2. Results

The whole analysis logic should be better reported in the entire results section. I provide the detail of the very first behavioral result to illustrate how hard it was to understand the very simple analysis they conducted. My argument is valid for the vast

majority of their results. I must insist on the fact that it's only by reading their script that I understood what they did.

- We can read "Consistent with previous results, this insight-accuracy effect was significant ($\chi^2(1)=94.98$, $p<.001$, odds ratio=1.31)". At this point, we have absolutely no clue about which analysis is conducted here. In the methods, there is nothing about that "insight-accuracy" effect. Only in the script we can read:

```
#####correlation between accuracy and insight#####
```

```
data_new = PPSS[ PPSS$ROI == 'l_Amygdala' & PPSS$tp == 1 & (PPSS$RT>=1.5 & PPSS$RT<=9.5),]
```

```
M0_cor2 <- glmer(cor2 ~ +sessionblock + (1|ID) + (1|Item),data= data_new,family = binomial(link = "logit"), na.action = na.omit)
```

```
M1_cor2 <- glmer(cor2 ~ insight_sum +sessionblock + (1|ID) + (1|Item),data= data_new,family = binomial(link = "logit"), na.action = na.omit)
```

```
anova(M0_cor2, M1_cor2)
```

```
tab_model(M1_cor2, show.std = T)
```

With that, we finally understand that what they did is a model comparison, with one model regressing accuracy against either only the session block or the session block and the insight measure. Then we understand that the chi2 statistics is the result of the model comparison, suggesting that a model with the insight measure better captures accuracy than a null model. The odds ratio corresponds to the odds ratio of the regression estimate of the insight measure in the winning model. I think it is crucial that the authors clearly explain the statistics they report.

- In the section "Insight predicts better subsequent memory", the authors indeed explain that they conducted two linear mixed models: "To investigate behavioral insight-related better memory, we estimated the effect of insight on subsequent memory using two general linear mixed (binomial) models including random subject and item intercepts and the respective run order (1-4). The first model predicted subsequent memory including a categorical insight factor consisting of not solved trials, correctly solved HI-I and LO-I trials."

They explain the first model, not the second, and then jump to the result of the insight factor. The methods state:

I. Subsequent memory \sim run + (1 | ID) + (1 | item) + ϵ

II. Subsequent memory \sim run + insight + (1 | ID) + (1 | item) + ϵ

Here, we understand that the first model is a control model, and that the second model includes the insight factor. We understand that the model comparison demonstrated that the second model was better than the first one such that the insight factor is needed to better explain the memory effect. This model comparison result is reported as " $\chi^2(2)=666.49$, $p<.001$ ". Then, odds ratios are reported without any indication of which analysis they correspond to. By looking at the script, I understand that it corresponds to the contrast of the regression estimate of HI-I versus unsolved and LO-I versus unsolved.

- "the model with the three-way insight*time*ROI interaction did not explain significant variance in representational strength" : please update that sentence as it makes no sense when we don't know which analysis is conducted. Here, I assume that the authors wanted to say that a model with the three-way interaction did not explain RC better than a model without it.

- Hypothesis 2: "there was a main effect for condition" > but what is condition? From the script I understand it's insight.

- "Posthoc analyses revealed that activity in amygdala activity ..." > please double check the text.

- "hippocampus significantly predicted insight" > If the dependent variable is activity, then it's the insight that "predicts" activity, not the reverse. Or the analysis should be done the other way around. In any case the use of the term "predict" suggest too much causality, here, it's a relationship that has been found.

- I have small doubts about the degrees of freedom, as the ones in the script seem to not match the manuscript in some cases. To be double-checked.

(Remarks on code availability)

The results are reproducible and the code is usable. I would just double-checked the degrees of freedom reported in the manuscript as they don't match the ones in the script. I'm not sure why.

Version 3:

Reviewer comments:

Reviewer #5

(Remarks to the Author)

The authors have carefully and appropriately addressed my comments. I believe the manuscript's clarity has been significantly improved and I have no further comments.

(Remarks on code availability)

I checked it for the previous version but not the latest.

GENERAL RESPONSE TO ALL REVIEWERS

We express our sincere gratitude to all three reviewers for their active involvement in improving this manuscript. All responses to reviewer feedback are typed in green, and corresponding changes in the manuscript are marked in yellow. The major changes made to the manuscript are summarised below:

1. Insight as a Continuous Measure:

- Reviewer #3 suggested to use a continuous instead of the binarized insight measure as the continuous one is more sensitive. In response to this reviewer and due to a limited number of forgotten high insight trials, we reanalyzed all insight-related analyses using the continuous insight measure. For methodological reasons, this modification was not applicable to connectivity and graph analyses.
- In response to reviewer #2, all analyses were performed excluding trials solved incorrectly and in less than 1.5 and more than 9.5 seconds. This aimed to eliminate trials confounded with stimulus onset or potentially reflecting pure object recognition trials, and stimulus offset.
- Notably, there were no significant changes in the results, except for insight-related memory outcomes using AlexNet, which are now discussed in the discussion section.

2. Differences in solution time between high and low insight trials might lead to a confound in the BOLD signal due to time-on-task:

- Reviewer #2 expressed a concern about the potential time-on-task confound in the BOLD signal, suggesting that the variations in solution time between high and low insight trials might lead to differences in the magnitude of the BOLD signal.
- We addressed this potential time-on-task confound differently depending on the respective analysis:
 - i. Univariate activity in amygdala & hippocampus: For the single-trial-analyses (standard space) investigating univariate activity as a function of the continuous insight measure, we adjusted for solution time (by removing fast high and slow low insight trials until the time to solve those trials did not differ significantly anymore [$p > .2$]) and the results did not change significantly. For univariate analyses in subject space, we controlled for solution time in SPM by incorporating it as an additional parametric regressor modulating the *solution-button* onset regressor in addition to the (orthogonal) continuous insight regressor. The results did not change significantly.
 - ii. Multivariate RSA analyses in VOTC-RC areas (pFusG, iLOC): We included the size of the beta values during solution as a covariate of no interest into all single trial analyses investigating changes in multivariate pattern similarity from pre to post solution as a function insight (continuous). The results did not change significantly.

3. Differences in solution time between high and low insight trials might lead to a confound due to differences in encoding time:

- Reviewer #2 also raised a concern regarding higher insight trials, suggesting that participants identify the Mooney object more rapidly, resulting in additional time for encoding the object compared to lower insight trials. To address this concern, we conducted a pre-registered (behavioral) control experiment which was mostly identical to the Mooney identification and subsequent memory task in the fMRI sample. However, in this control experiment, the Mooney image disappeared immediately after subjects pressed the solution button. Despite participants having less time to encode the solution in trials solved with higher insight (due to faster solution time), the experiment still demonstrated a consistent advantage in insight-related memory.

4. Adjustment for Accuracy Measure:

- Reviewer #2 and #3 noted that our accuracy measure was imprecise because subjects did not name their solution object directly after having found the solution but selected one out of four possible response categories leading to a certain percentage of false positives by chance. To estimate the actual amount of false positives, we explored whether an incorrect solution in the control experiment would have been deemed correct had participants been presented with the same four solution categories per Mooney image as the fMRI sample. On average, 18.7% ($SD=10\%$) of correctly classified trials would have been inaccurately labelled as correct, including solutions not fitting any of the presented categories, resulting in a 1/4 chance of misclassification. Despite the adjustment, subjects still performed above chance in the subsequent memory test.

5. Limitations Section:

- Although we have addressed all of the reviewers' points, we have nevertheless included a Limitation section in the discussion, as suggested by Reviewer #2. Here, we briefly discuss the accuracy measure, solution time as a potential confound, and the relatively low number of trials in the forgotten memory condition.

6. Exploration of Different Dimensions of Insight:

- In response to reviewer #2, we explored the individual effects of the three different dimensions of insight, i.e. suddenness, certainty and positive emotion, on univariate activity during solution and subsequent memory in the amygdala and hippocampus (evaluative component of insight). The results of this exploratory analysis suggest that certainty and, to a lesser extent, positive emotion drive the insight experience during solution, while suddenness drives insight-related better memory in the anterior hippocampus. Please note however, these dimensions of insight only have 4 values and are therefore less sensitive than the continuous insight sum measure (min. 3 - max. 12).

7. Relation to Other Insight Tasks:

- Responding to reviewer #1, we correlated the insight experience between Mooney images and anagrams, demonstrating the comparability of the insight experience between these different types of tasks.

Once again, we appreciate the invaluable contributions of the reviewers, which have significantly strengthened the quality of our manuscript.

REVIEWER COMMENTS

Response to Reviewer #1 :

General Comments

This study investigates the neural mechanisms underlying insight problem solving and its impact on memory. The authors employ multivariate pattern analyses to examine representational change in visual areas, going beyond traditional univariate approaches. Moreover, they use Representational Similarity Analysis (RSA) to track pattern similarity changes, as well as model-based RSA with AlexNet and Word2Vec to assess representational shifts associated with conceptual reorganization.

The results support key hypotheses about representational change (RC) in the visual cortex during insight and its connection to memory encoding. Notably, evidence points to robust RC in specific ventral visual stream regions during high insight solutions, with changes in patterns and enhanced conceptual representation confirmed by model-based RSA results. Insight was also found to modulate amygdala and hippocampal activity, with effects holding when controlling for potential confounding factors. Furthermore, functional connectivity and graph theory findings indicate that visual, hippocampal, and amygdala regions form an interconnected network during insight.

Overall, the experimental design and multivariate fMRI analyses provide an excellent basis to elucidate the neural mechanisms underlying insight and its impact on memory formation. A key strength of this work is testing of specific neural hypotheses. I commend the authors for approaching their analyses in such a systematic and rigorous way.

I found very little to criticize about the methods and experimental design. In my view, this is among the most rigorously conducted and innovative cognitive neuroscience studies of insight in the field. The paper is very well written.

My main (minor) question concerns the interpretation of the results, and whether the process studied should be called “creative”, or rather a basic perceptual restructuring. Further details are below.

Specific Comments

1) “The neural mechanisms driving creative problem-solving still remain largely unknown”; creative problem solving has been studied quite a bit, including by the co-authors. Is this really the case?

We agree with the reviewer that previous research has indeed successfully identified the neural correlates of insight, i.e. the brain regions implicated in creative problem-solving, and that our wording was therefore misleading. The goal of the current study was to advance our understanding of the specific mechanistic contribution of these brain regions to the insight phenomenon and insight driven memory. With the term 'neural mechanisms,' we intended to specifically refer to the functions of these brain regions, i.e. the realisation of representational change and how these processes are related to memory encoding as we mentioned in the manuscript:

“The main cognitive component is associated with *representational change* (RC) [...] This process is assumed to involve a change in the pattern of currently activated cell assemblies coding the problem in domain-specific cortex but neuroscientific evidence is currently lacking (Becker et al., 2024; Knoblich et al., 1999). [...]

Finally, the neural mechanisms of this *insight-related better memory* remain largely unknown (Kizilirmak et al., 2019).” (p.2f)

We believe that these mechanistic aspects are still underexplored despite the rich literature on the neural correlates of insight. In response to the reviewer's comment, we have revised and clarified this aspect in the abstract of the manuscript as follows:

“Despite the need for innovative solutions to contemporary challenges, the neural mechanisms driving creative problem-solving, **including representational change and its relation to memory**, still remain largely unknown.” (p.1)

2) The mention of “sudden” representation changes in the visual cortex raises a query considering the study employs fMRI, which inherently lacks the temporal resolution to precisely capture "suddenness"; perhaps an alternative descriptor could be more accurate.

Representational changes are commonly described as 'sudden' in the literature (Jones, 2003; Öllinger et al., 2008; Cushen & Wiley, 2012; Danek et al., 2020). However, we acknowledge the reviewer's point that the term “sudden” in this particular context may not be adequate; first, because the fMRI method employed in this study exhibits a low temporal resolution due to the sluggish BOLD response. Second, our analyses focus rather on the magnitude of the representational change using multivariate methods than its suddenness per se (given our control for solution time). For this reason and in accordance with the reviewer's suggestion it is more accurate to rename 'sudden' representational change as 'stronger'/'increased'/'enhanced' representational change. We have accordingly adjusted this terminology in the manuscript as follows:

“In sum, insight during problem solving likely reflects **stronger** multivariate pattern changes associated with enhanced solution-relevant representations. This **increased RC (or conceptual update)** is efficiently integrated into the solution process [...]

Next, we examined the assumption that the **stronger** conceptual update (RC) is efficiently integrated into the solution process. [...]

Hence, in the context of insight, the hippocampus may detect novel, unexpected internal events (e.g., **enhanced** conceptual **RC** of solution object in VOTC) by eliciting a prediction error signal (surprise) and subsequently encodes these events [...]

We presented consistent neural evidence indicating that insight in problem solving likely reflects **an enhanced** conceptual **RC** of solution representations efficiently integrated into a coherent whole.” (p.3ff)

Cushen, P. J., & Wiley, J. (2012). Cues to solution, restructuring patterns, and reports of insight in creative problem solving. *Consciousness and Cognition*, 21(3), 1166-1175.

Danek, A. H., Williams, J., & Wiley, J. (2020). Closing the gap: connecting sudden representational change to the subjective Aha! experience in insightful problem solving. *Psychological research*, 84, 111-119.

Jones, G. (2003). Testing two cognitive theories of insight. *Journal of Experimental Psychology: Learning, Memory, and Cognition*, 29(5), 1017.

Öllinger, M., Jones, G., & Knoblich, G. (2008). Investigating the effect of mental set on insight problem solving. *Experimental psychology*, 55(4), 269-282.

3) I was a little confused by the distinction between “cognitive and evaluative” components. Doesn’t evaluation involve cognition to some extent?

Yes, we agree with the reviewer, the evaluative component includes cognitive aspects. It is crucial to note that our description of the two insight components draws from Danek & Wiley (2017), who categorized insight into a cognitive aspect involving representational change and an affective aspect associated with the subjective AHA! experience. However, we opted to replace the term "affective component" with "evaluative component" because the AHA! experience itself, encompassing the perception that the solution appeared suddenly and is certainly correct, extends beyond mere affective elements. It is more accurately characterized as an evaluative component, considering questions such as how the problem was solved (suddenly) and the certainty of the solution's correctness (very sure). Both components can also be distinguished chronologically, as the evaluative component assesses what was previously solved cognitively. We had defined both terms in the manuscript before but apparently did not describe them well enough to make this distinction sufficiently clear, for which we apologise. The revised definition of both components in the manuscript as follows:

“The insight process has cognitive and evaluative components. The main cognitive component is associated with *representational change* (RC) whereby internal (conceptual or perceptual) representations of the solution are reorganised and integrated into a coherent whole (Ohlsson, 1984; Schilling, 2005). [...]

The evaluative component (or *AHA! experience*) is characterized by positive emotions and the conviction that the solution **arrived suddenly and is certainly correct** (Danek & Wiley, 2017; Kizilirmak & Becker, 2022). **Note, this component involves cognitive aspects as it assesses the solution and** has been associated with activations in the amygdala and the hippocampus [...]

Importantly, at the time of insight, cognitive and evaluative elements are unlikely to function in isolation but rather in a closely interconnected manner, facilitating the efficient processing of solution-relevant information (Mednick, 1962; Schilling, 2005).” (p.2f)

4) [note this was originally point 5] One limitation in my view is the restriction to Mooney images and the omission of a verbal task, which is often a more prevalent measure of insight.

We thank the reviewer for this suggestion. In fact, we had acquired data from the same participants performing a more classical verbal insight task: anagrams. The correlation (subject-level) between both tasks was significant. Those results were described as follows in the manuscript (note, the anagrams were not incorporated into the present manuscript due to the absence of meaningful subsequent memory measures and the absence of hypotheses regarding regions that might exhibit representational changes, unlike the visual Mooney paradigm).

Methods section:

“For a different research question, we also presented 120 anagrams in an interleaved fashion with the Mooney images. Those anagrams consisted of six to seven letter frequent words of concrete objects in German (e.g. “BAGGER”). The word’s letters were scrambled and for each anagram, participants had to rearrange a set of letters to form a meaningful word (e.g., “RGEAGB” can make “BAGGER”). Mean accuracy was 61.4% (SD=17.8%).”
(p.28)

Results section:

“Association with verbal insight tasks

We further investigated whether solving Mooney images reflects a similar underlying creative problem-solving process (see (Kenett et al., 2015) as the more commonly used verbal insight tasks. To assess this relationship, we computed a Pearson correlation between the participants’ average continuous insight experiences in Mooney images and an anagram task (see Fig.2-D). The obtained correlation was statistically significant ($r=0.68$, $CI[0.43, 0.83]$, $t(29)=5.013$, $p<.001$), indicating that the insight experience evoked by Mooney images is comparable to that of another verbal insight task.

Figure 2. Behavioral Insight Memory Effect.

Note. **Panel A.** Overall proportion of all trials per condition; **Panel B.** Behavioural insight memory effect; **HI-I= correctly solved high insight trials; LO-I= correctly solved low insight trials;** estimates are marginal mean percentages to correctly recall the previously identified Mooney image in the respective condition. Error bars represent 95% confidence intervals. **Panel C.** Relationship between continuous insight (3-12) and memory controlled for solution time; estimates are marginal mean percentages to correctly recall the previously identified Mooney image. Error bars represent 95% confidence intervals. **Panel D.** Correlation (subject level) between anagrams and Mooney images. (p.8f)

5) [note this was originally point 4] The task paradigm was well reasoned and elegant for invoking insight. But I wonder if the Mooney image task has been validated as a measure of *creative* problem solving? For example, do people who can more quickly/accurately identify the Mooney image tend to perform better on other creativity tasks? My question here is whether this task really measures an aspect of creativity (or even problem solving), or rather some basic perceptual/restructuring process. Are we really talking about high-level creative problem solving if everything happens at a perceptual level? I'm not entirely sure that this task measures 1) problem solving and 2) creativity. That doesn't say anything about the quality of the work (which is excellent) – it just speaks to how we interpret the results. I don't doubt that insight was measured here. And I recognize that definitions of creativity vary.

We have reasonable grounds to believe that Mooney images capture some aspect of general creativity. First, Kennett and colleagues (2015) explored the relationship between Mooney images, particularly unusual faces in the Mooney style, and creativity. Their findings revealed a positive correlation between the Mooney recognition task and verbal quality and

fluency scores on the Tel-Aviv University Creativity Test (TACT) by Wallach and Kogan (1965). We cited this paper in the manuscript now as follows:

“We further investigated whether solving Mooney images reflects a similar underlying creative problem-solving process (Kenett et al., 2015) as the more commonly used verbal insight tasks.” (p.8)

Moreover, as described in detail in response to #4), the insight experience of the Mooney images significantly correlates ($r=.68$) with the insight experience of anagrams - a verbal task that has been used to study creative problem solving before (Eisenstadt, 1966; Shaw & Conway; 1990; Aziz-Zadeh et al., 2009; Sinitsyn et al, 2020).

Nevertheless, we share the reviewer's perspective that tackling Mooney images likely demands less working memory compared to solving more complex tasks more classically associated with high-level creative problem solving such as magic tricks or the nine-dot task. Nevertheless, as detailed in the manuscript's introduction, it is crucial to note that insightful problem solving is a special case of creative cognition and the fundamental elements defining insight—namely, representational change (cognitive component) and the AHA experience (evaluative component), as per current insight research—are also discernible in the context of Mooney images. However, we understand the reviewers' concern. Therefore, to emphasize further, we have added another sentence underscoring the necessity to assess the generalizability to other, more high-level creative problem-solving tasks.:

“Taken together, these findings represent the first direct evidence of an integrated neural mechanism for insight and its impact on memory. Its generalizability to other, more complex creative problem-solving tasks requires further assessment. The four findings are discussed in separate sections below. [...] Future research could examine if insights tasks that require more abstract semantic processing (e.g., regrouping conceptual features in verbal insight tasks) yields RC in anterior VOTC regions.” (p.21)

Aziz-Zadeh, L., Kaplan, J. T., & Iacoboni, M. (2009). “Aha!”: The neural correlates of verbal insight solutions. *Human brain mapping*, 30(3), 908-916.

Eisenstadt, J. M. (1966). Problem-solving ability of creative and non-creative college students. *Journal of consulting psychology*, 30(1), 81.

Kenett, Y. N., Anaki, D., & Faust, M. (2015). Processing of unconventional stimuli requires the recruitment of the non-specialized hemisphere. *Frontiers in human neuroscience*, 9, 32.

Shaw, G. A., & Conway, M. (1990). Individual differences in nonconscious processing: The role of creativity. *Personality and individual differences*, 11(4), 407-418.

Sinitsyn, D. O., Bakulin, I. S., Poydasheva, A. G., Legostaeva, L. A., Kremneva, E. I., Lagoda, D. Y., ... & Piradov, M. A. (2020). Brain activations and functional connectivity patterns associated with insight-based and analytical anagram solving. *Behavioral Sciences*, 10(11), 170.

Wallach, M. A., and Kogan, N. (1965). *Modes of Thinking in Young Children*. NY, New York: Holt Rinehart and Winston Inc.

6) Another limitation is, to my knowledge, the absence of psychometric validation of Mooney image solving as a metric for creativity.

Please see response to comment #5.

Response to reviewer #2:

Becker et al. present a single experiment in which participants recognize degraded images (Mooney figures, or camouflage pictures) during an fMRI scan, self-reporting various dimensions contributing to high- or low-insight ratings; then 5(?) days subsequently perform a memory test. fMRI signal during initial object recognition was associated with the insight-related ratings in several interpretable brain regions (e.g., related to object recognition, reward, memory), as well as increased connectivity among these regions for the high-insight trials. The authors reported changes in the relevant areas from initial stimulus presentation to the moment of recognition, and some of these changes were greater on trials for which participants reported (through the 3-dimension rating) high-insight compared to low-insight trials. Moreover, some of the fMRI markers were associated with improved recognition of the stimuli 5 days later.

Overall, this is an ambitious, challenging, but very well-motivated study. Finding neural markers (both areas and connectivity) that relate to insight visual recognition, to (their definition of) restructuring, and to later memory could all be important findings. The overall structure of the paradigm is well-reasoned and mostly well-described. The results and conclusions seem to make sense... but unfortunately the challenges of such a design have not – yet – been overcome.

The main difficulties facing this study are the low number of [true] observations per condition cell and, related but even more challenging, potentially highly impactful confounds across their main conditions of interest. Specifically:

1) There are far fewer true object recognition trials (in the scanner) with low-insight than with high-insight (this difference is compounded at memory). This results in lower power (lower # of trials = lower signal to noise ratio) to detect both fMRI signal and changes across conditions for the low-insight trials. The actual numbers of such trials aren't provided in the report (that I see), and have to be derived indirectly; even then, it's not always clear for various reasons.

We agree with the reviewer that the absolute number of trials should be provided. Note, in accordance with the reviewer's power concern (and the suggestion of reviewer #3), we now use the continuous insight sum (sum of all three dimensions: certainty, suddenness & positive emotion; ranging from 3:12) scale. This holds for all analyses except for some behavioral analysis (for demonstration purposes) and the connectivity/graph analyses (in CONN, there is currently no means of calculating connectivity/graph measures for continuous variables). Since this affects all analyses, please see the revised manuscript, as it is too much to copy paste here.

Nevertheless, we now plot the absolute amount of trials in the supplement as follows:

Figure S5. Amount of trials for each insight and insight memory condition

Note. Panel A: *HI*= high insight trials, *HI-cor*= correctly solved high insight trials, *HI-false* = incorrectly solved high insight trials, *LO* = low insight trials; *LO-cor* = correctly solved low insight trials; *LO-false*= incorrectly solved low insight trials; Panel B: *HI-r* = correctly solved high insight trials & later correctly recognized as having seen before; *LO-r* = correctly solved low insight trials & later correctly recognized as having seen before; *HI-rs*= correctly solved high insight trials & later correctly recalled as having solved before; *LO-rs* = correctly solved low insight trials & later correctly recalled as having solved before; *HI-m* = correctly solved and remembered high insight trials, *HI-fg* = correctly solved & forgotten insight trials, *LO-m* = correctly solved & later remembered low insight trials, *LO-fg*= correctly solved and later forgotten low insight trials.” (p.4f, Supplement)

We agree with the reviewer regarding the subdivision of LO-I trials into remembered and not remembered trials, as it results in a limited number of trials for the not remembered ones, as depicted in Fig.S5-B (Supplement), which is suboptimal. This issue is particularly pertinent to the connectivity/graph analyses, where we have to keep utilising the median split insight variable (binary). Nevertheless, in these analyses, we do not observe significant insight-memory effects. Additionally, it is worth noting that low trial counts would be more likely to increase type II error rather than type I error, leading to an absence of a consistent memory effect in the connectivity/graph analyses.

2) On top of that, there is the ambiguity of whether participants actually identified objects scored as “correct”, since this was only indicated in a 4-choice task (hence 25% chance rate, before correcting for possibility of narrowing the choices due to logical guesses – e.g., if the object appears as all straight lines, it’s probably not a mammal).

Concerning the accuracy metric, we agree with the reviewer that we lack certainty about the specific object identified by the participant during problem-solving.

Nevertheless, we assume that the proportion of falsely classified correct responses should be less than pure chance of 25%. The reason for this is the following: First, subjects were not always guessing when choosing a response option for every single Mooney image. This experiment was not a forced-choice task, as subjects had the possibility to refrain from pressing the solution button when unable to find a solution. Additionally, we emphasised a conservative response criterion in the instruction, i.e. that participants should only press the solution button when they were convinced that they had identified a solution. We also explicitly mentioned that we intentionally designed challenging Mooney images. Consequently, we did not anticipate participants to successfully solve every image. Of note, subjects had no advantage of pressing the solution button, as this resulted in effortful ratings and prolonged the overall experiment.

However, to get a more realistic estimate of the actual proportion of falsely classified correct responses in the Mooney identification task of the fMRI sample, we conducted a preregistered behavioral control experiment (note, the control experiment was mainly conducted as a response to the reviewer's comment #3d). This behavioral experiment was identical to the experiment of the fMRI sample with the difference that in this control experiment, instead of choosing one out of four response categories, participants expressed their solutions by typing the Mooney object identity (refer to Supplementary Material for further details), eliminating any ambiguity in their response. The proportion of falsely classified correct responses in the Mooney identification task of the fMRI sample was estimated as follows: Each of the subjects' solution words from the Mooney identification task in the control experiment was compared with the four possible response categories to determine whether it matched one of the response options that the subjects saw for each Mooney image in the Mooney identification task in the fMRI experiment (note, the four response categories for each Mooney image in the Mooney identification task were the same for every subject in the fMRI sample). The incorrect solution words from the Mooney identification task in the control experiment were then categorized according to whether they a) fell into the correct response category (i.e. falsely classified as correct response), b) into one of the other response categories (i.e. correctly classified as incorrect response) or c) into none of the specified response categories (i.e. here we assume that subjects randomly chose one out of the four response categories and therefore there is a 25% chance that their response is falsely being classified as correct response). The proportion of falsely classified correct responses in the control experiment was calculated from this categorization (amount of falsely classified correct responses divided by the sum of correctly and falsely classified correct responses). This proportion amounted to 18.7% ($SD=10\%$). The amount of correctly identified and subsequently remembered Mooney images in the fMRI experiment was finally adjusted by this proportion.

For a detailed description of the control experiment and its results, please see response to comment #3d). The accuracy adjustment procedure for the Mooney identification and subsequent memory task in the fMRI sample is explained in the manuscript as follows:

***“Accuracy Adjustment.* As participants selected one of four response categories in the Mooney identification task, some incorrect responses were classified as correct just by chance. To assess the proportion of falsely classified correct responses, a behavioral control**

experiment was conducted. The Mooney identification task in this control experiment mirrored the task in the fMRI sample, except that participants typed the solution word instead of choosing one out of four response categories, eliminating ambiguity. The proportion of falsely classified correct responses in the fMRI experiment was estimated by comparing the solution words from the Mooney identification task in the control experiment with the response categories of the same task in the fMRI sample. The incorrect solution words from the control experiment were then categorized based on whether they a) aligned with the correct response category (i.e., falsely classified as a correct response), b) aligned with one of the other response categories (i.e., correctly classified as an incorrect response), or c) did not align with any specified response category (i.e., random guess and therefore 25% chance that response is falsely classified as a correct response). The proportion of falsely classified correct responses of all correct responses in the control experiment was calculated from this categorization and amounted to 18.7% ($SD=10\%$, 7% came from random guessing and 11.7% aligned with the correct response category). The amount of correctly identified and subsequently remembered Mooney images in the fMRI experiment was finally adjusted by this proportion.” (p. 31f)

The behavioral results with the accuracy adjustment for the Mooney identification task in the fMRI sample is now described the manuscript as follows:

“On average, participants indicated a solution for 68.4% ($SD=16\%$) of all presented Mooney images. The solution, i.e. the subsequently selected category of the object, was correct in 43.0% (chance adjusted, $SD=13\%$) of all cases. [...]

Out of the 120 presented Mooney images, participants correctly solved and subsequently correctly recognized 36% (chance adjusted, $SD=10\%$). Out of all presented Mooney images, participants correctly solved and subsequently correctly recalled having solved them in 32% (chance adjusted, $SD=11\%$). Finally, out of all presented Mooney images, participants correctly solved and subsequently correctly identified them by name in 29% (chance adjusted, $SD=11\%$)” (p. 6f)

Finally, we acknowledge and address this limitation of the imprecise accuracy measure in the following manner in the discussion section (see more detailed response to whole limitation section further below):

“*Limitations*

Three key limitations should be addressed. Firstly, the accuracy metric is not exact as participants could not enter specific names of identified objects during problem-solving due to technical constraints in the scanner. Nevertheless, we assessed the likelihood of incorrectly classified correct responses through a control experiment (18.7%, refer to

Supplementary Material) and then adjusted the accuracy and subsequent memory measures accordingly for this proportion. [...]” p.25

3) Trials were “solved” much more quickly (mean 2.75sec) when reported to be with high- than with low-insight (5.08sec).

3a. This creates an untenable confound when assessing fMRI activation. Some trials (high-insight) unfold with stimulus presentation and solution highly conflated; on other trials (low-insight), the solution was much less conflated with stim onset, but possibly conflated with the end of the image presentation (which was always at 10sec, if I understand p. 37 correctly).

3b. Also, longer time-on-task (effort at recognizing, or early visual processes) could yield bigger fMRI signal effects in some areas (and smaller in others, either in absolute terms or relative to the whole span of the trial).

We acknowledge the reviewer's point about the conflation of insight with solution time. The issue, however, arises because shorter solution times and increased accuracy are inherent characteristics of the trials categorized as high(er) insight trials—those resolved with a notably higher AHA experience, ultimately displaying shorter reaction times and more accuracy. This trend has been observed across various insight tasks in different domains, as evidenced in studies by Becker et al. 2020, 2021, 2022; Salvi et al., 2016; Danek & Salvi, 2018. Thus, adjusting for accuracy and solution time by excluding incorrect trials, as well as fast high insight and slow low insight trials as proposed by the reviewer, results in a diminished count of *true* high insight and low insight trials (or reduced slope for the continuous insight variable), leading to a reduction in statistical power.

Nevertheless, we agree with the reviewer's concern that variations in solution time among conditions could result in a distinct BOLD signal amplitude between conditions resulting in a confound for univariate analyses. In order to address this issue and sidestep potential power challenges, we initially employed solution time and accuracy (excluding memory trials, where only correctly solved trials had been analyzed) as covariates of no interest. Given that all main analyses were calculated on a single trial level, differences in solution time (and accuracy) for every trial were accounted for without having to reduce the amount of trials with this method. It is worth noting that this method is a standard practice and has been widely employed in previous studies (Cnaan et al., 1997; Bates et al., 2014; Baayen et al., 2008; Becker et al., 2020).

However, to additionally provide evidence that the relationship between solution time and insight, does not explain the observed insight-related behavioral and brain effects, we now approached this potential confound from different angles:

First, in response to the reviewer comment #3a, all original analyses were re-run with a reduced number of trials excluding trials solved before 1.5 seconds and after 9.5 seconds to reduce the confound related to the beginning and end of the stimulus presentation when assessing fMRI activation:

“To eliminate trials that might solely involve object recognition without insight components, we excluded all trials with solution times faster than 1.5 seconds from our analyses. Additional details regarding the estimation of this threshold can be found in the Supplement and Fig.S3. To further address a potential confound associated with the offset of the Mooney image (a concern specific to fMRI analyses but to maintain this consistently across all analyses), we also excluded any trials that were solved after 9.5 seconds. The average amount of remaining trials per condition is illustrated in Fig S5.” (p.31)

For a detailed description of how the 1.5 seconds after stimulus presentation were estimated, please see comment to reviewer question #3c. In summary, these modifications did not significantly change the obtained results (except for the insight-related memory results obtained via AlexNet, see response to #3c). Since this affects all analyses, please see the revised manuscript, as it is too much to copy paste here.

Second, in response to reviewer comment #3b, differences in time-on-task may affect the fMRI results, in particular the amplitude of the BOLD signal. We address this issue differently for univariate and multivariate analyses and dedicate a whole section in the Supplementary Material (“**Further control analyses to account for the influence of solution time**” p.6ff, Supplement) for this:

Univariate fMRI results:

Incorporating the amplitude of the BOLD signal as a covariate of no interest was not feasible, given that the BOLD signal serves as the dependent variable. As suggested by the reviewer (see suggestion further below), we therefore re-sampled the data, i.e. adjusted for solution time between high-insight and low-insight trials at the cost of reduced power (but please note that we have shifted to performing most analyses using the continuous insight measure rather than the median split, with the exception of connectivity/graph analyses, where such an analysis is not feasible). When we say 'adjust,' it means that we removed fast high-insight trials and slow low-insight trials until the time taken to solve both high and low insight trials was no longer significantly different ($p > 0.20$). Importantly, after adjusting for solution time, insight (continuous variable) still significantly positively predicted variance in the BOLD activity of hippocampus and amygdala:

“Further control analyses to account for the influence of solution time

We conducted additional control analyses to eliminate the potential influence of solution time as the sole factor responsible for the observed effects associated with insight. [...]

Univariate BOLD analysis: Similar to the behavioural analysis, we examined the impact of removing fast HI-I and slow LO-I trials on BOLD activity (fast HI-I and slow LO-I trials were removed until the time taken to solve both types of trials was no longer significantly different [$p > .20$]). Subsequently, we conducted repeated single-trial mixed-model analyses investigating BOLD activity in amygdala and hippocampus as a function of a) insight (continuous) as well as b) insight*memory controlling for run (order) and random subject and item intercepts. A) insight still significantly predicted BOLD activity during solution

($Chi^2(1)=32.43$, $p<.001$, $\beta=.09$) while there was no significant difference between all three ROIs (anterior, posterior hippocampus and amygdala) regarding this relationship ($Chi^2(2)=2.32$, $p=.31$). B) similarly for insight, there was a significant insight*memory interaction ($Chi^2(1)=8.09$, $p<.01$, $\beta=.08$) predicting BOLD activity in all three ROIs while there was no evidence for a three-way interaction (ROI*insight*memory: $Chi^2(6)=5.79$, $p=.44$) suggesting no significant difference between the ROIs. A Post Hoc analysis revealed that the slope for insight during remembered trials was significantly positive ($z=6.23$, $p<.001$, $\beta=.17$) while it was not significantly different from zero during forgotten trials ($z=1.65$, $p=.10$, $\beta=.05$).” (p.6ff, Supplement)

Multivariate fMRI (RSA) results:

We agree with the reviewer that time-on-task can influence the magnitude of the univariate BOLD signal. However, Spearman’s r and Pearson’s r , the correlational measures we used as proxy for representational change (RC), are magnitude insensitive (Dimsdale-Zucker & Ranganath, 2018). It is therefore not plausible to assume that differences in BOLD signal due to time-on-task should affect the results in multivariate voxel patterns.

It is furthermore essential to highlight that, due to the multivariate nature of these measures, the varied solution times of individual trials, independent of condition, are inherently intertwined with this measure. The reason for this is because this multivariate measure for RC consists of a correlation between the multivoxel pattern of a solved Mooney object in ROI x at time t with the multivoxel pattern of all other solved Mooney objects in the same ROI at time t . Therefore, excluding the fastest high insight and slowest low insight trials does not lead to the desired result, as each trial’s value reflects its correlation with all other trials, both fast and slow.

However, to still address a potential influence of the BOLD signal’s amplitude at the time of solution on the relationship between insight and multivariate voxel patterns in visual areas (reflecting RC), we carried out two different control analyses:

First, we investigated the BOLD signal in both VOTC-RC areas (pFusG, iLOC) at the time of the solution as a function of insight and solution time. That is to say, if the BOLD signal due to time-one-task has an influence on multivariate measures, then this is only relevant when there is an actual systematic BOLD signal difference in those areas due insight and/or solution time. We found no systematic relationship between the BOLD signal in both areas and insight or solution time. This was reported in the Supplement as follows:

“Univariate fMRI results of VOTC RC areas

For control purposes, aiming to examine whether univariate brain activity in VOTC-RC regions could account for the observed multivariate differences related to insight, we explored the univariate brain activity in pFusG and iLOC in relation to insight. To investigate correctly solved insight trials, we performed the same single trial analyses as for amygdala

and hippocampus controlling for run order, solution time and including random subject and item intercepts.

Regarding univariate effects of insight on VOTC-RC areas, there was no significant overall condition effect ($Chi^2(1)=3.15$, $p=.075$, $\beta=-.02$) but a condition*ROI interaction ($Chi^2(1)=46.91$, $p<.001$, $\beta=.12$) effect. Post Hoc analyses revealed a negative effect of insight on BOLD activity in iLOC ($z=-6.15$, $p<.001$) but no evidence for an effect in pFusG ($z=1.56$, $p=.12$) (see Fig.S5).

Moreover, we examined potential time-on-task influences on the BOLD signal in both VOTC-RC regions (Yarkoni et al., 2009). To explore this, we replicated the single-trial analyses from above, excluding insight as a predictor. Instead, we computed a solution time*ROI interaction, controlling for run order and incorporating random subject and item intercepts. We found a significant solution time*ROI interaction ($Chi^2(1)=131.87$, $p<.001$, $\beta=-.21$). Post Hoc analyses revealed a positive effect of solution time on BOLD activity in iLOC ($z=2.92$, $p=.003$) but a negative effect of solution time on BOLD activity in pFusG ($z=-13.70$, $p<.001$).

In summary, the inconsistent univariate effects observed in iLOC and pFusG in relation to insight and solution time make it implausible to attribute the observed consistent multivariate RC effects in both VOTC regions on univariate brain activity.

Figure S6. Relationship between univariate BOLD activity in RC VOTC areas and insight.”

(p.5f, Supplement)

Second, to further address the potential influence of the BOLD signal's amplitude at the time of solution on the relationship between insight and the multivariate voxel pattern in visual areas (representing RC), we incorporated the BOLD signal as a covariate of no interest into all RC analyses (instead of the solution time variable). It was anticipated that if the relationship between these multivariate measures and insight were entirely modulated by the size of the BOLD signal (as a function of time-on-task), the variance of different multivoxel pattern measures (such as pattern similarity and RSA measures) explained by insight would cease to be significant. However, the hypothesized relationship remained significant even with the inclusion of the BOLD signal as a covariate of no interest at the single trial level.

"Multivariate BOLD analysis: Making adjustments for solution time in these correlational measures by excluding fast HI-I and slow LO-I trials is not plausible, as each trial represents a correlation of the multivoxel pattern of the current trial in a given ROI with the pattern of every other trial, whether solved fast or slow. However, to account for the potential impact of BOLD activity on the relationship between insight and multivariate measures due to potential time-on-task effects, we conducted a repetition of the aforementioned multivariate RC analyses. This time, we directly incorporated BOLD activity during solution as a covariate of no interest in exchange for solution time additionally controlling for run (order) and random subject and item intercepts.

Insight remained a significant negative predictor for multivoxel pattern similarity (first RC measure) from pre to post solution ($Chi^2(1)=313.08$, $p<.001$, $\beta=-.14$) for both VOTC RC regions when controlling for run (order) and BOLD activity during solution. Post Hoc tests showed that pFusG ($Chi^2(1)=211.61$, $p<.001$, $\beta=-.22$) as well as iLOC ($Chi^2(1)=155.70$, $p<.001$, $\beta=-.25$) reduced their multivoxel pattern similarity from pre to post solution with increasing insight.

Furthermore, the positive insight*time(post solution) interaction predicting representational strength (AlexNet, second RC measure) remained significant ($Chi^2(1)=15.19$, $p<.001$, $\beta=.06$) for both RC regions (pFusG, iLOC) when controlling for BOLD activity during solution next to run (order) and random subject and item intercepts. There was no evidence that this effect differed between both ROIs ($Chi^2(3)=1.98$, $p=.57$). Post Hoc analyses revealed that insight negatively predicted representational strength during stimulus onset ($z=-2.27$, $p=.023$) but positively during solution ($z=2.79$, $p=.005$) over both ROIs. Regarding insight-related memory, there was no notable interaction between insight, memory(remembered), and time (post solution) ($Chi^2(1)=0.71$, $p=.40$) similar to the outcomes when employing solution time instead of BOLD activity at the solution as a covariate. In contrast, the interaction between memory(remembered) and time (post solution) remained significant ($Chi^2(1)=36.26$, $p<.001$, $\beta=.21$).

When using Word2Vec as a conceptual model to compute representational strength, the insight*time(post solution) interaction predicting representational strength (second RC measure) remained significant ($Chi^2(1)=16.12$, $p<.001$, $\beta=.06$) for both RC regions (pFusG, iLOC) when controlling for BOLD activity during solution next to run (order) and random subject and item intercepts. There was no evidence that this effect differed between both ROIs ($Chi^2(3)=1.57$, $p=.66$). Post Hoc analyses revealed that insight positively predicted representational strength during solution ($z=3.67$, $p<.001$) while there was no evidence for a difference in representational strength as a function of insight during stimulus onset ($z=-1.55$, $p=.122$). Furthermore, there was a significant positive insight*memory(remembered)*time(post solution) interaction ($Chi^2(1)=10.46$, $p<.01$, $\beta=.09$) predicting representational strength over both brain regions (iLOC, pFusG) while this interaction was not significantly different between both ROIs ($Chi^2(7)=6.63$, $p=.47$).

In summary, the outcomes remained largely consistent when incorporating BOLD activity during the solution as a covariate of no interest instead of solution time in the main analyses outlined in the results section.” (p.9ff, Supplement)

Baayen, R. H., Davidson, D. J., and Bates, D. M. (2008). Mixed-effects modeling with crossed random effects for subjects and items. *J. Mem. Lang.* 59, 390–412. doi: 10.1016/j.jml.2007.12.005

Bates, D., Mächler, M., Bolker, B., & Walker, S. (2014). Fitting linear mixed-effects models using lme4. arXiv preprint arXiv:1406.5823.

Becker, M., Wiedemann, G., & Kühn, S. (2020). Quantifying insightful problem solving: A modified compound remote associates paradigm using lexical priming to parametrically modulate different sources of task difficulty. *Psychological research*, 84, 528-545.

Becker, M., Kühn, S., & Sommer, T. (2021). Verbal insight revisited—dissociable neurocognitive processes underlying solutions accompanied by an AHA! experience with and without prior restructuring. *Journal of Cognitive Psychology*, 33(6-7), 659-684.

Becker, M., Davis, S., & Cabeza, R. (2022). Between automatic and control processes: How relationships between problem elements interact to facilitate or impede insight. *Memory & Cognition*, 50(8), 1719-1734.

Cnaan, A., Laird, N. M., & Slasor, P. (1997). Using the general linear mixed model to analyse unbalanced repeated measures and longitudinal data. *Statistics in medicine*, 16(20), 2349-2380.

Danek, A. H., & Salvi, C. (2020). Moment of truth: Why Aha! experiences are correct. *The Journal of Creative Behavior*, 54(2), 484-486.

Dimsdale-Zucker, H. R., & Ranganath, C. (2018). Representational similarity analyses: a practical guide for functional MRI applications. In *Handbook of behavioral neuroscience* (Vol. 28, pp. 509-525). Elsevier.

Salvi, C., Bricolo, E., Kounios, J., Bowden, E., & Beeman, M. (2016). Insight solutions are correct more often than analytic solutions. *Thinking & reasoning*, 22(4), 443-460.

3c. Another issue here is that VERY quick responses may simply reflect simple object recognition (they immediately see a puppy, in a slightly blobby picture), rather than “solving” in any normal usage; furthermore, these are more likely to get rated as “high-insight” (due to suddenness and certainty, and perhaps surprise that they’re getting an easy trial). But are they true insight-solving? Some in the field (including us) have taken to eliminating the rapid responses from analyses (see e.g. Cransford & Moss, 2012 J. Prob Solving).

We agree with the reviewer's assessment, that exceedingly rapid responses likely pertain solely to basic object recognition and should be excluded, as they might not reflect insight or problem solving trials. Object recognition typically transpires within the milliseconds range. To obtain a more precise estimation of object recognition time for our particular stimuli, we chose a subset (n=78) of our Mooney images, for which we possessed the corresponding gray-scaled (non-abstracted) images, and tested those images on a distinct sample. Object recognition time was ~753ms:

“Estimating threshold for insight trials based on object recognition time

To eliminate trials that might solely involve object recognition and no insight, we excluded all trials with solution times faster than 1.5 seconds ($>5*SD$ of average object recognition time) from our analyses. The time for object recognition was informed by a control experiment, wherein a subset of gray-scaled images (N=78), the same set used to generate the Mooney images, was presented to 40 subjects (19 females, $M=27.3$ years, $SD= 5.6$ years) in an online test recruited via Mechanical Turk and a German student platform of Humboldt University. Their objective was to press a solution button upon identifying the object, mirroring our Mooney identification task. Median correct object recognition time for these gray-scaled images was 753.2ms ($SD=140.1$ ms) (see Fig. S3).

Figure S3. Mean response time for object recognition of gray-scaled images.

Note. This is a subset (N=78) of those images that were used to create the Mooney images. Line represents median response time.” (p.3, Supplement)

Based on this information, we selected a threshold for 1500ms (= $>5 \times SD$ of the averaged correct object recognition time) to exclude those trials.

Note, as emphasized by the reviewer (and already discussed further above), the endpoint of stimulus presentation may also introduce a potential confounding variable in the BOLD signal that is unrelated to insight. In response to both concerns raised, we conducted a comprehensive reassessment and reanalysis of all the original manuscript analyses. Specifically, we excluded trials solved before 1.5 seconds and after 9.5 seconds (see response to reviewer question #3a).

For a detailed account of the changes made to the results, please refer to the revised manuscript, as it is impractical to reproduce all alterations here. In summary, the overall findings remained consistent, with the only notable difference being that the second restructuring measure, incorporating AlexNet as the conceptual model, no longer demonstrated an insight-memory advantage (while Word2Vec did). We interpret these results in the discussion section as follows:

“Note, the RSA analyses using AlexNet did not show a significant correlation with insight-related better memory, whereas Word2Vec did. This difference may be attributed either to issues of statistical power or to the observation that semantic representations, linked to better subsequent memory (Craik & Lockhart, 1972), may be more effectively captured by the Word2Vec model than by the vision-based AlexNet model.” (p.24)

3d. The time difference in solving also creates a confound for the subsequent memory task. This could be viewed two ways. If participants really only “studied” the image prior to “solving”, then they had less study time for high-insight than low-insight trials. This would suggest the better memory for high-insight trials is even better! OTOH, if participants continued to examine the image as long as it remained on the screen (again, a constant 10 sec), then they had longer to examine the OBJECT (10s – 2.75 = 7.25 sec) for high-insight trials than for low-insight trials (10s-5.08 = 4.92sec). I.e., they had more time to think of the object, to see how the image details comprise the object, etc., for high- than for low-insight trials.

The reviewer suspects that the time difference in solving could introduce a confounding factor for the subsequent memory task, as quicker solution times may leave more time for encoding the Mooney object. We acknowledge this as a plausible confound. In addition to using solution time as a covariate of no interest, which we had already implemented, we now also adjust for solution time in the behavioral analyses. In this context, "adjusting" solution time entails excluding slow low insight solution trials and fast high insight solution trials to the extent that solution time no longer significantly differs between both conditions ($p > .2$), as recommended by the reviewer. The results still show an insight memory advantage:

“Behavioral analysis: We adjusted solution time between HI-I and LO-I trials by excluding fast HI-I and slow LO-I trials until the time taken to solve both types of trials was no longer significantly different ($p > .20$). Subsequently, we repeated the mixed model analyses investigating memory as a function of insight (continuous) for correctly solved trials controlling for run (order) and random subject and item intercepts. The results did not change significantly, insight (continuous) still predicted memory ($Chi^2(1)=63.83$, $p < .001$, odds ratio: 1.46). This result dismisses differences in solution time as potential explanation for the insight memory advantage.” (p.6, Supplement)

We furthermore conducted a preregistered control experiment without stimulus blocks of 10 second, i.e. the stimulus disappears after pressing the solution button dismissing the chance for longer encoding time of the stimulus object in higher insight trials. The insight memory advantage was still replicated:

“We also conducted a preregistered control experiment (<https://aspredicted.org/xx7hv.pdf>) with originally 33 subjects and a final sample of N=27 subjects (70% females, M=23.0 years ($SD=4.1$ years)), recruited through the Humboldt University student online platform PESA. In this control experiment, participants replicated the same procedures as in the fMRI experiment, using the identical 120 Mooney images. There were two modifications. First, the stimulus disappeared upon pressing the solution button, eliminating potential confounds related to differences in encoding time for the identified Mooney object. Second, the subjects typed their solution instead of selecting a response category which eliminated ambiguity regarding their solution. Participants performed the Mooney identification task in the lab, and

five days later ($SD=0.3$ days), they underwent an online subsequent memory test. Six participants were excluded from further analyses due to producing too few correct hits (see preregistration). The data analysis followed the same procedures as for the fMRI sample, with the exception that there was no run (order) variable, as participants completed the experiment in a single run. A distribution of all trials is depicted in Fig. S7-A.

On average, participants found a solution in 45.5% ($SD=15\%$) for all presented Mooney images (see Fig. S7-A). The solution was correct in 60.2% ($SD=17\%$) of these cases. Of all correctly identified Mooney images, 57% were solved with *HI-I* and 43% with *LO-I* trials ($SD=12\%$). Consistent with previous results and the fMRI sample, this *insight-accuracy* effect was significant ($Chi^2(1)=157.18$, $p<.001$, odds ratio=1.54) (Becker, Wiedemann, et al., 2020; Danek & Salvi, 2020). The median response time for correct solutions was 4.3sec ($SD=0.77$ sec) where participants were faster during *HI-I* (3.6sec, $SD=0.67$ sec) than *LO-I* solutions (5.3sec, $SD=1.0$ sec). Similar to the fMRI sample, insight significantly predicted response time ($Chi^2(1)=207.7$, $p<.001$, $\beta=-.21$).

Participants recognized 49.0% ($SD=16.5\%$) of the 120 presented Mooney images, regardless of whether they had correctly identified them, with a false alarm rate of 16.3% ($SD=12.1\%$). Out of all presented Mooney images, participants correctly solved and subsequently correctly recognized 18% ($SD=9\%$) and they correctly solved and subsequently correctly recalled having solved them in 15% ($SD=8\%$). Finally, out of all presented Mooney images, participants correctly solved and subsequently correctly identified them in 15% ($SD=8\%$) (see Fig.S7-B for an overview of a distribution across all conditions). The insight factor (*HI-I*, *LO-I* and unsolved trials) significantly predicted variance in subsequent memory ($Chi^2(2)=114.83$, $p<.001$, odds ratio (linear trend for not solved < *LO-I* < *HI-I*) = 7.48; see Fig.S7-C). Importantly, when exchanging the continuous insight variable in the mixed model with the binarized insight variable additionally controlling for solution time (excluding unsolved trials), insight still significantly predicted subsequent memory ($Chi^2(1)=25.26$, $p<.001$, odds ratio = 1.28) (see Fig.S7-D).

Figure S7. Behavioral Insight Memory Effect - control experiment.

Note. **Panel A.** overall proportion of all trials per condition; **Panel B:** (remembered = Mooney object correctly identified) *HI-r* = correctly solved high insight trials & later correctly recognized as having seen before; *LO-r* = correctly solved low insight trials & later correctly recognized as having seen before; *HI-rs* = correctly solved high insight trials & later correctly recalled as having solved before; *LO-rs* = correctly solved low insight trials & later correctly recalled as having solved before; *HI-m* = correctly solved and remembered high insight trials, *HI-fg* = correctly solved & forgotten insight trials, *LO-m* = correctly solved & later remembered low insight trials, *LO-fg* = correctly solved and later forgotten low insight trials. **Panel C.** Behavioural insight memory effect; *HI-I* = correctly solved high insight trials; *LO-I* = correctly solved low insight trials; estimates are marginal mean percentages to correctly recall the previously identified Mooney image in the respective condition. Error bars represent 95% confidence intervals. **Panel D.** relationship between continuous insight variable (3-12) and memory controlled for solution time; estimates are marginal mean percentages to correctly recall the previously identified Mooney image. Error bars represent 95% confidence intervals.

To summarize, the behavioral control experiment replicated the insight memory advantage observed in the fMRI sample. These findings dismiss longer encoding time of Mooney objects as a potential explanation for the observed insight memory advantage.” (p.6ff, Supplement)

What to do? I recognize the very clear challenges of implementing such a paradigm,

and the important nature of the questions being asked. It's also possible that the confounds outlined above do not adversely affect results or interpretations, but I don't see an obvious way to be sure – as the paper now stands. These confounds are very important, and must be dealt with better. I know the authors report that they included reaction time and (sometimes) accuracy as “covariates of no interest”, but I don't think that is sufficient. One possible path is to re-sample the number of trials. Specifically, drop the fastest “high-insight” trials (which were probably included a fair number of instant recognition trials anyway), and the slowest “low-insight” trials, until you have data sets that are more comparable (I'd love them to have equal means and distributions, but that's probably not achievable with this dataset). If the same results occur, you have more assurance they are not due to the time confound. Problem is, there are already so few accurate low-insight trials (their reporting of exactly how much data [trials/sub per condition cell] remains unclear), so it might be hard to examine meaningful subsamples.

Because the paper is potentially important, it could merit publication IF the authors address the issues (with evidence, hard stats on the least-confounded data they can provide) as much as possible, but are also completely transparent and forthright about the potential weaknesses – low # of trials and possible confounds. When I say “forthright” I mean such info is given in passing or in a footnote, but is seriously discussed. (This would be a welcome trend for the whole field, not just this paper – too many papers try to sell conclusions as rock solid when they are actually hints that need further verification through additional papers being published).

We hope that we have provided additional evidence to demonstrate that our findings are not just task confounds that are unrelated to the insight memory effect. Nevertheless, we agree with the reviewer to be completely transparent about the potential problems with a relatively low number of trials per condition, resulting in low power and an inherent correlation between solution time, accuracy and insight. For this reason, we now discuss this issue more in depths in the discussion's limitation section:

“Limitations

Three key limitations should be addressed. Firstly, the accuracy metric is not exact as participants could not enter specific names of identified objects during problem-solving due to technical constraints in the scanner. Nevertheless, we assessed the likelihood of incorrectly classified correct responses through a control experiment (18.7%, refer to Supplementary Material) and then adjusted the accuracy and subsequent memory measures accordingly for this percentage. Secondly, the limited number of trials in the memory condition, especially for forgotten trials, may compromise the statistical power of results and introduce interindividual variability. However, these limitations are anticipated to impact type II errors more than type I errors. Thirdly, we observe a consistent correlation between solution time and our insight measure, a phenomenon documented in previous studies (Becker et al., 2021), which, however, can induce time-on-task effects on the size of the BOLD signal and therefore bias the results (Yarkoni et al., 2009). To address this issue, we

incorporated solution time as a covariate in all reported analyses on a single trial level next to other efforts detailed in the Supplement (adjustment of solution time between *HI-I* and *LO-I* trials etc.). However, all control analyses consistently demonstrated a systematic relationship between insight/memory and the corresponding brain measure.” (p. 25)

Beyond the above issues, there are some minor interpretive ones, easily handled in revisions.

4) Conceptually, is this type of vis-aha the same as other insights, such as semantic reinterpretation? This isn't a big issue for me, and there have long been some connections between the phenomena (including by Gestalt psychologists). But it would be worth discussing, and some relevant literature (e.g., Bar et al., 2001, but others I list below) that should probably be noted and discussed in relation to the specific results uncovered (e.g., VOTC, LOC & Fusiform areas). There are certainly similarities between vis-aha! and other insights, and many people have treated it as more or less the same as other insights. But there could be some important differences. E.g., is it the same to go from basic visual features (in this case, blobby black or white components) to object recognition, as it is to go from a standard but incorrect semantic association or problem representation to a less salient one? For these visual stimuli, there isn't necessarily an initial misinterpretation to be overcome, or a dominant solving path that leads the solver/viewer astray. In fact, this paradigm is similar to straightforward object recognition paradigms (e.g., Bar et al., *Neuron*, 2001; Doniger et al., *JoCN*, 2000), although those paradigms gradually increased the amount of visual information contributing to object recognition (but still see some top-down effects, e.g. Bar et al., 2006). Much of the current discussion ignores the object recognition literature or glosses over the distinction between different types of insight. This is not a fundamental problem, nor is it unique to these authors. I'd just like to see both ideas noted in the manuscript (initially framing the paradigm as eliciting “a type of insight”, noting it shares some but not all characteristics of other insights; AND noting the similarity to these object recognition literature like Bar, [etc]. In this paper it matters because some of the key DVs (including fMRI regions) could be pretty closely yoked to object recognition.

The issue of demonstrating restructuring is important, and it's great that this paper tries to provide some evidence. Many/most studies (including most of ours) have simply assumed or posited that restructuring occurs. There have been some prior attempts to provide evidence, including using graph theory. One had subjects make semantic relatedness judgments and found that subjects who had immediately prior solved problems with related concepts demonstrated different (changed) semantic networks compared to subjects who had not solved the problems (Durso et al., 1994). (The PsyArXiv paper you cite by Bieth, Kennett et al. could also be discussed more, as it's highly relevant - but I'm not sure it's published yet). As noted above it may remain a question whether going from no-object (so, only activation for basic visual features) to object perception is the same as semantically restructuring a problem.

But that's just a conceptual that could be addressed (one could argue it's perhaps a nomenclature issue); further, while that may explain the increased signal in object recognition areas, it might not as easily explain some of the other effects (other areas & connectivity results).

The reviewer raised an interesting and relevant point. First, we now mention more specifically where “visual” insight is (conceptually) different from “verbal” insight and also cite Durso et al. (1994):

“The main cognitive component is associated with *representational change* (RC) [...]. **As an example, in verbal problem-solving, RC may encompass semantic reinterpretation or regrouping conceptual relationships** (Becker, Sommer, et al., 2020; Durso et al., 1994). **In contrast,** visual problems that demand object recognition, RC may involve breaking up perceptual chunks into their constituent elements to make them available for a novel recombination (Tang et al., 2015).” (p.2)

“However, we did not find RC in anterior VOTC (aFusG and aITG), which processes more abstract concepts and semantics (Bussey et al., 2005; Clarke & Tyler, 2014). This could reflect the typically lower signal-to-noise ratios (SNR) in the MRI of this region (Embleton et al., 2010) or the visual nature of the Mooney identification task. **Verbal restructuring likely involves regrouping conceptual relationships in verbal tasks** (Becker, Sommer, et al., 2020; Durso et al., 1994). **Hence,** future research could examine if insight tasks that require more abstract semantic processing (**e.g. verbal restructuring**) yields RC in anterior VOTC regions.” (p.22)

Note, Bieth, Kennett et al., (2021) is not published yet, for this reason why we did not discuss the paper in full length yet.

Second, we rename insight to “visual” insight to emphasize that what we are investigating here is specifically related to “visual” insight, which is one type of insight.

“To investigate this neural mechanism of **(visual)** insight and its impact on subsequent memory, the current study scanned participants with fMRI while performing a *visual insight task*, [...] Perceptually rechunking or regrouping these contours can lead to sudden object identification, eliciting a **type of** insight (Kizilirmak, Thuerich, et al., 2016; Ludmer et al., 2011). [...] To evaluate this neural mechanism of **(visual)** insight and its impact on memory, we tested four specific fMRI hypotheses. ” (p.3f)

“Using a visual Mooney identification task, we found **(visual)** insight-related RC in middle VOTC regions (pFusG and iLOC), which are areas that process complex visual features (Bussey et al., 2005; Clarke & Tyler, 2014).” (p.21)

Third, we now briefly discuss the similarity of our results with the suggested object recognition literature (we thank the reviewer for those literature suggestions):

“We focused on six VOTC regions-of-interest (ROIs) associated with visual object recognition (posterior and anterior fusiform gyrus–pFusG, aFusG; inferior lateral occipital complex–iLOC; anterior, mid, and posterior inferior temporal gyrus–aITG, mITG, pITG) (Bar et al., 2001).” (p.5)

“From pre to post solution, the conceptual representation of the target objects increased in pFusG and iLOC, suggesting a change from representations coding for meaningless black and white Mooney image blobs to representations coding for meaningful objects. These results align with prior research, reporting the presence of bilateral ERP components (N_{ci}) in LOC during the recognition of objects in fragmented images, similar to Mooney images (Doniger et al., 2000; Sehatpour et al., 2006). Note, there is limited knowledge whether the process that leads up to RC, e.g. the transition from meaningless to meaningful objects, occurs gradually or discretely and is only perceived as sudden. However, previous research suggests a linear correlation between BOLD activity in the fusiform gyrus as well as occipital temporal sulcus and awareness of object identity (Bar et al., 2001), indicating a more continuous process.” (p.21)

Ultimately, to illustrate a degree of similarity in the problem-solving process between visual Mooney images and verbal insight tasks, we conducted a subject-level correlation between the insight experiences in solving Mooney images and anagrams. The observed correlation was statistically significant ($r = 0.68$). This was described in the manuscript as follows:

“Association with verbal insight tasks

We further investigated whether solving Mooney images reflects a similar underlying creative problem-solving process (see Kenett et al., 2015) as the more commonly used verbal insight tasks. To assess this relationship, we computed a Pearson correlation between the participants' average continuous insight experiences in Mooney images and an anagram task (see Fig.2-D). The obtained correlation was statistically significant ($r=0.68$, $CI[0.43, 0.83]$, $t(29)=5.013$, $p<.001$), indicating that the insight experience evoked by Mooney images is comparable to that of another verbal insight task.” (p. 8)

Additional specific / short (and probably garbled) notes:

5) It's not entirely clear whether the authors are only analyzing “correct” trials when examining fMRI during high- vs low-insight recognition. Probably, but definitely should be, and they should be explicit (EVERY time say “on correct high-insight trials” etc).

We only include correctly solved trials into all (brain) analyses and only the ones solved between 1.5 and 9.5 seconds (to exclude mere object recognition trials as well as trials

confounded with the stimulus offset, see response further above). We now have mentioned every time that only correct trials were used for analyses (note we use the continuous insight measure for almost all analyses now, hence without the differentiation between HI-I and LO-I):

“Behavioural results

Insight predicts better subsequent memory

[...] The first model predicted subsequent memory including a **categorical insight factor consisting of not solved trials, correctly solved HI-I and LO-I trials**. [...]

First, by correlating the activity patterns pre (0.5 sec after Mooney image onset) and post solution (button press), and by determining if the correlation decreased more strongly (i.e., greater change) with **an increase in correct insight** (see Fig.3-A). Second, in areas showing greater **multivariate activity pattern** change as a necessary condition for **correct** insight-related RC we tested whether this change reflects a stronger conceptual reorganisation of the identified Mooney object from meaningless to meaningful. [...]

RC from pre to post solution: Representational strength - AlexNet. [...] To identify areas where the representational strength increased more strongly from pre to post solution for **correct** insight (see Fig.3-B), we estimated a mixed model with the factors time (pre vs. post), insight and ROI as well as their interaction. [...]

Hypothesis 2: Amygdala and hippocampus show insight-related activity

To examine our second hypothesis that the evaluative component of insight is mediated by the amygdala and anterior/posterior hippocampus, we **estimated whether** amygdala and hippocampal activity during **correct** Mooney object identification **would parametrically vary depending on the strength of insight**. [...]

Amygdalar and hippocampal activity are significantly parametrically modulated by insight during **(correct)** solution ($t(182)=5.043, p<.001$) but there are also differences between the ROIs ($Chi^2(2)=22.33, p<.001$). [...]

Activity in amygdala and hippocampus is associated with insight on a trial by trial level when controlling for response time [...]

For this analysis, we compared two models of **correctly** solved trials for insight as dependent variable and response time and run order (1-4) as covariates of no interest as well as a random subject and item intercepts. [...]

Increased functional connectivity of the solution network during insight

[...] The resulting connectivity values were predicted by the insight factor (**correct HI-I>LO-I**) in CONN. [...]

Insight-related RC predicts better subsequent memory.

First, we tested whether insight-related better memory **for correctly solved trials** is associated with our two consecutive measures of RC, i.e. [...]

Insight-related anterior hippocampal activity predicts better subsequent memory.

[...] For this, we used the data from the single-trial analyses containing a beta estimate of univariate amygdala, anterior and posterior hippocampal activity (ROI) for the solution button press at each **correctly solved** trial. " (p.7ff)

6) Still, those "correct" trials are contaminated by lucky guesses on the categorization response made in the scanner, and in a confounded manner.

a) What qualifies as "correct" in initial task in scanner is less than ideal: Correctly selecting the category (e.g., mammal or "objects in the house") from 4 choices. So, participants start with 25% chance of choosing correct category. And, even without having full object recognition, they might be able to narrow down categories based on features – e.g., a picture with more straight lines and sharp angles is less likely to be a "mammal" than an "object". So, "chance" might be better than 25%. And yet, as far as I can derive from presented data (%ages in different conditions), it appears that subject made "correct" response on only 38.5% of the trials in which they responded but trial was categorized as low-insight.

b) Because of the confound with accuracy, the "low-insight" trials, besides being lower in number, are more contaminated by lucky guesses on the categorization task. (If 25% were lucky guesses, only 13.5% were correct/real identifications – and it could be worse if we factor in strategic guesses). Thus, again, the signal for the already small # of low-insight trials might be contaminated by a great deal of noise.

We kindly direct the reviewer to our responses for reviewer points 1) and 2), where we have addressed the concerns raised in this regard.

7) OTHER

Claims for insight boosting memory should be a bit more restrained; each of the cited studies has some limitations. Ludmer et al, e.g., didn't compare insight "solving" to non-insight memory, per se (if someone "solved" a stimulus, that trial was excluded from analyses); other older studies on memory (Auble et al, 1979) sometimes had a confound with study time/ time on task, etc

We agree with the reviewer that specifically the study by Ludmer has methodological problems but we are not aware that all studies showing the insight memory advantage in particular the study by Kizilirmak et al. 2016 (Mooney images) or Danek et al 2013, 2020

(Magic tricks) have those mentioned methodological problems. For this reason, we have changed the citations as follows:

“Insight problem solving, which is often studied in the context of scientific discoveries (Kounios & Beeman, 2015), **has been** associated with boosted subsequent long-term memory, sometimes after a single experience, unlike other forms of learning that require multiple repetitions (**Danek et al., 2013; Danek & Wiley, 2020; Kizilirmak et al., 2016; Kizilirmak & Becker, 2023**).” (p.2)

Note, we have now added a review book chapter (Kizilirmak & Becker, 2023) on the insight memory advantage which discusses the current insight-memory literature (including weaknesses of the studies) and is therefore less problematic to cite.

In another sentence we interchanged “demonstrated” for “indicated” to restrain the insight-memory claim a bit more:

“However, both insight components likely contribute to this effect, as **indicated** in behavioural studies (Ash & Wiley, 2008; Auble et al., 1979; Danek et al., 2013; Danek & Wiley, 2020; Kizilirmak, Galvao Gomes da Silva, et al., 2016).” (p.3)

8) RESULTS

P 6:

Would like very clear upfront statements about # of trials in each condition being analyzed - presumably, e.g., 68.5% of 120, but please state, it helps with assessing stability of data measures. Also would be good to give a range for each of these measures. E.g., 68.5% (responded) x 51.8% (correct) x 34.7% (low-insight) x 120 =~ 14.8 low-insight identifications. Some subs were below average, so you had very few trials from this category to analyze, let alone in the remembered vs non-remembered subdivisions of the category. Did you include subs who actually had very few correct low-insight trials in the scanner? (I know it's very hard to get large numbers, but just need to be transparent about it.)

Note, we now use the continuous insight measure for most analyses (except for some behavioral analyses for demonstration purposes and connectivity/graph analyses where we still use the binary median split insight measure). Nevertheless, for transparency reasons, we now provide bar plots with confidence intervals and individual subject points divided by high and low insight in Figure S5 in the Supplement. For a more detailed response, please see response to reviewer point #1).

9) More challenging: There are several important confounds.

The type of trial (i.e., the participants' response category) is highly confounded with time (2.75s for high-insight, 5.08s for low-insight). Authors try to handle this by including RT and accuracy [I still am not clear whether you're only analyzing CORRECT ID trials] as covariates of no interest in analyses, “if appropriate. I don't believe this is adequate for such an important confound. fMRI is hugely susceptible to time confounds (e.g, the shorter trials strongly conflate stimulus with response, moreso than the longer trials; and activity in a given region is integrated or summated over time- the longer you engage a process/cortical region, the stronger the BOLD

signal). **AND, the 'high-insight' trials are more likely to contain simple object recognition ("sudden" recognition immediately after presentation)**

We kindly refer the reviewer to our detailed response to point #3) regarding this issue.

10) Did we ever get the #/% correct for high- vs low-insight responses? The way the averages are presented make the actual number of trials in each condition cell unclear. As is often found, accuracy was better when people report high-insight, so you have even bigger differences in # of trials per response type. More importantly, if subsequent memory measures (behavioral and back-sorted fMRI trials) start with ALL responses, then there are different %ages of correctly identified trials in the high- vs low- insight categories. This is still true if you include only trials that were "correct" in the scanner, but that category is (proportionately or in absolute terms) more contaminated by 'lucky guesses' in the low-insight condition.

We agree with the reviewer's suggestion to include the absolute number of trials per condition. Please see detailed response to reviewer point #1). Even though we use the continuous insight scale now for most analyses, we have still included a bar plot now that shows all trials for all subjects per high/low insight condition in Figure S5 in the Supplement. We included only correct trials in all analyses. Regarding the "lucky guesses", please see response to response to reviewer point #2).

11) RC: Fig 3a: of course you get bigger change in category with more correct responses. Are you "saved" by showing roughly equivalent LARGE change in some other areas? Maybe helps.

The insight-related positive slope ("bigger change") in iLOC and pFusG cannot be explained just by more correct responses because the other areas show no or the opposite pattern in multivoxel pattern similarity change but include the same amount of correct responses as iLOC and pFusG. Please note, during the revision we have changed those analyses to continuous insight (including only correct trials solved after 1.5 and before 9.5 seconds).

[Figure Redacted]

Figure 3. VOTC regions show Representational Change (RC)" (p.10)

12) SUBSEQUENT MEMORY RESULTS

What I (and I think readers) want to know is:

How many trials (#, and % of all 120) were ‘correctly’ (with big caveat) recognized with high insight in scanner? Of these, how many were ‘remembered’ (by your criteria) 5 days later? Then, same two things for trials recognized with low-insight.

The absolute amount of trials per condition with and without memory are given now in Figure S5. The % of those trials that were correctly remembered with (correct) high and low insight in the scanner are shown in Figure 2-B (see below).

Please note again that, in response to feedback from reviewer #3, we present all results using the continuous insight measure. Consequently, detailed information on HI-I and LO-I is omitted. Nevertheless, here, we provide a description for you of how the paragraph would have appeared if we had maintained the categorical (median split) analysis of insight:

“Out of the 120 presented Mooney images, participants correctly solved and subsequently correctly recognized 36.0% ($SD=10\%$) divided into *HI-I* 24.4% ($SD=7\%$) and *LO-I* 11.6% ($SD=5\%$) trials. Out of all presented Mooney images, participants correctly solved and subsequently correctly recalled having solved them in 32.7% ($SD=11\%$) divided into *HI-I* 23% ($SD=7\%$) and *LO-I* 9% ($SD=5\%$) trials. Finally, out of all presented Mooney images, participants correctly solved and subsequently correctly identified them in 29.2% ($SD=11\%$) divided into *HI-I* 21% ($SD=7\%$) and *LO-I* 7% ($SD=5\%$) trials. Note, all values were chance adjusted.”

“

Figure S5. Amount of trials for each insight and insight memory condition

Note. Panel A: *HI*= high insight trials, *HI-cor*= correctly solved high insight trials, *HI-false* = incorrectly solved high insight trials, *LO* = low insight trials; *LO-cor* = correctly solved low insight trials; *LO-false*= incorrectly solved low insight trials; **Panel B:** *HI-r* = correctly solved high insight trials & later correctly recognized as having seen before; *LO-r* = correctly solved low insight trials & later correctly recognized as having seen before; *HI-rs*= correctly solved high insight trials & later correctly recognized as having solved before; *LO-rs* = correctly solved low insight trials & later correctly recognized as having solved before; *HI-m* = correctly solved and remembered high insight trials, *HI-fg* = correctly solved & forgotten insight trials, *LO-m* = correctly solved & later remembered low insight trials, *LO-fg*= correctly solved and later forgotten low insight trials.

“ (p.4f, Supplement)

“

Figure 2. Behavioral Insight Memory Effect.

Note. Panel A. overall proportion of all trials per condition; **Panel B.** Behavioural insight memory effect; *HI-r*= correctly solved high insight trials; *LO-r*= correctly solved low insight trials; estimates are marginal mean percentages to correctly recall the previously identified Mooney image in the respective condition. Error bars represent 95% confidence intervals. **Panel C.** relationship between continuous insight variable (3-12) and memory controlled for solution time; estimates are marginal mean percentages to correctly recall the previously identified Mooney image. Error bars represent 95% confidence intervals. **Panel D.** correlation (subject level) between anagrams and Mooney images.” (p.8f)

As is, I can't tell whether the #s stated (at bottom of p.6 into top of p. 7) are 68.7% of

ALL trials elicited “correct” responses, then 74% of those were remembered. I want to know # of images ID’d first time that are then recognized (by all your criteria) 5 days later.

We regret any confusion caused by our previous description. The paragraph has been revised in accordance with the reviewer's suggestions, aiming for a more unequivocal description.

“Out of the 120 presented Mooney images, participants correctly solved and subsequently correctly recognized 36.0% ($SD=10\%$) trials (chance adjusted). Out of all presented Mooney images, participants correctly solved and subsequently correctly recalled having solved them in 32.7% (chance adjusted, $SD=11\%$). Finally, out of all presented Mooney images, participants correctly solved and subsequently correctly identified them by name in 29.2% (chance adjusted, $SD=11\%$).” (p.7)

Please refer to our response above for a detailed breakdown of the memory results for HI-I and LO-I trials.

Trials solved via high vs low insight are remembered better

Again: are you starting with all responses, or just correct ones? For these analyses, you should only consider CORRECTLY identified trials. If not, several problems. One would be if you are including more Incorrect trials in one condition (probably the low-insight one).

For all memory measures and analyses, we have included only *correctly* identified trials (as for all other analyses now due to the revision).

13) METHODS

13a) Subjects (not sure why sub-header here is “Experimental model and...”)

Good info. I prefer to put the final N as close to the top as possible vs buried at end of the paragraph. Could even say the first sentence of “The final analyses included 31 participants” (age/gender)... “Initially, 38 participants were recruited.....”

Similarly, I’d prefer an early and explicit note on the number of trials (on average) analyzed per participant. Total of 120 image trials, but avg x correct w/ high-insight and x correct low-insight, etc.; make explicit the actual # of trials in each condition cell

Agreed, we changed the sub-header and the final sample information is now put to the beginning of the paragraph:

“Subject details

The final sample consisted of N=31 [age (in year): range = 19 – 33, 20 females: M= 25.13; 11 males: M= 26.27]. Initially, the 38 participants [age (in years): range = 20 – 34, 23 females: M= 26.6; 15 males: M= 25.7] were recruited via an online student platform in Berlin.” (p.27)

In addition, we incorporated a statement in the methodology section directing the reader to Figure S5 for the average number of trials analyzed per participant.:

“The average amount of remaining trials per condition is illustrated in Fig S5.” (p. 27)

13b) Assessing insight:

Probably best to reduce description and text emphasis on the multi-dimensional scoring of insight. You spend time/words convincing the audience it’s important, only to later say you have to drop it. (I’m completely in agreement with you – it would be important, IF it could be done – but it would take MANY MANY trials). Trivial side issue: “originally” insight in self-report (but not neuroimaging) paradigms was assessed on a scale of 1-5 (Bowden & Beeman, 2003). But, it becomes difficult to compare across participants that use scales differently, especially with few responses, so a binary form was adopted for most later neuroimaging and related studies. However, typical neuroimaging paradigms – and certainly here – elicit fewer correct responses. So, attempting to use complex scales is difficult. Hence, here they ultimately binarized the response anyway. So, perhaps, instead of emphasizing and justifying the multiple dimensions (each with 4 point scale), might reverse and say you ultimately used binary categorization, despite attempting to use complex scale. (The attempt to use the multiple scales does seem to be the right approach – IF there are enough usable trials and participants to make use of such data, but there aren’t enough solutions).

We appreciate the reviewer's detailed clarification and fully concur. If the manuscript had not undergone significant changes, we would have omitted the explanation of the multi-dimensional scoring of insight. However, in alignment with the suggestions from reviewer #3 and due to the limited number of non-remembered high insight trials, we have chosen to analyze most data (where feasible) using the more sensitive continuous insight measure instead of the binary one (also see responses to points #1 and #2). Notably, reviewer #3 also requested a separate analysis of all three insight dimensions to investigate which dimension is influencing the univariate effects in the amygdala and hippocampus. Conducting these analyses on a single trial level provided a sufficient number of trials for a meaningful differentiation between the three assessed insight dimensions, hence a separate analysis was possible. It's crucial to note that these analyses are exploratory in nature, clearly labelled as such. We report and briefly discuss the results as follows:

“Activity in amygdala and hippocampus during solution is predominantly associated with insight dimensions certainty and emotion

To explore the factors influencing the impact of insight in the hippocampus and amygdala, we delved into the three dimensions of insight. Note, our measurement model revealed correlations among all three dimensions—certainty, suddenness, and emotion. However, our focus was on identifying which dimension contributes unique variance beyond the shared variance among the three (note, these insight dimensions have only four values and are less

sensitive than the insight sum measure). For this, we repeated the above reported mixed effects single-trial analysis (standard space), substituting the insight measure with certainty, suddenness, and emotion. The results showed that for bilateral amygdala, both emotion ($t(2778.8)=2.02$, $p=.044$, $\beta=.05$) and certainty ($t(2959.2)=5.39$, $p<.001$, $\beta=.13$) individually explained variance in the BOLD signal. For the anterior hippocampus ($t(2715.4)=6.77$, $p<.001$, $\beta=.17$) and posterior hippocampus ($t(2753.7)=4.07$, $p<.001$, $\beta=.10$), only certainty explained individual variance beyond the influence of suddenness and emotion. Hence, the BOLD effects associated with the evaluative component of insight appear to be primarily influenced by the level of certainty regarding the correctness of the solution and (to a lesser degree) positive emotions.” (p. 14)

“For exploratory purposes, we investigated which insight dimension drives this insight*memory interaction in anterior hippocampus. For this, we repeated the above mentioned mixed effects single trials analysis, substituting the insight measure with either certainty, suddenness or emotion. Only the interaction between suddenness and memory ($t(2962.2)=2.69$, $p\text{-Bonferroni}<.01$, $\beta=.10$) predicted significant variance in anterior hippocampus while no interaction between memory and emotion ($p>.25$) or memory and certainty ($p>.09$) was found.” (p. 19)

We briefly discuss those results in the manuscript as follows:

Evaluative component of insight: amygdala and hippocampal activity

Our second finding was that insight - in particular its dimensions certainty and emotion - boosted activity in the amygdala and hippocampus (Jung-Beeman et al., 2004; Shen et al., 2017; Tik et al., 2018). In previous insight studies, amygdala activity has been attributed to positive emotions (Kizilirmak, Thuerich, et al., 2016; Ludmer et al., 2011), aligning with the broader role of this region in emotional arousal (Rouhani et al., 2023), and hippocampal activity to sudden solution awareness (Cabeza et al., 2020; Kizilirmak & Becker, 2024), in line with its role as a novelty or “mismatch” detector (Kumaran & Maguire, 2006; Nyberg, 2005). [...] In our task, as a result of RC in visual cortex, the originally meaningless stimulus becomes suddenly meaningful, yielding a prediction error, which triggers hippocampal activity and the storage of the salient event. This aligns with our finding that the insight memory effect in the anterior hippocampus is specifically driven by the suddenness dimension of insight.” (p. 22)

In summary, with the inclusion of these new analyses, the importance of the multi-dimensional scoring of insight in the description and text is once again too significant to disregard. However, we have shortened the descriptions of the different insight dimensions:

“Assessing insight. Originally, insight was only binarized as present or absent (Bowden & Jung-Beeman, 2003; Jung-Beeman et al., 2004; Kounios & Beeman, 2014). However, recent

studies showed that the overall subjective insight (AHA) experience is more continuous and can be decomposed into three different main dimensions: (1) positive emotional response or internal reward upon finding the solution, (2) experienced suddenness of the solution and (3) certainty about the solution (Danek & Wiley, 2017, 2020). To have better control over these three dimensions, we assessed them separately (see section: Mooney image paradigm), and then combined them into a compound insight measure by adding up the three ratings into a continuous scale (3-12). The individual dimensions were only employed for exploratory investigations to determine which ones influence univariate activity in amygdala and hippocampus during both insight and insight-related memory processes. Analysing the data with the continuous insight variable for connectivity and graph analyses was not applicable. Hence, for those analyses, the continuous insight measure was median split into high (HI-I) and low (LO-I) insight trials.” (p.28)

15) Mooney image paradigm

Good description. But, need to ultimately give # of trials and the distribution of response times within the conditions (correct hi- and correct low-insight).

In the original version of the manuscript, we had provided an image with the response time distribution (divided by correct high- and low-insight trials) in the figure S4 in the Supplement:

Assessing insight. Originally, insight was only binarized as present or absent (Bowden & Jung-Beeman, 2003; Jung-Beeman et al., 2004; Kounios & Beeman, 2014).

“Figure S4. Response time distribution divided by high and low insight solutions

Note. *LO-I* = low insight trials; *HI-I* = high insight trials. (all) = all solved trials; (correct) = only correctly solved trials. Dashed line = median response time for *HI-I* trials; Full line = median response time for *LO-I* trials. **Panel A:** Representation of the aggregated solution time distribution across each subject; **Panel B:** Illustration of the solution time distribution at the individual trial level.” (p.4, Supplement)

Additionally, we now provide the number of trials for correct high and low insight trials in figure S5 in the supplement:

“Figure S5. Amount of trials for each insight and insight memory condition

Note. **Panel A:** *HI*= high insight trials, *HI-cor*= correctly solved high insight trials, *HI-false* = incorrectly solved high insight trials, *LO* = low insight trials; *LO-cor* = correctly solved low insight trials; *LO-false*= incorrectly solved low insight trials; **Panel B:** *HI-r* = correctly solved high insight trials & later correctly recognized as having seen before; *LO-r* = correctly solved low insight trials & later correctly recognized as having seen before; *HI-rs*= correctly solved high insight trials & later correctly recalled as having solved before; *LO-rs* = correctly solved low insight trials & later correctly recalled as having solved before; *HI-m* = correctly solved and remembered high insight trials, *HI-fg* = correctly solved & forgotten insight trials, *LO-m* = correctly solved & later remembered low insight trials, *LO-fg*= correctly solved and later forgotten low insight trials. “

(p.4f, Supplement)

16) Subsequent memory test

Again, need a clear statement of the number of trials being analyzed. E.g., on average subs had [68?] correct identifications of Mooney images during fMRI, so we’re only looking at those. Of those, how many (#) were correctly recognized? And (above) problem with lucky guess “correct” trials during initial viewing in scanner.

We provide the number of trials analyzed per insight and memory condition in figure S5 (see

response to previous point) but, note, we now analyze the insight not per high/low insight group but on a continuous scale).

1st paragraph:

Minor writing: “2) the Mooney image was correctly recognized AS having been....”

We changed this typo in the manuscript: “2) the Mooney image was correctly recognized **as** having been identified in the scanner” (p.32)

2nd paragraph: need to explicitly state that remembered/forgotten trials were those “initially identified, but then subsequently not or not correctly remembered”. OR, were you assessing memory for all trials (including not identified – that would seem to violate the criteria laid out in 1st paragraph of this section).

We only assessed memory for initially correctly identified trials and changed the wording now according to the reviewer’s suggestions:

“Therefore, the identification of the Mooney object was considered as *remembered*, when it was **correctly** identified in the MR scanner *and* 1) the Mooney image was correctly recognized as seen in the scanner, 2) the Mooney image was correctly recognized **as** having been identified in the scanner and finally 3) the hidden Mooney object was correctly named in the Subsequent Memory Test (see section *Subsequent Memory Test*). [...] Forgotten trials were defined as trials that were **initially correctly identified, but then** subsequently not or not correctly remembered.” (p.32)

SUPPLEMENT / METHODS

Anatomical: are you saying individual subject anatomical-based ROIs for amygdala and hippocampus (very good, if so)

Yes, this exactly what we were trying to say, we specified this now: “**Bilateral** amygdala and the anterior and posterior hippocampus ROIs were first resampled into subject space (**i.e. creating individual subject anatomical-based ROIs**) using the ANTs function *antsApplyTransforms* (Avants et al., 2009).” (p.43)

“Correction: Where it applies, we USED FDR or Bonferroni corrected ALPHA LEVELS” (or just correction)

We changed the wording in the manuscript now as follows:

“For the resulting p-values, we used Bonferroni-corrected alpha levels because we did not have a hypothesis which ROI may exhibit insight-related changes in multivoxel pattern similarity.” (p. 36)

Minor writing issues and typos

There are a number of typos or minor grammatical errors, too many to list (but for the most part I do not think these affected my comprehension of their text).

In at least one place (Fig caption) they state that memory test was 7 days after initial fMRI session, most others state 5 days.

We thank the reviewer for noting this in Figure 1 and changed it accordingly: “Panel C illustrates the subsequent memory test five days after the solving phase in the MRI” (p.4)

“To investigate these issues, we scanned participants with fMRI while they identified Mooney images, and tested their memory for the solutions five days later.” (p.21)

Writing issue:

Representational change terminology. I’m OK with using this term as long as there’s at least a brief note of how it may differ in this paradigm versus, say, classic insight problems (or even simple word puzzles). But then several different versions are used (“conceptual change” Etc) often followed by the RC acronym, which is just a bit confusing, especially to people less familiar with the area. Even if it gets repetitious, consistency would be preferred.

Please see response to point #4) where we describe how verbal and visual restructuring is different (including the reference in the manuscript). In the introduction, we define representational change (RC) as internal (conceptual OR perceptual) representations of the solution that are reorganised and integrated into a coherent whole. Hence, representational change can be either conceptual or perceptual. During the paper, we often talk about conceptual change because the models we use to test this are primarily conceptual models (penultimate layer of AlexNet and Word2Vec). However, we agree with the reviewer, the terminology should be as consistent as possible. We tried to stay with the terminology of representational change (RC) as much as possible now:

“In sum, insight during problem solving likely reflects stronger multivariate pattern changes associated with enhanced solution-relevant representations. This increased RC (or conceptual update) is efficiently integrated into the solution process [...]” (p.3)

“Although these results provide a strong candidate for a neural mechanism of RC, i.e. the cognitive component of insight, they cannot explain the evaluative component of insight,” (p.12)

“Next, we examined the assumption that the stronger conceptual RC is efficiently integrated into the solution process.” (p.15)

“Hence, in the context of insight, the hippocampus may detect novel, unexpected internal events (e.g., enhanced conceptual RC of solution object in VOTC) by eliciting a prediction error signal (surprise) and subsequently encodes these events, proportional to the magnitude of this conceptual RC.” (p.24)

“We presented consistent neural evidence indicating that insight in problem solving likely reflects an enhanced conceptual RC of solution representations efficiently integrated into a coherent whole. This RC generates solution awareness alongside an emotional and suddenness/novelty response in the amygdala and hippocampus, boosting solution encoding and subsequent memory.” (p.25)

Figure 3: Label the cortical areas in Figure (could just note the colors per area in the caption, but in-figure labels would be most helpful)

We have now included in-figure labels for the ROIs:

[Figure Redacted]

Figure 3. VOTC regions show Representational Change (RC)” (p.10)

References, other relevant work

Reference for earlier effort to find behavioral evidence of restructuring involved in insight , using graph-theoretic evaluation of semantic relatedness judgments: Durso, Rea, & Dayton (1994). Graph-theoretic confirmation of restructuring during insight. *Psychological Science*, 5, 94-98.

Sampling of References regarding object recognition /visual closure / visual Aha! that may prove helpful. I strongly recommend citing & discussing the first (Bar 2001), but

the others are also potentially related to interpretation (for the most part compatible with current interpretation, but viewed from a different angle, so to speak).

*Bar, M. et al. (2001). Cortical mechanisms specific to explicit object recognition. *Neuron*, 29, 529-535.

Bar, M. et al., (2006). Top-down facilitation of visual recognition. *PNAS*, 103, 449-454

Doniger et al., (2000). Activation time course of ventral visual stream object recognition areas: High density electrical mapping of perceptual closure processes. *J. Cog Neuroscience*, 12, 615-621.

Doniger et al. (2021). Visual perceptual learning in human object recognition areas: A repetition riming study using high-density electrical mapping. *NeuroImage*, 13, 305-313.

Sehatpour, P. et al., Spatiotemporal dynamics of human object recognition processing: An integrated high-density electrical mapping and functional imaging study of "closure" processes. *NeuroImage*, 29, 605-618

We thank the reviewer for those relevant literature suggestions. As the reviewer will see, we incorporated almost all of those references into the manuscript (see response to point #4 further above).

Reference related to continuous vs binary insight ratings (does not need to be cited – I never ‘require’ anyone to cite any of my papers – but it’s relevant in that we, too, started by trying to use the continuous nature of strength of “aha” ratings, but found it too difficult to work with statistically so switched to binary (insight/non or analytic/insight, etc; sometimes offering “other” as an option, but that also creates complications) in later studies.

Bowden, E.M. & Jung-Beeman, M. (2003). Aha! Insight experience correlates with solution activation in the right hemisphere. *Psychonomic Bulletin & Review*, 10, 730-737. PMID: 14620371

As we have mentioned, we now switched to mostly using insight as a continuous measure. Nevertheless, we have now additionally included this citation into the manuscript:

“**Assessing insight**. Originally, insight was only binarized as present or absent (Bowden & Jung-Beeman, 2003; Jung-Beeman et al., 2004; Kounios & Beeman, 2014).” (p.28)

Response to Reviewer #3:

The paper reports on an exciting study investigating the synergy between creativity and memory, with a particular focus on insight. The authors report on “sudden representational changes” in visual regions, insight-related activity in MTL regions, and the finding that these effects are predictive of later memory for the solutions. Most studies examining creativity directly struggle with the fact that insight is unpredictable and infrequent, rendering it difficult to examine with neuroimaging. However, here the authors use a task that more reliably provides an experience of insight, enabling them to examine the neural processes underlying the phenomenon underlying insight, albeit outside of the typical creative context. Overall the manuscript is well written, the analyses highly novel (especially the analyses on RC and representational strength), and the results for insight particularly compelling (less so for memory).

(1) General comment:

It should be noted somewhere that while insight is most certainly a phenomenon associated with a rapid shift, in the context of creative problem solving the process leading up to this shift might not be as discrete and sudden. For example, if one has thought about the problem for days or months, likely with slower changes in semantic networks, etc., insight might not necessarily reflect “rapid learning and memory formation”.

We agree with the reviewer, we did not investigate the timing of the encoding process itself, therefore we cannot say anything about how quickly a content is encoded in memory. What we mean by “rapid learning and memory formation” only refers to the fact that subjects’ memory is boosted after one single exposure/encoding instance with the problem/solution when they solved it with increasing memory. We have therefore, erased the word “rapid” in this context in memory:

“The study of insight can shed light on mechanisms of learning and memory formation, and has practical implications in educational settings, as insight-driven learning experiences can be effective in improving knowledge retention (Gruber et al., 2014; Ludmer et al., 2011).”
(p.2)

Concerning the transition process leading up to this shift (RC), it remains uncertain whether it occurs suddenly or continuously, primarily due to the current absence of neural evidence. In the insight literature, representational change (RC) is frequently characterised as sudden but likely because RC is not (if at all) measured independently of the AHA! experience which is perceived as sudden/surprising etc. However, we fully concur with the reviewer’s perspective that the process leading to this shift is likely not discrete. We have incorporated a relevant resource (Bar et al., 2001), which offers remote evidence suggesting that this process may be more continuous than discrete, and have now included this information in the manuscript as follows:

“Note, there is limited knowledge whether the process that leads up to RC, e.g. the transition from meaningless to meaningful objects, occurs gradually or discretely and is only perceived as sudden. However, previous research suggests a linear correlation between BOLD activity in the fusiform gyrus as well as occipital temporal sulcus and awareness of object identity (Bar et al., 2001), indicating a more continuous process.” (p.21)

Moreover, we revised the terminology from 'sudden' representational change to 'stronger'/'increased'/'enhanced' representational change. This adjustment reflects the fact that we did not measure the actual 'suddenness' of those changes but rather observed an increased magnitude in the change:

“In sum, insight during problem solving likely reflects **stronger** multivariate pattern changes associated with enhanced solution-relevant representations (RC). This **increased** conceptual change (RC) is efficiently integrated into the solution process leading to awareness of the solution accompanied by an emotional and suddenness/surprise response in hippocampus and amygdala resulting in enhanced encoding of the solution and better subsequent memory. [...]

Next, we examined the assumption that the **stronger** conceptual update (RC) is efficiently integrated into the solution process. [...]

Hence, in the context of insight, the hippocampus may detect novel, unexpected internal events (e.g., **enhanced** conceptual updates of solution object in VOTC) by eliciting a prediction error signal (surprise) and subsequently encodes these events, proportional to the magnitude of this conceptual update. [...]

We presented consistent neural evidence indicating that insight in problem solving likely reflects **an enhanced** conceptual update of solution representations efficiently integrated into a coherent whole.” (p.3ff)

(2) Behavioral results:

2a) It is notable that almost half of the incorrectly identified images were rated as high insight - does this present a concern with using insight ratings as the main variable of interest in this study?

We have adopted a methodology that involves exclusively utilizing correctly solved high insight trials for all analyses. Generally, we do not perceive it as a significant concern that incorrectly solved trials were still categorized as high insight, as there can be several reasons for this: First, subjects may have had a correct alternative solution (triggering an AHA! experience) which however did not align with the presented response categories. Second, there is a possibility that participants were genuinely convinced that they have suddenly and correctly identified object X in ambiguous blobs (even if those blobs did not represent this object X), potentially triggering an AHA! Experience. This phenomenon of so-called 'false' insights, albeit not happening too often, has been documented previously (Danek & Wiley, 2017). However, due to the inherent limitation of not being able to further explore the nature of these 'false' insights using the four different response categories, we opted to focus solely on correct trials, as their interpretation is more straightforward.

Nevertheless, we acknowledge the reviewer's valid point that relying solely on the subjective AHA! experience may not be a sufficient criterion for identifying insight trials (refer also to Danek & Wiley, 2017). Therefore we also tried to identify other biologically plausible mechanisms of insight, such as representational change (see also reviewer #2 point 4).

Danek, A. H., & Wiley, J. (2017). What about false insights? Deconstructing the Aha! experience along its multiple dimensions for correct and incorrect solutions separately. *Frontiers in psychology*, 7, 2077.

2b) Selecting the correct category is not a precise measure of accuracy; are analyses chance-adjusted? This also means that, at test, an incorrect yet category-consistent answer could reflect either a memory failure OR not having generated the correct solution to begin with. Is there any data that can allay this concern?

Concerning the accuracy metric, we agree with the reviewer that we lack certainty about the specific object identified by the participant during problem-solving.

We assume that the proportion of falsely classified correct responses should be less than pure chance of 25% because we emphasised a conservative response criterion in the instruction, i.e. that participants should only press the solution button when they were highly confident they had identified a solution. Furthermore, we stated explicitly that we had intentionally created very difficult Mooney images and would therefore not expect that they would find the solution for each of the images, hence this experiment was not a forced-choice task as the subjects had the option not to press the solution button when they did not find a solution.

However, to get a more realistic estimate of the actual proportion of falsely classified correct responses in the Mooney identification task of the fMRI sample, we conducted a preregistered behavioral control experiment (note, the control experiment was mainly conducted as a response to the reviewer 2's comment #3d). This behavioral experiment was identical to the experiment of the fMRI sample with the difference that in this control experiment, instead of choosing one out of four response categories, participants expressed their solutions by typing the Mooney object identity (refer to Supplementary Material for further details), eliminating any ambiguity in their response. The proportion of falsely classified correct responses in the Mooney identification task of the fMRI sample was estimated as follows: Each of the subjects' solution words from the Mooney identification task in the control experiment was compared with the four possible response categories to determine whether it matched one of the response options that the subjects saw for each Mooney image in the Mooney identification task in the fMRI experiment (note, the four response categories for each Mooney image in the Mooney identification task were the same for every subject in the fMRI sample). The incorrect solution words from the Mooney identification task in the control experiment were then categorized according to whether they a) fell into the correct response category (i.e. falsely classified as correct response), b) into one of the other response categories (i.e. correctly classified as incorrect response) or c) into none of the specified response categories (i.e. here we assume that subjects randomly chose one out of the four response categories and therefore there is a 25% chance that their

response is falsely being classified as correct response). The proportion of falsely classified correct responses in the control experiment was calculated from this categorization (amount of falsely classified correct responses divided by the sum of correctly and falsely classified correct responses). This proportion amounted to 18.7% ($SD=10\%$). The amount of correctly identified and subsequently remembered Mooney images in the fMRI experiment was finally adjusted by this proportion. Importantly, those 18.7% falsely classified correct responses divided into 7% random guessing and 11.7% aligned with the correct response category.

For a detailed description of the control experiment and its results, please see the supplementary material in the manuscript. The accuracy adjustment procedure for the Mooney identification and subsequent memory task in the fMRI sample is now explained in the manuscript as follows:

“Accuracy Adjustment. As participants selected one of four response categories in the Mooney identification task, some incorrect responses were classified as correct just by chance. To assess the proportion of falsely classified correct responses, a behavioral control experiment was conducted. The Mooney identification task in this control experiment mirrored the task in the fMRI sample, except that participants typed the solution word instead of choosing one out of four response categories, eliminating ambiguity. The proportion of falsely classified correct responses in the fMRI experiment was estimated by comparing the solution words from the Mooney identification task in the control experiment with the response categories of the same task in the fMRI sample. The incorrect solution words from the control experiment were then categorized based on whether they a) aligned with the correct response category (i.e., falsely classified as a correct response), b) aligned with one of the other response categories (i.e., correctly classified as an incorrect response), or c) did not align with any specified response category (i.e., random guess and therefore 25% chance that response is falsely classified as a correct response). The proportion of falsely classified correct responses of all correct responses in the control experiment was calculated from this categorization and amounted to 18.7% ($SD=10\%$, 7% came from random guessing and 11.7% aligned with the correct response category). The amount of correctly identified and subsequently remembered Mooney images in the fMRI experiment was finally adjusted by this proportion.” (p. 31f)

The behavioral results with the accuracy adjustment for the Mooney identification task in the fMRI sample is now described the manuscript as follows:

“On average, participants indicated a solution for 68.4% ($SD=16\%$) of all presented Mooney images. The solution, i.e. the subsequently selected category of the object, was correct in 43.0% (chance adjusted, $SD=13\%$) of all cases. [...]

Subsequent memory of each Mooney object identification was measured five days later. Participants recognized 67.0% ($SD=15\%$) of the 120 presented Mooney images, regardless of whether they had correctly identified them, with a false alarm rate of 19.3% ($SD=14\%$). [...]

Out of the 120 presented Mooney images, participants correctly solved and subsequently correctly recognized 36.0% (chance adjusted, $SD=10\%$). Out of all presented Mooney images, participants correctly solved and subsequently correctly recalled having solved them in 32.7% (chance adjusted, $SD=11\%$). Finally, out of all presented Mooney images, participants correctly solved and subsequently correctly identified them by name in 29.2% (chance adjusted, $SD=11\%$)” (p. 6f)

Finally, we acknowledge and address this limitation of the imprecise accuracy measure in the following manner in the discussion section (see more detailed response to whole limitation section further below):

“*Limitations*

Three key limitations should be addressed. Firstly, the accuracy metric is not exact as participants could not enter specific names of identified objects during problem-solving due to technical constraints in the scanner. Nevertheless, we assessed the likelihood of incorrectly classified correct responses through a control experiment (18.7%, refer to Supplementary Material) and then adjusted the accuracy and subsequent memory measures accordingly for this proportion. [...]

(3) fMRI analyses – overall comments:

3a) A number of analyses compare high vs low insight trials; showing that the effects are not at all evident for the no-solution trials would be even more compelling. e.g., showing that RC is not different from zero for no-solution trials would demonstrate that the shift in representation is specific to trials on which a solution was generated (and then modulated by degree of insight).

We agree with the reviewer that demonstrating a comparable effect for no-solution trials would enhance the persuasiveness of our findings. Unfortunately, achieving comparability in this context poses inherent challenges for two main reasons:

- 1) **Cognitive Reasoning:** Trials without a solution lack a button press indicating resolution. In these instances, participants pressed an alternate button after observing the fixation cross at the conclusion of the trials. Comparing this "non-event" with the two other events involving solutions becomes intricate due to the considerable time lag (10 seconds as opposed to approximately 3.5 seconds for the solved trials). This discrepancy in timing may introduce potential time-on-task effects in the fMRI signal, adding complexity to meaningful comparisons. Even if one were to artificially define an event in no-solution trials at ~3.5sec, cognitive processes are not

directly comparable between solution and no-solution trials. In the latter, subjects are actively exploring and searching for a solution, engaging working memory resources, while in the former (solution trials), the search process has already concluded.

- 2) **Methodological Rationale:** In computing the brain Representational Similarity Matrix (RSM) for the second measure of representational change (RSA) during the solution button press, we specifically correlated the multivoxel pattern between only solved trials to minimize data noise. Correlating all trials during the solution event, regardless of whether they were solved or not, would introduce numerous trials reflecting mere noise, diminishing the power of this RSA measure. This design choice, however, prevents a meaningful comparison between RSA measures for solved and unsolved trials, as both types of trials were not correlated with the same subset of trials but only within their respective event types (solved OR unsolved).

3b) Analyses using a categorical insight variable, even though the continuous variable is described as more sensitive, with some memory effects only evident using this. Should it be used throughout?

We fully agree with the reviewer's perspective that utilising the continuous variable serves as a superior measure for detecting subtle differences in our brain measures due to its higher sensitivity. Furthermore, we decided to now include only correct trials that were solved within the timeframe of 1.5 to 9.5 seconds, aimed at including only true insight trials (not simple object recognition trials) and eliminating potential confounds related to stimulus onset/offset. This, however, reduces the overall amount of trials. For both reasons, we have adopted the continuous insight measure for nearly all analyses. The exceptions are instances where discrete measures are employed for demonstration purposes (behavioral data analysis) and for connectivity analyses (note, functional connectivity and specifically graph theoretic measures are more aptly calculated between non-continuous conditions in CONN). Due to space limitations, we direct the reviewer to the results section. Importantly, the overall results remain largely consistent with the previous version, except in the case of the memory measure with AlexNet, where we now observe insight and memory effects but no insight*memory interaction anymore (note, this measure had been weak before). We discuss this new result in the discussion as follows:

“Note, the RSA analyses using AlexNet did not show a significant correlation with insight-related better memory, whereas Word2Vec did. This difference may be attributed either to issues of statistical power or to the observation that semantic representations, linked to better subsequent memory (Craik & Lockhart, 1972), may be more effectively captured by the Word2Vec model than by the vision-based AlexNet model.” (p.24)

(4) Evaluative aspect of insight:

4a) The 3 ratings are summed to form an “insight” variable. However, it would be useful to know what was driving the insight effect. For instance, in the discussion the authors suggest that effects in the amygdala reflected emotion, and in anterior hippocampus both novelty and encoding. The data exists to answer all of these questions, including to determine whether anterior hippocampal effects on subsequent memory hold when controlling for the effect of novelty (e.g., BOLD signal could be residualized for the effect of novelty prior to running subsequent memory analyses, or using a parametric modulation approach).

We thank the reviewer for this suggestion. Given the measurement model of insight (see Supplement) we already know that all three dimensions share a substantial amount of variance. However, an interesting question we can ask is whether any of those three variables explains additional unique variance in the BOLD signal over and beyond the other two ones. We have now examined the individual insight dimensions for amygdala and hippocampus and described the results as follows:

“Activity in amygdala and hippocampus during solution is predominantly associated with insight dimensions certainty and emotion

To explore the factors influencing the impact of insight in the hippocampus and amygdala, we delved into the three dimensions of insight. Note, our measurement model revealed correlations among all three dimensions—certainty, suddenness, and emotion. However, our focus was on identifying which dimension contributes unique variance beyond the shared variance among the three (note, these insight dimensions have only four values and are less sensitive than the insight sum measure). For this, we repeated the above reported mixed effects single-trial analysis (standard space), substituting the insight measure with certainty, suddenness, and emotion. The results showed that for bilateral amygdala, both emotion ($t(2778.8)=2.02$, $p=.044$, $\beta=.05$) and certainty ($t(2959.2)=5.39$, $p<.001$, $\beta=.13$) individually explained variance in the BOLD signal. For the anterior hippocampus ($t(2715.4)=6.77$, $p<.001$, $\beta=.17$) and posterior hippocampus ($t(2753.7)=4.07$, $p<.001$, $\beta=.10$), only certainty explained individual variance beyond the influence of suddenness and emotion. Hence, the BOLD effects associated with the evaluative component of insight appear to be primarily influenced by the level of certainty regarding the correctness of the solution and (to a lesser degree) positive emotions.” (p.14)

To determine the individual effects of the three dimensions on subsequent memory in anterior hippocampus, we were not able to include all three variables into one model together with their memory interaction term due to too high multicollinearity (VIFs > 20). For this reason, we repeated the insight*memory analysis three times exchanging suddenness, certainty or emotion with insight and Bonferroni-corrected the p-value:

“For exploratory purposes, we investigated which insight dimension drives this insight*memory interaction in anterior hippocampus. For this, we repeated the above mentioned mixed effects analysis, substituting the insight measure with either certainty, suddenness or emotion. Only the interaction between suddenness and memory ($t(2962.2)=2.69$, $p\text{-Bonferroni}<.05$, $\beta=.10$) predicted significant variance in anterior hippocampus while no interaction between memory and emotion ($p>.25$) or memory and certainty ($p>.09$) was found.” (p.19f)

Those results were incorporated into the amygdala and hippocampus discussion in the discussion section:

“Our second finding was that insight - in particular its dimensions certainty and emotion - boosted activity in the amygdala and hippocampus (Jung-Beeman et al., 2004; Shen et al., 2017; Tik et al., 2018). In previous insight studies, amygdala activity has been attributed to positive emotions (Kizilirmak, Thuerich, et al., 2016; Ludmer et al., 2011), aligning with the broader role of this region in emotional arousal (Rouhani et al., 2023), and hippocampal activity to sudden solution awareness (Cabeza et al., 2020; Kizilirmak & Becker, 2023), in line with its role as a novelty or “mismatch” detector (Kumaran & Maguire, 2006; Nyberg, 2005). [...] In our task, as a result of RC in visual cortex, the originally meaningless stimulus becomes suddenly meaningful, yielding a prediction error, which triggers hippocampal activity and the storage of the salient event. This aligns with our finding that the insight memory effect in the anterior hippocampus is specifically driven by the suddenness dimension of insight.” (p.22)

4b) Relatedly, the authors state that “Using covariates to remove unwanted variance is not feasible in the SPM GLM framework” (p.12-13); however this can be achieved using parametric modulation analysis with insight as the first parameter (= variable of interest) and response time and accuracy as subsequent parameters (= nuisance covariates).

We agree with the reviewer. Because we have now used the continuous insight measure it is indeed possible to reduce the shared variance between solution time and insight from the insight parametric modulator (note, this is not possible when having two different conditions as a parametric modulator does not extract unwanted variance from the onset regressor). We reanalyzed the data and described them in the manuscript as follows. Note, because we were now able to directly correct for solution time in SPM via a second parametric modulator for solution time, we skipped one of the three univariate analysis for amygdala and hippocampus due to redundancy and now only report differences in BOLD activity as a function of 1) insight in form of a parametric modulation in subject space and 2) insight via the single trial analysis in standard space.

“Hypothesis 2: Amygdala and hippocampus show insight-related activity

To examine our second hypothesis that the evaluative component of insight is mediated by the amygdala and anterior/posterior hippocampus, we estimated whether bilateral amygdala and hippocampal activity during correct Mooney object identification would vary depending on the strength of insight. Note, we used univariate activity because both brain regions are unlikely to represent visual information like VOTC. We conducted the analyses with the data from the single-trial analyses containing a beta estimate of univariate amygdala, anterior and posterior hippocampal activity (ROI) for the solution button press at each trial. We compared two models of correctly solved trials for insight (continuous) as dependent variable and response time and run order (1-4) as covariates of no interest as well as a random subject and item intercepts.

Activity in amygdala and hippocampus is associated with insight

As illustrated by Fig.4, there was a main effect for condition ($Chi^2(1)=47.56, p<.001, \beta=.09$) implying a positive relationship between brain activity and insight over all three ROIs. There was also a condition*ROI interaction effect ($Chi^2(2)=26.91, p<.001$) implying differences between the ROIs in this condition. Post Hoc analyses revealed that activity in amygdala activity ($z=7.54, p<.001, \beta=.17$) and anterior ($z=5.57, p<.001, \beta=.13$) but not posterior ($z=1.11, p=.27, \beta=.02$) hippocampus significantly predicted insight (see Fig.4). A control analysis was performed adjusting solution time between HI-I and LO-I trials (i.e. removing fast HI-I and slow LO-I trials until the time taken to solve both types of trials was no longer significantly different [$p>.20$]) and repeating the above mentioned analysis. The results did not change significantly (see Supplement).

Editorial Note: The brain image in Figure 4 of this Peer Review file was generated using CONN⁵³ [RRID:SCR_009550]

Figure 4. Amygdala and hippocampal activity during solution are associated with insight.

Note. Values = estimated marginal means \pm SEM from single trials analysis in standard space. * represents p -value $< .001$. Correlation between the strength of insight and amygdala (Amy), anterior (aHC) and posterior (pHC) hippocampal activity.

Activity in amygdala and hippocampus during solution is predominantly associated with insight - control analysis in subject space

To exclude potential normalisation artefacts due to the small volume of the amygdala and hippocampal ROIs, we additionally conducted another ROI analysis in subject space. Univariate activity during solution, i.e. onset of the button press, was parametrically modulated by insight and solution time and estimated using GLMs as implemented in SPM (Ashburner, 2012). To test whether amygdala and hippocampal activities parametrically vary depending on the strength of insight, we computed a linear mixed model including a random subject intercept and controlling for the run order (1-4) and the respective ROI.

Bilateral amygdalar and hippocampal activity are significantly parametrically modulated by insight during (correct) solution ($t(159.9)=3.93$, $p<.001$) but there are also differences between the ROIs ($Chi^2(2)=28.59$, $p<.001$). However, Post hoc analyses revealed that amygdala activity ($t(121.7)=5.21$, $p<.001$, $\beta=0.18$), anterior hippocampus ($t(121.9)=3.24$, $p<.005$, $\beta=0.09$) as well as posterior hippocampus ($t(120.4)=4.37$ $p<.001$, $\beta=0.14$) were all significantly parametrically modulated by insight." (p.12ff)

(5) Functional connectivity (FC) analyses:

Some of the analyses seemed redundant. The authors start with all ROIs, then re-run on those ROIs with strongest effects, and end up with the subset of regions identified in other analyses (i.e., pFusG, iLOC, amygdala & anterior HC). This also meant it's not always clear what analyses/results reflect the full network or the subnetwork (e.g., graph theory analyses; plot Fig 5B). The FC analyses are described as defining a "solution network" but this isn't accurate given the full 10 second trial is used (not just the time of insight/solution). For this reason, it is also not surprising that the FC effects are not predictive of memory – this should be mentioned in the discussion.

We agree with the reviewer that certain analyses are redundant, and as a result, we have omitted all analyses conducted solely for a subset of the 'solution network.' Importantly, by 'solution network,' we intend to convey not the temporal aspect of the solution, but rather the brain regions likely implicated in processing the solution object, encompassing all VOTC regions, including the Amygdala and HC ("For this, we created a *solution network* consisting of all bilateral VOTC ROIs (aITG, mITG, pITG, aFusG, pFusG, iLOC) relevant for object identification (DiCarlo et al., 2012). The *solution network* (18 ROIs in total) further included amygdala and anterior as well as posterior hippocampus because they are likely involved in the evaluation of the solution content." p.15). Consequently, all analyses reported for functional connectivity and graph measures pertain to a singular (solution) network, comprising the same amount of above mentioned ROIs.

To ensure a sufficient number of data points for a reliable connectivity analysis, we used the entire 10-second time window, recognizing the limited temporal resolution of MRI. Our rationale for choosing this extended time window also stems from the assumption that both the processing and encoding of the solution may require more time than the brief insight moment. However, the encoding time for the solution object remains unknown, and it could indeed be considerably shorter than the 10-second time window. Hence, we thank the reviewer for this valuable suggestion and have mentioned this matter in the discussion section:

"Note however, no FC evidence predicts subsequent memory, possibly due to limited power, independent contributions of both regions to insight-related memory enhancement **or the possibility that the chosen time window (10 seconds) is too extensive to capture the transient encoding advantage associated with insight.**"(p.25)

(6) Memory

Even though significant, the memory effects are not all that striking. There are effects of insight evident for solutions that were later forgotten, and relative to that, there really is only a small increase for those that were remembered. This should at least be mentioned in the discussion.

The behavioral findings demonstrate strong robustness, with a Mooney image having a ~75% chance of being remembered when previously solved with high insight, compared to less than 50% with low insight. This result has been replicated in a preregistered behavioral control experiment (refer to Supplement and Figure S7). However, regarding the brain results, we acknowledge the reviewer's point that while significant, they may not exhibit the same strength. This is partly attributed to our use of an insight*memory interaction as a stringent measure of successful memory—a method unprecedented in demonstrating a clear insight memory advantage in brain data. The choice of this strict interaction criterion reduces the power to detect effects, given the high correlation between insight and memory. We now mention this issue in the manuscript as follows:

“Note, however, the high variability between insight and memory (few forgotten trials with higher insight), led to generally low statistical power to detect an insight*memory interaction.” (p.24)

“Note, the RSA analyses using AlexNet did not show a significant correlation with insight-related better memory, whereas Word2Vec did. This difference may be attributed either to issues of statistical power [...]”. (p.24)

Minor comments

- **Please mention the final N of the sample in the introduction or beginning of results.**

“Behavioural results

A final sample of N=31 subjects was analysed.” (p.5)

- **Fig 4 and Fig 6: It is extremely difficult to see the different ROIs; using color might help.**

We agree with the reviewer and have colored the ROIs accordingly. We hope that the ROIs are better distinguishable now::

Editorial Note: The brain image in Figure 4 and 6 of this Peer Review file was generated using CONN⁵³ [RRID:SCR_009550]

Figure 4. Amygdala and hippocampal activity during solution are associated with insight.

Note. Values = estimated marginal means \pm SEM from single trials analysis in standard space. * represents p -value < .001. Correlation between the strength of insight and amygdala (Amy), anterior (aHC) and posterior (pHC) hippocampal activity. (p.13)

Figure 6. Representational change in VOTC and hippocampal activity during solution are associated with insight-related better subsequent memory. (p. 20)

- **Fig 5: define “NW”**

“*Note.* NW = network; VOTC-RC = Ventral occipital temporal cortex regions that elicit representational change; all network measures stem from contrast: $HI-I > LO-I$. $HI-I$ = trials solved with high accompanied insight; $LO-I$ = trials solved with low accompanied insight.” (p.17)

- **VOTC ROIs are always described as bilateral. Is the same true for MTL ROIs?**

Yes, all MTL ROIs are bilateral as well: “The left and right ROI masks for amygdala and hippocampus were extracted from the FSL Harvard-Oxford (HO) Atlas (Desikan et al., 2006).” (p.41)

We clarify this more in the manuscript now:

“[...] we estimated whether bilateral amygdala and hippocampal activity during correct Mooney object identification would vary depending on the strength of insight.” (p.12)

“Bilateral amygdalar and hippocampal activity are significantly parametrically modulated by insight during (correct) solution [...]” (p.14)

“Statistical analyses were conducted with the data from the single-trial analyses containing a beta estimate of univariate bilateral amygdala and anterior as well as posterior hippocampus activity for the solution button press at each trial [...]” (p. 41)

“Bilateral amygdala and the anterior and posterior hippocampus ROIs were first resampled into subject space [...]” (p.43)

- **p. 4: Unlike positive emotion and novelty/suddenness, there’s no explanation in the introduction as to why certainty ratings are included as a measure of insight measure**

In the introduction we originally wrote the following: “The evaluative component (or *AHA! experience*), which is characterized by positive emotions and the conviction that the solution is correct and arrived suddenly (Danek & Wiley, 2017; Kizilirmak & Becker, 2023)”.

The “conviction that the solution is correct” is what we mean by certainty. Prior studies (Danek & Wiley, 2017) have shown that certainty is part of the AHA experience and this has been replicated several times (Danek & Wiley, 2020; Becker, Wang, Cabeza, 2023) but see also the latent measurement model in the current manuscript in the Supplement where the latent “AHA” factor loads equally high onto certainty [$\lambda=.6$] as onto emotion or suddenness.

To make the wording more clear, we changed the manuscript as follows:

“The evaluative component (or *AHA! experience*), which is characterized by positive emotions and the conviction that the solution arrived suddenly and is certainly correct (Danek & Wiley, 2017; Kizilirmak & Becker, 2023)” (p.3).

Becker, M., Wang, X., & Cabeza, R. (2023). Surprise!—Clarifying the link between insight and prediction error. *Preprint*. *Open Science Framework*. <https://doi.org/10.31219/osf.io/tkhn5>.

Danek, A. H., & Wiley, J. (2017). What about false insights? Deconstructing the Aha! experience along its multiple dimensions for correct and incorrect solutions separately. *Frontiers in psychology*, 7, 2077.

Danek, A. H., & Wiley, J. (2020). What causes the insight memory advantage?. *Cognition*, 205, 104411.

- **p. 5: “Third, solution-relevant information is more efficiently integrated in a “solution network” consisting of VOTC, amygdala, and hippocampus during insight”. This is poorly worded as the information isn’t integrated into the network, but by the network**

“Third, *solution-relevant information is more efficiently integrated by a “solution network” consisting of VOTC, amygdala, and hippocampus during insight.*” (p. 5)

- **p. 35: If German as first language was an eligibility criteria, was the study conducted in German?**

Yes, the study was conducted in Germany and therefore also in German, we have mentioned this additionally in the manuscript now:

“**The study was conducted in German.** Inclusion criteria were German as mother language [...]” (p. 27)

- **p. 35: “andtoo”**

This typo was changed in the manuscript: “We excluded 7 subjects from data analysis due to technical issues at the scanner (N=1), excessive head movement in the scanner (N=3), pathological findings in brain anatomy (N=2) **and too** low performance (0% correct recall) in the subsequent memory test (N=1).” (p. 27)

Reviewer #4 (Remarks to the Author):

This study is interesting and well-conceived, and is potentially an important contribution to the neuroscience of insight. However, it has a number of issues that need to be addressed. The following are in no particular order. (For the record, I am not an fMRI specialist.)

1) The high error rate in the problem-solving part of the study is a concern. (The paper should describe how the data were “chance adjusted”?) This, together with the fact that subjects only had to identify the categories of the objects rather than the objects’ identities is also of concern.

We agree with the reviewer that the error rate in the problem-solving part of the study is relatively high. We understood that the reviewer views this as a potential concern due to the resulting lower trial count. We address this concern in detail below in response #7 of this response letter.

Regarding chance adjustment: Please note that in the Methods section, we describe how the data were chance adjusted:

“Accuracy Adjustment. As participants selected one of four response categories in the Mooney identification task, some incorrect responses were classified as correct just by chance. To assess the proportion of falsely classified correct responses, a behavioral control experiment was conducted. The Mooney identification task in this control experiment mirrored the task in the fMRI sample, except that participants typed the solution word instead of choosing one out of four response categories, eliminating ambiguity. The proportion of falsely classified correct responses in the fMRI experiment was estimated by comparing the solution words from the Mooney identification task in the control experiment with the response categories of the same task in the fMRI sample. The incorrect solution words from the control experiment were then categorized based on whether they a) aligned with the correct response category (i.e., falsely classified as a correct response), b) aligned with one of the other response categories (i.e., correctly classified as an incorrect response), or c) did not align with any specified response category (i.e., random guess and therefore 25% chance that response is falsely classified as a correct response). The proportion of falsely classified correct responses of all correct responses in the control experiment was calculated from this categorization and amounted to 18.7% (SD=10%, 7% came from random guessing and 11.7% aligned with the correct response category). The amount of correctly identified and subsequently remembered Mooney images in the fMRI experiment was finally adjusted by this proportion.” (p.31f)

Nevertheless, we agree with the reviewer, that the categorization instead of identification of the Mooney object was not ideal even though due to technical constraints in the MR, there was no other way to get the subject’s response. Therefore, we have noted this technical caveat in the limitation section as follows:

“Firstly, the accuracy metric is not exact as participants could not enter specific names of identified objects during problem-solving due to technical constraints in the scanner. Nevertheless, we assessed the likelihood of incorrectly classified correct responses through a control experiment (18.7%, refer to Supplementary Material) and then adjusted the accuracy and subsequent memory measures accordingly for this proportion.” (p.24f)

2) Instead of the usual insight judgment procedure in which a participant judges a solution as having been derived by insight, analysis, or “don’t know/other), the authors tasked their participants with judging the amount of positive emotion, suddenness, and certainty. Unfortunately, no opportunity was given to subjects to respond “don’t know/other” which could have led to some guess or random

responding. Subsequently, they found that these 3 measures were correlated, so they summed them into a single insight measure anyway, suggesting that the three dimensions are parts of a single mechanism. (Is the distribution of the unitary insight variable bi-modal, which would suggest that the median split is not the right way to divide trials between high and low insight?) And yet, they subsequently break these measures apart to determine whether different brain regions underlie these dimensions. The logic of this procedure is murky.

We apologize for not being clearer about our approach. Some evidence suggests that it is reasonable to assume that the Aha experience is not a unitary, all-or-nothing phenomenon but a multifaceted construct that can be measured based on several individual dimensions. The most consistently used dimensions are suddenness, certainty, and positive emotion (Becker et al., 2024; Danek et al., 2017; Danek & Wiley, 2020; see also: Webb et al., 2016). This implies that we inherently assume the Aha experience is a combination of these individual scales. In the Supplement (**Relationship between different dimensions of insight experience**, see below), we demonstrate that these individual dimensions significantly positively load onto one latent factor, justifying why a) summing up the measures into one Aha measure is justified and b) why our assumption that these dimensions collectively form a single mechanism is empirically sound. Note that the distribution of the median split does not appear bimodal (see Fig. 1), however, as we will describe in further detail in response to #7, we only use the median split instead of the dimensional Aha (sum) variable for the connectivity/graph analysis because this cannot be done based on a continuous variable and for visualization purposes of the behavioral data. Please, also see response to #4 further below (*Concern of a non-monotonic Aha distribution*).

Figure. 1 – Distribution of insight-sum measure across all conditions

We report the effect of the individual scales for two reasons: a) because reviewer #3 specifically requested it, and b) reporting individual scales can still be meaningful as they may contribute differently to the overall Aha experience depending on the trial or brain region. For instance, it can be informative to see that the suddenness scale drives the insight*memory interaction in the hippocampus, consistent with its function as a match/mismatch detector (Axmacher et al., 2010). So during memory formation, it may be particularly the suddenness aspect of the Aha experience that explains the subsequent memory effect (in hippocampus).

The reasoning for using three scales and summing them up again as well as the reasoning for reporting just an individual scale was described in the method section:

“Assessing insight. Originally, insight was only binarized as present or absent (Bowden & Jung-Beeman, 2003; Jung-Beeman et al., 2004; Kounios & Beeman, 2014). However, **some evidence suggests** the overall subjective insight (AHA) experience is more continuous and can be decomposed into three different main dimensions: (1) positive emotional response or internal reward upon finding the solution, (2) experienced suddenness of the solution and (3) certainty about the solution (Danek & Wiley, 2017, 2020). To have better control over these three dimensions, we assessed them separately (see section: Mooney image paradigm), and then combined them into a compound insight

measure by adding up the three ratings into a continuous scale (3-12). The individual dimensions were only employed for exploratory investigations to determine which ones influence univariate activity in amygdala and hippocampus during both insight and insight-related memory processes. Analysing the data with the continuous insight variable for connectivity and graph analyses was not applicable. Hence, for those analyses, the continuous insight measure was median split into high (*HI-I*) and low (*LO-I*) insight trials.” (p.27)

“Relationship between different dimensions of insight experience

[...] The latent insight factor loaded significantly onto all three variables: *Certainty* ($\lambda=.617$, $z=23.70$; $p<.001$), *Emotion* ($\lambda=.650$, $z=25.17$; $p<.001$) and *Suddenness* ($\lambda=.531$, $z=20.31$; $p<.001$) suggesting that those variables contribute significantly to the latent insight factor. Practical fit indices (CFI= 1.0; RSME = 0.000; SRMR =0.000) suggested a good fit of the model to the data. Note, the significant factor loadings did not change significantly when including all trials (i.e. also incorrectly identified Mooney images). In sum, those results justified combining the three insight dimensions into one single measure for further analyses.

Figure S2. Measurement model for latent insight variable

Note. Asterisk = significant loadings at $p<.001$.” (p.2, Supplement)

Becker, M., Wang, X., & Cabeza, R. (2024). Surprise!—Clarifying the link between insight and prediction error. *Psychonomic Bulletin & Review*, 1-10.

Danek, A. H., & Wiley, J. (2017). What about false insights? Deconstructing the Aha! experience along its multiple dimensions for correct and incorrect solutions separately. *Frontiers in psychology*, 7, 2077.

Danek, A. H., & Wiley, J. (2020). What causes the insight memory advantage?. *Cognition*, 205, 104411.

Webb, M. E., Little, D. R., & Cropper, S. J. (2016). Insight is not in the problem: Investigating insight in problem solving across task types. *Frontiers in psychology*, 7, 1424.

3) On a related note, insights are often described in the literature as conferring a feeling of confidence. However, this is probably not true for all tasks. For example, for arithmetic tasks such

as (147 X 7,892), it is likely that a person would feel more confident in an analytically derived solution rather than a solution derived by insight. Are there any data to support the idea that perceiving Mooney objects with a unitary Aha experience leads to more confidence?

We are not aware of any study that has attempted to measure a unitary Aha experience in addition to its individual scales (suddenness, certainty, emotion, etc.) using Mooney images. However, as mentioned in response to #2, we conducted a measurement model in the Supplementary Material, demonstrating that all three dimensions, including certainty, significantly positively loaded onto a single "Aha" factor (see response to #1). This result was replicated in the study by Becker, Wang, & Cabeza (2024) using Compound Remote Associates (CRAs) and we also replicated those factor loadings with Mooney images in another online study (currently unpublished data).

Additionally, confidence generally appears to positively correlate with the Aha! experience in other studies. For instance, Danek et al. (2017) showed that for magic tricks, certainty significantly positively correlated with a unitary Aha! experience measure at $r=0.52$ (considering only correct responses, as in our analysis). Furthermore, Webb et al. (2016) compared Aha! ratings with their individual dimensions across classic insight tasks, non-insight tasks, and CRAs. They found that "Across problem types, Aha was significantly and positively related to Confidence, Pleasure, and Surprise (see Figure 2, see Supplementary Materials for correlation matrices). Confidence was the most strongly related to Aha! ratings across problem types, having a moderate to strong positive relationship with Aha! in insight and non-insight problems, and in CRAs" (Webb et al., 2016).

In summary, we have no reason to believe that perceiving Mooney objects with a unitary Aha! experience is not also associated with greater confidence.

Becker, M., Wang, X. & Cabeza, R. Surprise!—Clarifying the link between insight and prediction error. *Psychon Bull Rev* (2024). <https://doi.org/10.3758/s13423-024-02517-0>

Danek AH and Wiley J (2017) What about False Insights? Deconstructing the Aha! Experience along Its Multiple Dimensions for Correct and Incorrect Solutions Separately. *Front. Psychol.* 7:2077. doi: 10.3389/fpsyg.2016.02077

Webb ME, Little DR and Cropper SJ (2016) Insight Is Not in the Problem: Investigating Insight in Problem Solving across Task Types. *Front. Psychol.* 7:1424. doi: 10.3389/fpsyg.2016.01424

4) The methods section does not describe the task procedure in sufficient detail. The instructions given to subjects are not described. One concern is how the 4-point scale is described. Is "1" considered zero emotion, suddenness, or certainty? The concern is how the scale maps on to the unitary Aha experience. Does 1-4 refer to 0-100% Aha or 90-100% Aha? For example, does "1" refer to analytic judgements or insights of moderate strength? Do all subjects interpret the scale in the same way? And if the scale just refers to levels of insight and excludes pure analytic judgments, this procedure excludes the possibility of directly contrasting the end points of the scale. For example, what if the whole distribution is non-monotonic?

We appreciate the reviewer's comments and would like to clarify our approach to interpreting the Aha experience as a multifaceted phenomenon.

Instructions: We acknowledge that there may be some interindividual differences in how subjects interpret the individual scales. However, this is a common challenge across paradigms that assess the

Aha experience also as a unitary construct. To address this, we provided personalized oral instructions to each subject, clearly explaining what the Aha experience is and the meaning of each scale. We have now included this explanation in the manuscript as follows:

“Before entering the MRI scanner, participants were instructed orally and received three test trials to assure they understood the task and the concept of the Aha experience with its individual scales:

“When solving problems including identifying Mooney images, you may experience the solution in different ways. Sometimes, the solution comes to you in a sudden manner, with a strong sense of certainty and a strong positive emotion—this is what we commonly refer to as an Aha experience. However, the solution can also emerge more gradually, accompanied by little or no positive emotion and less certainty about its correctness. It’s important to note that these three aspects—suddenness, emotion, and certainty—may vary independently or together. So after identifying each Mooney image you are asked about the solution and subsequently HOW you experienced finding the solution: You will first be asked to rate the degree of suddenness with which you became aware of the solution, using a scale from 1 (indicating a gradual solution awareness) to 4 (indicating a sudden solution awareness). Next, you will be asked to assess your emotional response upon finding the solution, on a scale from 1 (no positive emotion) to 4 (strong positive emotion). Finally, you will rate your certainty about the correctness of the solution, on a scale from 1 (very uncertain) to 4 (very certain).” (p.28)

Interpretation of the individual scales: As indicated by prior research (e.g., Danek et al., 2017), the Aha experience is understood to have multiple dimensions (please refer to our response to #2). In our study, we interpret “analytical judgements” as instances lacking insightful judgments, without categorizing them as a distinctly different qualitative type (see Methods section: Assessing Insight, p.27f). Consequently, the lowest end of the individual Aha dimensions was coded to reflect the absence of certainty, positive emotion, and suddenness. To further clarify this in the manuscript, we have now included more detailed information regarding how these scales were presented to participants and how they were instructed to interpret them (see copy-pasted paragraph above p.28):

Concern of a non-monotonic Aha distribution: As we have demonstrated in response to #2, the distribution is not non-monotonic. Note, this distribution pattern is observed not only when summing the individual scales but also when analyzing the individual Aha dimensions for the same Mooney images. In a previous paper (see Figure below), we reported that when one sample of participants solved Mooney images and was asked about their overall (unitary) Aha experience, and a different sample solved the same images but rated their Aha experience on three distinct scales (certainty, suddenness, positive emotion), both distributions resulted in comparable means and were bell-shaped rather than non-monotonic.

Figure Redacted

Panel C from Figure 3 in Becker et al., 2023: Experiment (study) 1 (yellow): Distribution of scaled unique Aha experience; Experiment (study) 2 (purple) Distribution of scaled summed Aha experience assessed via individual scales from suddenness, certainty, positive emotion. Both Aha experience assessments stem from the same Mooney images but a different sample. The dashed line represents the mean average for experiment (study) 1 and the full line is the average for experiment (study) 2.

Becker, M., Yu, Y. & Cabeza, R. The influence of insight on risky decision making and nucleus accumbens activation. *Sci Rep* **13**, 17159 (2023). <https://doi.org/10.1038/s41598-023-44293-2>

5) Another concern is that the authors pick ROIs that mostly make sense, but shouldn't they compare these ROIs to other areas that, from known theory, should not be expected to play a role? And what if there are stronger results in areas other than the amygdala, hippocampus, and VOTC? Not knowing the answer to this question leaves open the possibility that the real action is elsewhere. Furthermore, Oh et al. (2020) showed a putative reward system response associated with insight in orbitofrontal cortex. Why not include OFC rather than, or in addition to, the amygdala? What about the ACC which has been implicated in past insight studies?

In response to the reviewer's request, we have added whole-brain activity analyses in the supplement to show correlations with the intensity of the Aha experience. However, we would like to stress that our study is fundamentally hypothesis-driven, a strength highlighted by the other reviewers. In that sense, our primary goal was not to replicate previous univariate studies on the Aha experience, but rather to propose a coherent mechanism for visual restructuring and identify brain regions associated with both the evaluative component of insight (Aha experience) and subsequent memory, as informed by prior research (e.g., Ludmer et al., 2011). However, this approach does not exclude the possibility of activation in other regions, particularly those linked to the reward system (as part of insight's evaluative component).

In summary, we did not find evidence that the "real action is elsewhere". We identified three clusters that positively correlate with the intensity of the Aha experience. The main cluster was located in hippocampus and amygdala including ventral striatum which reinforced our decision to originally hypothesize/prioritize these two areas. The second cluster was primarily in the precuneus consistent with semantic retrieval (Flanagin et al., 2023), and the third cluster involved the ACC and OFC, as suggested by the reviewer. These results are reported and briefly discussed in the Supplementary Material as follows:

"For completeness, we also report univariate whole-brain activity (see Supplementary Material, Fig. S8)." (p.12)

"Exploratory whole-brain analysis

For exploratory purposes, we also performed a whole-brain analysis to examine whether activity in other brain regions might be parametrically modulated by the intensity of the Aha! experience.

First-Level Analysis. We utilized the beta values previously estimated during the first-level analysis (as detailed in the ROI analysis, see Method section). For each participant, simple contrast t-images were generated from the beta weights associated with the onset regressors.

Second-Level Analysis. The contrast images representing the parametric modulation of the Aha! experience during the solution phase (button press) were analyzed using SPM's one-sample t-test.

Multiple comparisons were corrected by applying a voxel-level threshold of $p < .001$ (Eklund et al., 2016) and a cluster-level threshold of $p < .05$ (family-wise error corrected), and with a height threshold of $t = 3.40$. All anatomical regions were identified using the AAL3 atlas (Rolls et al., 2020), based on the percentage of voxels within an activated cluster from the one-sample t-test that intersected the respective anatomical region.

Whole-Brain Analysis Results: The intensity of the Aha! experience showed a positive correlation with three distinct clusters (extent threshold $k = 132$ voxels; see Fig. S8). The primary cluster (size: 1,081 voxels, $xyz[22, -8, -20]$) was located around bilateral hippocampus (covers 32% of this cluster) and extended to bilateral amygdala (11.5%), parahippocampal gyrus (12.5%), ventral striatum (7.5%), olfactory bulb (5.5%), and bilateral putamen (14%). The fact that the largest cluster appeared around the hippocampus and amygdala validates our previous hypothesis that these regions are particularly associated with insight's evaluative component.

The other two clusters were medially situated, with an anterior cluster (size: 341 voxels, $xyz[-2, 34, -8]$) encompassing the anterior cingulate cortex (70%) and bilateral medial orbital frontal cortex (14%). The ACC has been frequently linked to insight (Becker et al., 2021; for meta-analysis see Dietrich & Kanso, 2010;), and the OFC has also been reported as part of the reward network during insight (Oh et al., 2020). The posterior cluster (size: 458 voxels, $xyz[-14, -52, 36]$) included the bilateral precuneus (70%), bilateral posterior (15%) and middle (7.5%) cingulate cortex, and the left cuneus (2.5%). The involvement of the precuneus, which is associated with semantic retrieval (Flanagin et al., 2023), is plausible as participants were likely retrieving the semantic content of the object at that moment.

Becker, M., Kühn, S., & Sommer, T. (2021). Verbal insight revisited—Dissociable neurocognitive processes underlying solutions accompanied by an AHA! experience with and without prior restructuring. *Journal of Cognitive Psychology*, 33(6-7), 659-684.

Dietrich, A., & Kanso, R. (2010). A review of EEG, ERP, and neuroimaging studies of creativity and insight. *Psychological bulletin*, 136(5), 822.

Eklund, A., Nichols, T. E., & Knutsson, H. (2016). Cluster failure: Why fMRI inferences for spatial extent have inflated false-positive rates. *Proceedings of the national academy of sciences*, 113(28), 7900-7905.

Flanagin, V. L., Klinkowski, S., Brodt, S., Graetsch, M., Roselli, C., Glasauer, S., & Gais, S. (2023). The precuneus as a central node in declarative memory retrieval. *Cerebral Cortex*, 33(10), 5981-5990.

Oh, Y., Chesebrough, C., Erickson, B., Zhang, F., & Kounios, J. (2020). An insight-related neural reward signal. *NeuroImage*, 214, 116757.

Rolls, E. T., Huang, C. C., Lin, C. P., Feng, J., & Joliot, M. (2020). Automated anatomical labelling atlas 3. *Neuroimage*, 206, 116189.

Figure S8. Whole-Brain Activation positively correlated with intensity of Aha! experience

Note. Threshold for visualisation: voxel-based threshold $p < .001$, cluster-threshold $p\text{-FWE} < .05$.” (p.11ff, Supplementary Material)

6) Pre and post solution intervals were divided according to the time of the button press. However, we know from the Jung-Beeman et al. (2004) and Oh et al. (2020) EEG studies that an insight solution occurs 300-500 ms before such a button press because, once a solution has been derived, it takes time to program and execute the button press. This, together with the limited temporal resolution of fMRI makes the division between pre- and post-solution potentially blurry.

For the univariate analyses in the hippocampus and amygdala, we focused on BOLD activity at the moment of the solution, which is why we modeled only the solution phase (button press). In response to the reviewer’s comment, we re-ran the analyses using onsets 400 ms before the button press, yielding very similar (no significantly different) results. To maintain consistency with the multivariate analyses, we have chosen to report the original analysis and results.

Regarding the multivariate analyses, we concur with the reviewer that the temporal resolution of fMRI generally poses a limitation, and the precise moment of solution awareness might precede the button press by 300-500ms, leading to potential blurring between pre- and post-solution phases. However, we argue that this does not compromise our findings.

First, in our analysis, we included only trials where pre- and post-solution events were separated by at least 2 seconds but had failed to mention this in the manuscript previously, which we now do report as follows: “Additionally, response time (RT) and run order were added as covariate of no interest and we included only trials where pre- and post-solution phases were at least 2 seconds apart to reduce overlap of the hemodynamic response.” (p.38)

Second, the moment of button press (solution awareness) is comparable between high and low insight trials (approximately 400ms in both conditions). However, we still observed a significantly stronger increase in representational strength (measured by second-order correlation between the brain and model RDM) for high compared to low insight trials. Furthermore, this effect can also not be explained away with potential time-on-task effects as already discussed during the last revision round (see also response to #R2-3). First, because differences in solution time are not pertinent to multivariate analyses, as discussed in Dimesdale-Zucker & Ranganath (2018) and we also did not find consistent effects of time-on-task in our two VOTC areas (see Supplement: “**Univariate fMRI results of VOTC RC areas**”). Moreover, we controlled for solution time as additional regressor of no interest in all of these analyses to ensure robustness of our results.

Finally and related to the second point, if it would be true that the observed increase in representational strength from stimulus onset to solution button press in high insight trials might be influenced by a “blurry” temporal difference between stimulus onset and solution time, we would expect a correlation between representational strength and solution time. Using a linear mixed model, we assessed the effect of solution time on representational strength, including subject and item as random effects and trial order as a covariate of no interest. Our analysis revealed no significant effect of solution time on representational strength ($\beta = .02$; $p = .23$; 95% CI: [-.02, .07]). [note, we had already done a similar analysis in the Supplement, see Figure S6]. This signifies that we find no evidence that representational strength is systematically affected by the “blurry” temporal difference between stimulus onset and solution time.

In sum, while it is true that the difference between pre- and post-solution phases may be somewhat blurry due to the sluggishness of the BOLD signal, we have no reason to believe that this has systematically biased our results between high and low insight trials.

7) The low n per subject per condition is concerning. Judging by Figure S5, for some subjects, the n per cell was extremely low, even 0. I found the authors’ response to the other reviewer insufficient. I would want a more detailed statistical justification for retaining subjects whose n per cell is low or zero.

We believe there may be some misunderstanding regarding the low trial count concern, and we would like to clarify this issue in more detail. There are three instances of analyses, each involving a different number of trials, which therefore raise different concerns about potential false positives or reduced power: 1) all insight analyses, 2) all insight-memory analyses (except connectivity/graph analyses) and as a subset here 3) insight-memory related connectivity/graph analyses. As we will show below, 1) and 2) raise little/no concern due to either sufficient trials or applied bootstrapping/permutation tests

not significantly changing the results. Merely, 3) is of justified concern but no consistent results are reported/interpreted for this subset of analyses.

1) All insight analyses

Sufficient number of trials for insight analyses: As shown in Figure S5, the insight trials generally have an adequate number of trials per participant. While one individual shows fewer trials for correct low-insight trials, we do not view this as problematic since Linear Mixed Models (LMMs) are well-equipped to handle such variability (discussed below). Furthermore, we do not analyze incorrect trials, ensuring that no subject has a trial count of zero. The low trial count issue primarily affects the insight-memory trials, not the insight-memory trials (see below).

Use of dimensional insight measure: In response to feedback from reviewers 2 and 3, we shifted from using the median-split insight measure (as depicted in Figure S5) to using the dimensional insight sum measure (which combines certainty, suddenness, and positive emotion, ranging from 3 to 12). This approach eliminates the need to split insight values into high and low categories, which would have halved the number of trials per condition. This change applies to all analyses except for some behavioral analyses (for visualization purposes) and the connectivity/graph analyses (due to current limitations in CONN regarding continuous variables).

Use of Linear Mixed Model: We employed LMMs at the single-trial level, which offer the following advantages:

1. *Random Effects:* LMMs incorporate random effects, allowing them to account for variability at different levels (e.g., subject- and item-level differences).
2. *Handling Unbalanced Data:* LMMs are designed to manage unbalanced datasets, where trial numbers per condition vary. Unlike traditional ANOVA, LMMs can accurately estimate fixed effects even when some subjects contribute more data in one condition than another.
3. *Increased Statistical Power:* By modeling both fixed and random effects, LMMs enhance statistical power, even with fewer trials per condition. This is because LMMs utilize all available data, including variability across subjects, to produce more reliable estimates.

Baayen, R. H., Davidson, D. J., & Bates, D. M. (2008). *Mixed-effects modeling with crossed random effects for subjects and items*. *Journal of Memory and Language*, 59(4), 390-412.

Barr, D. J., Levy, R., Scheepers, C., & Tily, H. J. (2013). *Random effects structure for confirmatory hypothesis testing: Keep it maximal*. *Journal of Memory and Language*, 68(3), 255-278.

2) All insight-memory analyses (except connectivity/graph analyses)

As shown in Figure S5, the trial counts for memory analyses are reduced, though not to the extent the figure suggests, due to our use of the dimensional insight measure and linear mixed models (LMMs) that account for unbalanced and lower trial counts. To further ensure the robustness of our statistical inferences, we conducted permutation tests for all brain-derived insight-memory analyses and employed a bootstrapping approach for the behavioral insight-memory analysis. It was necessary to use a different simulation technique for the behavioral data because we used a binomial model to fit the data to memory (due to the binary dependent variable), while the permutation technique was

only applicable to linear mixed models with a ratio-scaled variable. Importantly, none of the bootstrapped or permuted results changed significantly, confirming that our findings were not driven by false-positives due to lower trial counts.

2a) Behavioral Insight-Memory Effects:

“Due to the relatively lower trial count for the insight-memory analyses (see Fig. S5), we used a non-parametric bootstrapping approach with 1,000 resamples to estimate the odds ratios and their 95% confidence intervals for the effects of interest. This was implemented using the `boot` function (v-1.3-30) in R, where the model was refitted to each resampled dataset providing robust estimates of effect size and confidence intervals. (p.32)

“Insight predicts better subsequent memory

To investigate behavioral insight-related better memory, we estimated the effect of insight on subsequent memory using two general linear mixed (binomial) models including random subject and item intercepts and the respective run order (1-4). The first model predicted subsequent memory including a categorical insight factor consisting of not solved trials, correctly solved HI-I and LO-I trials. This insight factor significantly predicted variance in subsequent memory ($Chi^2(2)=666.49, p<.001$, see Fig.2-A and B). HI-I (OR [odds ratio]=7.78 95%CI[6.42, 9.53]) and LO-I (OR=3.72, 95%CI[3.12, 4.44]) trials were significantly better remembered compared to unsolved trials. Critically, HI-I trials were also significantly more likely to be remembered compared to LO-I trials (OR=2.10, 95%CI[1.80, 2.45]), confirming insight-related better memory.

The second model was identical to the first model but included a continuous (excluding unsolved trials) instead of a binarized insight variable additionally controlling for solution time. Insight still significantly predicted subsequent memory ($Chi^2(1)=76.34, p<.001, OR=1.47, 95%CI[1.33, 1.61]$) (see Fig.2-C). This result was replicated in a behavioral control experiment (see Supplement and Fig.S7).” (p.7)

2b) Brain-related Insight-Memory Effects:

Method section for multivariate fMRI data analysis:

“To enhance statistical robustness for the insight-memory analyses (see Fig. S5), we implemented permutation tests to derive p-values from comparing nested random effects models (`permlmer`) using the `predictmeans` package (v.1.1.0) in R, with 999 permutations (Lee & Braun, 2012). (p.40)

Method section for univariate fMRI data analysis

“To enhance statistical robustness for insight-related memory on brain activity, p-values from comparing nested models IV-VI were obtained from permutation tests (n=999) using the `predictmeans` package (v.1.1.0) (Lee & Braun, 2012). (p.41)

Note, we decided to use the `predictmeans` package in R for the permutations, because it allowed a very simple implementation of exactly the same linear mixed models that we had already used before using the `lme4` package.

“Hypothesis 4: VOTC-RC and hippocampal activity predict insight-related better subsequent memory

Having linked RC in VOTC and amygdala as well as hippocampal activity to the visual insight process, we examined whether the underlying neural correlates of both insight components also predicted better subsequent memory. To enhance statistical robustness, p-values were obtained through permutation tests (n=999) (Lee & Braun, 2012).

Insight-related RC predicts better subsequent memory

[...]

1) RC from pre to post solution: Multivoxel pattern similarity. There was an overall memory effect ($Chi^2(1)=14.40, p=.001, \beta=.10$) controlling for insight as well as an insight-memory interaction effect ($Chi^2(1)=9.27, p=.003, \beta=.07$) for pre to post solution activity pattern similarity for both RC ROIs. The significant three-way interaction between insight, memory and ROI ($Chi^2(3)=11.48, p=.011$) indicated differences between both ROIs in insight-related better memory. Posthoc analyses revealed that pFusG showed a main effect for memory ($Chi^2(1)=3.75, p=.048, \beta=.06$) when controlling for insight and a significant insight-memory interaction ($Chi^2(1)=5.84, p=.014, \beta=.07$) (see Fig. 5-A). Similarly, iLOC showed a memory effect ($Chi^2(1)=8.74, p=.005, \beta=.12$) and also an insight-memory interaction effect ($Chi^2(1)=4.08, p=.038, \beta=.07$) (see Fig. 5-A).

2a) RC from pre to post solution: Representational strength - AlexNet. Using the AlexNet model to compute representational strength, there was no overall memory effect over both time points controlling for insight ($Chi^2(1)=1.35, p=.231, \beta=-.02$) but there was a significant memory*time interaction ($Chi^2(1)=50.84, p=.001, \beta=.24$) controlling for insight. However, there was no evidence for a insight*memory*time interaction ($Chi^2(1)=0.71, p=.401, \beta=-.03$) (see Fig.6-B). Furthermore, the four-way interaction with ROI was also not significant ($Chi^2(7)=4.04, p=.79$) suggesting no evidence for a difference in this insight*memory*time relationship between the ROIs.

2b) RC from pre to post solution: Representational strength - Word2Vec. There was an overall memory effect over both time points ($Chi^2(1)=10.27, p=.002, \beta=.07$) controlling for insight and a significant time*memory interaction ($Chi^2(1)=9.82, p=.004, \beta=.10$). Importantly, the three-way insight*memory*time interaction showed a significant effect ($Chi^2(2)=10.46, p=.004, \beta=.09$) (see Fig.6-D). Furthermore, the four-way interaction with ROI was not significant ($Chi^2(7)=6.71, p=.45$) suggesting no evidence for a difference in this insight*memory*time relationship between the ROIs.

Insight-related anterior hippocampal activity predicts better subsequent memory.

[...]

There was no overall effect of memory ($Chi^2(1)=0.17, p=.67, \beta=.01$) controlling for insight but a significant insight*memory interaction ($Chi^2(1)=8.06, p=.006, \beta=.06$) for all three regions. The three-way interaction between insight*memory*ROI ($Chi^2(6)=31.08, p<.001$) indicates differences between the ROIs regarding this interaction. Posthoc analyses revealed that amygdala showed no evidence for an insight*memory interaction effect ($Chi^2(1)=1.01, p=.321, \beta=.04$). Anterior hippocampus showed a significant an insight*memory interaction ($Chi^2(1)=5.73, p=.016, \beta=.09$). Posterior hippocampus showed a trend for an insight*memory interaction ($Chi^2(1)=3.12, p=.087, \beta=.07$) (see Fig.6-C).

For exploratory purposes, we investigated which insight dimension drives this insight*memory interaction in anterior hippocampus. For this, we repeated the above mentioned mixed effects single trials analysis, substituting the insight measure with either certainty, suddenness or emotion. Only the interaction between suddenness and memory ($t(2962.2)=2.69, p\text{-Bonferroni}=.027, \beta=.10$) predicted

significant variance in anterior hippocampus while no evidence for an interaction between memory and emotion ($p>.268$) or memory and certainty ($p>.096$) was found.” (p.17f)

Insight-memory analyses for connectivity/graph analyses

We fully agree with the reviewer that the low trial count might reduce statistical power for insight-memory analyses in the connectivity/graph measures and therefore poses a justified concern for false positives. In this case, we could not use the dimensional insight scale or LMMs (neither is implemented in CONN) and the low trial count here led to smaller reliability and subsequently a smaller power. However, the risk of reporting false positive results is minimal, as no consistent results were observed nor interpreted:

“For reasons of completeness, we also explored whether FC and graph theoretical measures also predicted insight-related better memory. However, these results provided no consistent evidence” (p.19)

Nevertheless, we have noted this limitation in the manuscript as follows:

“Secondly, the limited number of trials in the memory condition, particularly for forgotten trials, may reduce statistical power of our results and increase interindividual variability. To mitigate this, we conducted permutation tests for all brain-derived insight-memory analyses, except for connectivity/graph analyses, where this method was not available.” (p.25)

We hope this clarification and particularly our simulations (bootstrapping & permutation tests) addresses the reviewer’s concerns regarding trial count in our analyses.

8) Fig. 2D includes an obvious outlier that seems to boost the correlation coefficient. What is the correlation coefficient without the outlier?

We thank the reviewer for pointing this out. After excluding the outlier, the Pearson correlation coefficient is $r=0.521$ with $p < 0.003$, indicating that the correlation remains significant. Additionally, we re-analyzed the data including the outlier using Spearman's rank correlation, which does not assume normality. The results show that the correlation still remains significant with $p=0.633$ and $p<0.001$. We changed the results in the manuscript as follows (note we decided to use the Spearman’s rank correlation to be able to include all data points):

“To assess this relationship, we computed Spearman’s rank correlation between the participants’ average continuous insight experiences in Mooney images and an anagram task (see Fig.2-D). The obtained correlation was statistically significant ($p=.633$, $p<.001$), indicating that the insight experience evoked by Mooney images is comparable to that of another verbal insight task.

Figure 2. Behavioral Insight Memory Effect.” (p.8)

Overall, this study is on the right track. If these points can be adequately addressed, then I would support publication. I’ve read through Reviewer #2’s feedback and the authors’ responses. I will address these points according to the numbering in the authors’ responses.

R2-1. The low trial counts in particular conditions are a significant concern. They state, “that low trial counts would be more likely to increase type II error rather than type I error, leading to an absence of a consistent memory effect in the connectivity/graph analyses.” The phrase “more likely” is not reassuring. I would want something more convincing, such as simulations to show the likely effects. I would also want a statistician to review this issue.

We have addressed this issue in our response to Comment #7 and have now included simulations for the subset of analyses where the trial count is sufficiently low to warrant this additional analysis, i.e. all insight-related memory analyses.

R2-2. Regarding the effects of having subjects classify, rather than identify, the objects in the Mooney images, this was not an optimal choice for a procedure. The authors’ control experiment gives some reassurance. However, it does not fully reassure the reader in light of this non-optimal procedure choice.

We have addressed this issue in our response to comment #1.

R2-3. Regarding potential time-on-task effects on the BOLD signal, the authors’ fixes help, but may not be sufficient. A low RT cutoff of 1.5 seconds seems somewhat arbitrary and may not be enough. The authors say that “we removed fast high-insight trials and slow low-insight trials until the time taken to solve both high and low insight trials was no longer significantly different ($p > 0.20$).” Here,

statistical significance is uninformative. Just because the RT difference between HI and LI is now nonsignificant doesn't mean that there isn't enough of a difference to cause a BOLD time-on-task confound. Simulations could be helpful in clarifying this issue.

We agree with the reviewer that only showing that the RT difference between HI and LO is non-significant is an indicator but not yet enough evidence that there is no BOLD time-on-task confound. However, this is not the only analysis that we had run to control for this issue. Here, we would like to explain our previous control analyses regarding this issue to demonstrate that we have tackled this time-on-task confound in several different ways by now:

- 1) First, regarding the "somewhat arbitrary" 1.5sec RT cutoff: This cutoff was determined via a control experiment, and describes the point in time at which one can speak of an insight process (5 SD above the average object recognition time of the real-world images of the Mooney images). For a more detailed description of how this threshold is determined, see the Supplement: **Estimating threshold for insight trials based on object recognition time** (p.3)
- 2) Aligning the response times between the HI and LO insight trials was implemented as a variation to address the potential time-on-task issue. Importantly, in our analyses, we used a dimensional insight variable for all uni- and multivariate insight-related analyses (with the exception of the connectivity/graph analyses where time-on-task is less relevant due to the consistent 10-second time window used). In these uni- and multivariate analyses, we included RT as a covariate to account for its specific influence on the BOLD signal. Our results remained consistent, indicating that insight-specific effects were not merely due to RT. The reasoning is that if time-on-task significantly affected the BOLD signal, this effect would be captured by the RT variable. Consequently, any remaining significant variance in the BOLD signal after accounting for RT is attributed to insight itself. However, our analyses consistently demonstrated that insight explained additional variance beyond what is attributed to RT.
- 3) Multivariate analysis: Moreover, the influence of time-on-task on 2nd order correlations in multivariate analyses is unclear (Dimsdale & Zucker et al., 2018). However, we still addressed a potential concern in the Supplementary Material in two ways (p.5f):
 - a. First we estimated the influence of solution time on the univariate BOLD signal in iLOC and pFusG (the two areas showing consistent RC effects): "Moreover, we examined potential time-on-task influences on the BOLD signal in both VOTC-RC regions (Yarkoni et al., 2009). To explore this, we replicated the single-trial analyses from above, excluding insight as a predictor. Instead, we computed a solution time*ROI interaction, controlling for run order and incorporating random subject and item intercepts. We found a significant solution time*ROI interaction ($Chi^2(1)=100.38$, $p<.001$, $\beta=-.20$). Posthoc analyses revealed a positive effect of solution time on BOLD activity in iLOC ($z=2.87$, $p=.004$) but a negative effect of solution time on BOLD activity in pFusG ($z=-12.11$, $p<.001$). In summary, the inconsistent univariate effects observed in iLOC and pFusG in relation to insight and solution time make it implausible to attribute the observed consistent multivariate RC effects in both VOTC regions on univariate brain activity.

Figure S6. Relationship between univariate BOLD activity in RC VOTC areas and insight. ” (p.5f, Supplement)

- b. Second, to account for the potential impact of BOLD activity on the relationship between insight and multivariate measures due to potential time-on-task effects, we conducted a repetition of the aforementioned multivariate RC analyses. This time, we directly incorporated BOLD activity during solution (i.e. its beta estimate) as a covariate of no interest in exchange for solution time additionally controlling for run (order) and random subject and item intercepts. Even with this adjustment, the relationship between insight and 2nd order correlations (multivoxel pattern similarity and representational Strength using AlexNet and Word2Vec) remained significant and unchanged (for a detailed description please refer to the Supplement, p.9ff).

In sum, all of our control analyses consistently indicate that while reaction time (RT) systematically differs between higher and lower AHA tasks, there remains significant unexplained variance in the brain signal that clearly cannot be attributed solely to RT (time-on-task) differences. Instead, this variance is explained by insight, even when controlling for RT (or time-on-task via the beta estimates of the BOLD signal). Nonetheless, we have acknowledged the potential time-on-task confound in the limitations section to ensure full transparency. However, we believe that the coherence across all these control analyses provides strong support for our conclusions.

R2-4. Rev #2 makes good points about restructuring. In particular, on some trials subjects may be, to coin a term, “structuring” rather than restructuring if they are going from no structure to some structure. Thus, the findings might show a mixture of structuring and restructuring, which would still be fine because it would not negate the authors’ general point. I’m OK with the authors’ handling of this issue.

Thank you.

The rest of Rev #2’s points mostly focus on the number of datapoints per condition. Again, this is a concern, and I would want to see simulations evaluated by a statistician to determine the level of concern.

We have addressed this issue in our response to comment #1.

Overall, considering Reviewer #2’s points and my own review, my feeling is that this is an ambitious study that addresses an important point. However, there are many issues, some small and some not so small, that individually may not be deal-killers, but that together give this study a “death of a thousand cuts” feel. In an ideal world, it would be great if the authors could refine the design and procedure and rerun the study with a large enough sample size that they could drop subjects who do not provide enough data points per condition, etc. But I recognize that we do not live in a

research environment with unlimited resources and time. Therefore, the options are to reject the paper and hope for better in the future or to publish a version of this paper that highlights the necessary caveats. My feeling is that it would be better to publish a version (that addresses my separate review) with caveats as a way to stimulate additional research along these lines.

We would like to thank the reviewer for their thorough effort in not only reading the paper but also carefully considering the extensive comments and our corresponding responses to Reviewer 2, as well as for the thoughtful insights provided in their own review. We believe that we have successfully addressed all concerns raised. Specifically:

1. Regarding the issue of categorization versus identification of the solution: At the group level, we were able to quantify the bias (number of false positives) precisely but this noise remains at the single-trial level, which is why we have acknowledged this in the limitations.
2. We clarified that not all analyses suffer from low trial counts and conducted simulations (permutation tests/bootstrapping) to demonstrate that the low trial count in the insight-memory analyses do not introduce bias, and all effects remained significant.
3. We have described the Aha experience as a dimensional, multifaceted construct and clarified how the ends of the respective scales should be interpreted.
4. We had already addressed the potential time-on-task confound in detail in our previous response and have also described several control analyses in the Supplementary Materials to tackle this issue while additionally mentioning it in the limitations.
5. Finally, we now provide and interpret exploratory whole-brain activations during insight despite our hypothesis-driven approach.

In sum, we appreciate the reviewer's careful consideration and believe our revisions adequately resolve the concerns raised.

Reviewer #5 (Remarks to the Author):

Overall, this study brings novel and interesting results, that are of interest for a wide community. The methods seem appropriate, including bootstrapping and permutation testing. However, the conducted analyses are particularly difficult to understand, and yet, I am familiar with all of them. To understand the results, I had to carefully read the methods, which were not truly insightful (if I may) and led me to actually have a look at the open scripts used to obtain the results. I can confirm that the scripts are very clear and well organized, and the results obtained with the scripts match the ones reported in the manuscript. In my opinion, a reader should not have to go into deep details of the methods and scripts to actually grasp the results, especially in a journal such as Nature Communication. One main problem in the report of the results is that the authors do not explain which analyses they conducted and report results with confusion between model comparison results (all the chi2 statistics) and the value of the fixed effect coefficients (odds-ratio). Moreover, some phrasings do not exactly match the results, and some conceptual aspects of the scientific questions are quite vague. I have detailed below each of these aspects that seriously lower the overall quality of the manuscript.

1. Conceptual aspects

1.1 There are not a lot of studies addressing representational changes during insight but the authors should at least the ones that address this question, such as Bieth et al 2024 (<https://www.nature.com/articles/s44271-024-00100-w>).

We previously cited Bieth et al. (2024; originally referenced as Bieth et al., 2021 in our submission, as only the preprint was available when we submitted in September 2023, prior to the official publication) in the context of networks, which is the primary strength of Bieth's study:

“Insight-related integration of the solution network

[...] Those results suggest that during insight, solution-relevant information was more efficiently processed, as has been proposed before for the creative process (Beaty et al., 2018; Mednick, 1962) and particularly for insight (Bieth et al., 2024; Durso et al., 1994; Luchini et al., 2023; Schilling, 2005).” (p.22)

We are, however, cautious in citing Bieth et al. (2024) specifically in the context of restructuring. In our opinion, their study does not provide a convincing measure of restructuring, as the lack of an expected three-way interaction — $\text{isSolved} = \Delta\text{Metric} * \text{impact rating} * \text{semantic distance}$ — limits their ability to show that solvers connect semantically distant concepts that are also solution-relevant more effectively than non-solvers, which is fundamental to the understanding of restructuring in the context of the representational change theory (Ohlsson, 1984).

At any rate, we have ensured coverage of restructuring studies across theoretical, behavioral, and neuroimaging contexts. Theoretical perspectives are represented by Ohlsson (1984) and Schilling (2005), while behavioral studies include Durso et al. (1994) and Knoblich et al. (1999). For neuroimaging evidence, we cited Becker, Sommer et al.

(2020a) and Tang et al. (2015). We have also added two additional restructuring studies by Öllinger et al. (2014), Patrick et al. (2015) and Luo & Knoblich (2007) and emphasized more which other studies have previously addressed restructuring in the discussion:

“The main cognitive component is associated with *representational change* (RC) whereby internal (conceptual or perceptual) representations of the solution are reorganised and integrated into a coherent whole (Ohlsson, 1984; Schilling, 2005). This process is assumed to involve a change in the pattern of currently activated cell assemblies coding the problem in domain-specific cortex but neuroscientific evidence is currently lacking (Becker et al., 2024; Knoblich et al., 1999). As an example, in verbal problem-solving, RC may encompass semantic reinterpretation or regrouping conceptual relationships (Becker, Sommer, et al., 2020a; Durso et al., 1994; Patrick et al., 2015). In contrast, visual problems that demand object recognition, RC may involve breaking up perceptual chunks into their constituent elements to make them available for a novel recombination (Tang et al., 2015; Knoblich et al., 1999; Öllinger et al., 2014).” (p.2)

“*Cognitive component of insight: Representational change in VOTC*

While insight-related RC has been examined behaviorally (Knoblich et al., 1999; Öllinger et al., 2014) and through univariate BOLD activity (Tang et al., 2015; Luo & Knoblich, 2007), this study presents the first multivariate evidence.” (p.21)

“Luo, J., & Knoblich, G. (2007). Studying insight problem solving with neuroscientific methods. *Methods*, 42(1), 77-86.” (p.53)

“Öllinger, M., Jones, G., & Knoblich, G. (2014). The dynamics of search, impasse, and representational change provide a coherent explanation of difficulty in the nine-dot problem. *Psychological research*, 78, 266-275.” (p.54)

“Bieth, T., Kenett, Y., Ovando-Tellez, M., Lopez-Persem, A., Lacaux, C., Oudiette, D., & Volle (2024). Changes in semantic memory structure support successful problem-solving and analogical transfer. *Commun Psychol* 2, 54. <https://doi.org/10.1038/s44271-024-00100-w>.” (p.47)

“Patrick, J., Ahmed, A., Smy, V., Seeby, H., & Sambrooks, K. (2015). A cognitive procedure for representation change in verbal insight problems. *Journal of Experimental Psychology: Learning, Memory, and Cognition*, 41(3), 746.” (p.54)

1.2 In the introduction, the authors state that “Amygdala activation may reflect the positive emotions (Janak & Tye,2015), and the hippocampus activation, the sudden arrival and/or surprising content of the solution.”. In my opinion, this is reverse inference and that should be avoided.

We have attenuated the phrasing to reduce the impression of reverse inference as follows: “Activation in the amygdala has been suggested to play a role in processing positive emotions (Janak & Tye, 2015), and the detection of novelty or surprise, such as during the sudden arrival or unexpected content of a solution, has been associated with hippocampal activation.” (p.3)

1.3 In the introduction, the authors state their hypotheses and mention a “solution process”, I don’t see to which process they are referring to, which make their hypotheses vague when we read them and try to make sense of them.

We agree with the reviewer that our writing was unclear, and we have now specified what we mean by solution process as follows:

“In sum, insight during problem solving likely reflects stronger multivariate pattern changes associated with enhanced solution-relevant representations. This increased RC (or conceptual update) is efficiently integrated into the solution process – a series of cognitive steps from problem representation and active visual or memory search to subsequent solution retrieval (Ohlsson, 1984) – leading to awareness of the solution. This realization is often accompanied by an emotional and suddenness/surprise response in hippocampus and amygdala resulting in enhanced encoding of the solution and better subsequent memory.” (p.3)

1.4 When the fourth hypothesis is announced, cognitive and evaluative components are mentioned but it is difficult to immediately match cognitive with RC and evaluative with hippocampus (maybe the authors could phrase it differently, or at least add between parentheses “cognitive” and “evaluative” after RC and and hippocampal activity respectively.

We changed the phrasing as follows:

“Finally, we hypothesized that *both insight components (cognitive and evaluative) are associated with better subsequent memory*. In particular, we expected that RC (cognitive) in VOTC and hippocampal activity (evaluative) predict *insight*-related better memory.” (p.5)

2. Results

2.1 The whole analysis logic should be better reported in the entire results section. I provide the detail of the very first behavioral result to illustrate how hard it was to understand the very simple analysis they conducted. My argument is valid for the vast majority of their results. I must insist on the fact that it’s only by reading their script that I understood what they did. We can read “Consistent with previous results, this insight-accuracy effect was significant ($\chi^2(1)=94.98$, $p<.001$, odds ratio=1.31)”. At this point, we have absolutely no clue about which analysis is conducted here. In the methods, there is nothing about that “insight-accuracy” effect. Only in the script we can read:

####correlation between accuracy and insight ####

```
data_new = PPSS[ PPSS$ROI == 'l_Amygdala' & PPSS$tp == 1 & (PPSS$RT>=1.5 & PPSS$RT<=9.5),]
```

```
M0_cor2 <- glmer(cor2 ~ +sessionblock + (1|ID) + (1|Item),data= data_new,family = binomial(link = "logit"), na.action = na.omit)
```

```
M1_cor2 <- glmer(cor2 ~ insight_sum +sessionblock + (1|ID) + (1|Item),data= data_new,family = binomial(link = "logit"), na.action = na.omit)
```

```
anova(M0_cor2, M1_cor2)
```

```
tab_model(M1_cor2, show.std = T)
```

With that, we finally understand that what they did is a model comparison, with one model regressing accuracy against either only the session block or the session block and the insight measure. Then we understand that the chi2 statistics is the result of the model comparison, suggesting that a model with the insight measure better captures accuracy than a null model. The odds ratio corresponds to the odds ratio of the regression estimate of the insight measure in the winning model. I think it is crucial that the authors clearly explain the statistics they report.

We agree with the reviewer that our analytic approach was often unclear and we now describe the analysis logic of model comparison at the beginning of the results section, including what the reported effect sizes correspond to:

“All reported inferential statistics (χ^2) are based on nested model comparisons between two (general) linear mixed models where a baseline model is compared to a full model incorporating the independent variable of interest. For testing interaction effects, the baseline model additionally incorporates two independent variables, while the full model adds their interaction. Effect sizes—either odds ratios (OR) for binary outcomes or standardized beta estimates (β) for continuous outcomes—correspond to the regression estimate of the respective independent variable or interaction between two or three independent variables in the full model.” (p.5f)

With that, we now describe the results for the insight-accuracy effect as follows:

“Of all correctly identified Mooney images, 65.3% were solved with *HI-I* and 34.7% with *LO-I* trials ($SD=8\%$). The full model including insight and run order as predictor for accuracy along with random item and subject intercepts was significantly better than the baseline model without insight ($\chi^2(1)=94.98, p<.001, OR=1.31$). This suggests that accuracy for *HI-I* was significantly higher than for *LO-I*, consistent with previous results (Becker, Wiedemann, et al., 2020; Danek & Salvi, 2020).” (p.6)

Additionally the difference in response time between *HI-I* and *LO-I* trials is now described as follows:

“The median response time for correct solutions was 3.5sec ($SD=0.8$ sec) where participants were faster during *HI-I* (2.7sec, $SD=0.81$ sec) than *LO-I* solutions (5.1sec, $SD=1.12$ sec). This difference was significant ($\chi^2(1) = 423.8, p<.001, \beta = -0.22$), as shown by the significantly improved fit of the full model—which included insight and run order as predictors for logged response time, along with random subject and item intercepts—compared to the baseline model without insight.” (p.6)

We also adjusted the phrasing for the fMRI results where this analysis logic applies:

“Hypothesis 1: VOTC exhibits visual insight-related RC

[...] *RC from pre to post solution: Multivoxel pattern similarity.* To identify ROIs that showed the hypothesised RC related changes in the correlation between pre- and post solution activity patterns, we estimated the effect of insight in the above named VOTC regions on this correlation and their interaction. This was done using nested model comparisons: the baseline model included random subject and item intercepts, controlling for response time and run order, while two full models further included insight and the interaction between insight and ROI, respectively. [...]

“RC from pre to post solution: Representational strength - AlexNet. We restricted the representational strength analyses to areas (iLOC, pFusG) that showed the hypothesised greater change in pre- to post solution activity patterns. To identify areas where the representational strength increased more strongly from pre to post solution for correct insight (see Fig.3-B), we performed a nested model comparison. The baseline model included insight, time (pre vs. post), response time, run order and random subject and item intercepts. The full models further included an insight*time or insight*time*ROI interaction, respectively.

The model with the insight*time interaction performed significantly better ($\chi^2(1)=15.19, p<.001, \beta=.06$) than the model without this two-way interaction indicating an overall insight*time interaction over both regions. Furthermore, the model including the three-way insight*time*ROI interaction did not explain representational strength (RC) significantly better than a model without ROI as an additional interaction term ($\chi^2(3)=1.79, p=.62$), suggesting no differences between the two ROIs. [...]

RC from pre to post solution: Representational strength - Word2Vec. To ensure that the interaction results obtained via the AlexNet model are robust and generalize to other conceptual models, we also measured RC using a purely verbal conceptual model – Word2Vec (Mikolov et al., 2013) and conducted the same nested model comparisons as in the analyses with AlexNet. As displayed in Fig.3-B, the results largely replicated those found with AlexNet. The model with the two-way insight*time interaction ($Chi^2(1)=16.12, p<.001, \beta=.06$) performed significantly better than a model without this interaction indicating an insight effect over both regions.” (p.10f)

“Hypothesis 2: Amygdala and hippocampus show insight-related activity

[...] To assess whether amygdala and hippocampal activity varies parametrically with the strength of insight, we compared a baseline model including a random subject intercept and controlled for run order and the specific ROI with a full model additionally including insight. Furthermore, we compared this full model with a third model additionally including an interaction term between insight and ROI. For completeness, we also report univariate whole-brain activity (see Supplementary Material, Fig. S8).

Activity in amygdala and hippocampus is associated with insight

As illustrated by Fig.4, the full model including insight performed significantly better than the baseline model without insight ($Chi^2(1)=47.56, p<.001, \beta=.09$) implying a positive relationship between brain activity and insight over all three ROIs. Furthermore, the model with the insight*ROI interaction performed better than the model without the interaction term ($Chi^2(2)=26.91, p<.001$) implying differences between the ROIs in this condition. [...]

Activity in amygdala and hippocampus during solution is associated with insight - control analysis in subject space

To exclude potential normalisation artefacts due to the small volume of the amygdala and hippocampal ROIs, we additionally conducted another ROI analysis in subject space. Univariate activity during solution, i.e. onset of the button press, was parametrically modulated by insight and solution time and estimated using GLMs as implemented in SPM (Ashburner, 2012). To test whether amygdala and hippocampal activities parametrically vary depending on the strength of insight, we computed two models, the baseline model including a random subject intercept and controlling for the run order and the full model additionally including the respective ROI as predictor.

The intercept of the baseline model was significant suggesting that bilateral amygdalar and hippocampal activity are significantly parametrically modulated by insight during (correct) solution ($t(159.9)=3.93, p<.001$). The full model performed significantly better than the baseline model suggesting differences between the ROIs ($Chi^2(2)=28.59, p<.001$). However, post-hoc analyses showed that the intercept in the baseline model was significant regardless of whether amygdala activity ($t(121.7) = 5.21, p<.001, \beta=.18$), anterior ($t(121.9) = 3.24, p=.002, \beta=.10$), or posterior hippocampus ($t(120.4)=4.37, p<.001, \beta=.14$) was modeled. This suggests that all three ROIs are significantly modulated by insight in a parametric manner.

Activity in amygdala and hippocampus during solution is predominantly associated with insight dimensions certainty and emotion

To explore the factors influencing the impact of insight in the hippocampus and amygdala, we delved into the three dimensions of insight. [...] For this, we repeated the above reported mixed effects single-trial analysis (standard space), substituting the insight measure with certainty, suddenness, and emotion and report the t-test performed on their individual regression weights.” (p.12ff)

“Hypothesis 4: VOTC-RC and hippocampal activity predict insight-related better subsequent memory

[...] *Insight-related RC predicts better subsequent memory*

First, we tested whether insight-related better memory for correctly solved trials is associated with our two consecutive measures of RC, i.e. 1) the changes of multivoxel activity patterns (i.e., decrease in multivoxel pattern similarity) and 2) an increase in representational strength of the object from pre to post solution in those regions that elicit RC (pFusG, iLOC). To test insight-related better memory for 1), we compared a series of nested models that included insight (continuous variable), memory (remembered vs. forgotten), and ROI, along with interactions among these predictors. For 2), we performed a similar series of nested model comparisons as in 1), but also incorporated time (pre- vs. post-solution) and its interactions with the other predictors of interest. All models additionally included random subject and item intercepts controlling for run order and solution time (see Methods section for details).

1) *RC from pre to post solution: Multivoxel pattern similarity.* The model that included memory, controlling for insight, showed a significantly better fit than the model without memory indicating an overall memory effect ($Chi^2(1)=14.40$, $p=.001$, $\beta=.10$). Additionally, the model with a two-way interaction between insight and memory for pre to post solution activity pattern similarity across both RC ROIs, performed significantly better than the model without this interaction ($Chi^2(1)=9.27$, $p=.003$, $\beta=.07$). Finally, the model including a three-way interaction between insight, memory and ROI provided a significantly better fit than the model excluding it ($Chi^2(3)=11.49$, $p=.011$) indicating differences in insight-related better memory across the two ROIs [...]

2a) *RC from pre to post solution: Representational strength - AlexNet.* When predicting representational strength computed via AlexNet, the model including memory as predictor did not fit significantly better than the model without memory, suggesting no overall memory effect over both time points controlling for insight ($Chi^2(1)=1.35$, $p=.231$, $\beta=-.02$). However, the model with a memory*time interaction ($Chi^2(1)=50.84$, $p=.001$, $\beta=.24$), controlling for insight, performed significantly better than the model without this interaction. There was no evidence, though, for an insight*memory*time interaction ($Chi^2(1)=0.71$, $p=.401$) (see Fig.6-B). [...]

2b) *RC from pre to post solution: Representational strength - Word2Vec.* The model that included memory fit significantly better than the model without memory, indicating an overall memory effect over both time points when controlling for insight ($Chi^2(1)=10.33$, $p=.002$, $\beta=.07$). Additionally, there was a time*memory interaction ($Chi^2(1)=9.82$, $p=.004$, $\beta=.10$). Importantly, the model with the three-way insight*memory*time interaction also showed a significantly better fit than the model without this three-way interaction ($Chi^2(2)=10.46$, $p=.004$) (see Fig.6-D). [...]

Insight-related anterior hippocampal activity predicts better subsequent memory.

We tested whether insight-related better memory is associated with amygdala and hippocampal activity. For this, we used the data from the single-trial analyses containing a beta estimate of univariate amygdala, anterior and posterior hippocampal activity (ROI) for the solution button press at each correctly solved trial. A series of nested models were compared, regressing the beta estimates onto insight (continuous), memory (*Remembered, Forgotten*), ROI as well as an interaction between variables, while also controlling for response time, run order and random subject and item intercepts (see Methods section for details).

The model including memory did not significantly outperform the model without it suggesting no overall effect of memory when controlling for insight ($Chi^2(1)=0.17$, $p=.67$, $\beta=.02$). However, the model which included an insight*memory interaction provided a significantly better fit than the model without this interaction ($Chi^2(1)=8.06$, $p=.006$, $\beta=.06$) across all three regions. Additionally, the better fit of the model with the three-way insight*memory*ROI interaction ($Chi^2(6)=31.08$, $p<.001$), compared to the two-way insight*memory interaction, suggests that this interaction varied by ROI. Post-hoc analyses indicated no

evidence for an insight*memory interaction in the amygdala ($Chi^2(1)=1.00$, $p=.334$, $\beta=.04$), a significant interaction in anterior hippocampus ($Chi^2(1)=5.68$, $p=.015$, $\beta=.09$), and a trend in the posterior hippocampus ($Chi^2(1)=3.07$, $p=.087$, $\beta=.07$) (see Fig.6-C).

For exploratory purposes, we investigated which insight dimension drives this insight*memory interaction in anterior hippocampus. For this, we repeated the above mentioned mixed effects single trial analysis with the insight*memory interaction, substituting the insight measure with either certainty, suddenness or emotion. Only the regression weight of the interaction between suddenness and memory ($t(2962.2)=2.69$, $p\text{-Bonferroni}=.028$, $\beta=.10$) predicted significant variance in anterior hippocampus while no evidence for an interaction between memory and emotion ($p>.268$) or memory and certainty ($p>.096$) was found.” (p.17ff)

2.2 In the section “Insight predicts better subsequent memory”, the authors indeed explain that they conducted two linear mixed models: “To investigate behavioral insight-related better memory, we estimated the effect of insight on subsequent memory using two general linear mixed (binomial) models including random subject and item intercepts and the respective run order (1-4). The first model predicted subsequent memory including a categorical insight factor consisting of not solved trials, correctly solved HI-I and LO-I trials.” They explain the first model, not the second, and then jump to the result of the insight factor. The methods state:

I. Subsequent memory ~ run + (1 | ID) + (1 | item) + ϵ

II. Subsequent memory ~ run + insight + (1 | ID) + (1 | item) + ϵ

Here, we understand that the first model is a control model, and that the second model includes the insight factor. We understand that the model comparison demonstrated that the second model was better than the first one such that the insight factor is needed to better explain the memory effect. This model comparison result is reported as “(Chi²(2)=666.49, $p<.001$ ”. Then, odds ratios are reported without any indication of which analysis they correspond to. By looking at the script, I understand that it corresponds to the contrast of the regression estimate of HI-I versus unsolved and LO-I versus unsolved.

We thank the reviewer for pointing out this inconsistency, what we meant by “model” was in fact a “model comparison”. We have rephrased the Results section as “Insight predicts better subsequent memory” where we describe the analysis logic more coherently now. Note, as the analysis logic of model comparisons is already laid out at the beginning of the results section, we did not describe it here again, but it should be clear now what is meant by the baseline and full model and what the effect size estimates [OR and β] refer to (see response to #2.2.):

“To investigate behavioral insight-related better memory, we estimated the effect of insight on subsequent memory using two binomial linear mixed model comparisons with a respective full model including insight and run order as predictor along with random item and subject intercepts and a corresponding baseline model without insight. For the first model comparison, the full model predicted subsequent memory including a categorical insight factor consisting of not solved trials, correctly solved HI-I and LO-I trials. The full model was significantly better than the baseline model indicating that the insight factor significantly predicted variance in subsequent memory ($Chi^2(2)=666.49$, $p<.001$, see Fig.2-A and B). Post-hoc analyses revealed that HI-I (OR[odds ratio]=7.78, 95%CI[6.42, 9.53]) and LO-I (OR=3.72, 95%CI[3.12, 4.44]) trials were significantly better remembered compared to unsolved trials. Critically, HI-I trials were also significantly more likely to be remembered compared to LO-I trials (OR=2.10, 95%CI[1.80, 2.45]), confirming insight-related better memory.

The second model **comparison** was identical to the first one with the exception that **the full model included** a continuous (excluding unsolved trials) instead of a binarized insight variable **predicting subsequent memory while** additionally controlling for solution time. **The full model was significantly better than the baseline model indicating that** insight still significantly predicted subsequent memory ($Chi^2(1)=76.34$, $p<.001$, $OR=1.47$, 95%CI[1.33, 1.61]) (see Fig.2-C). This result was replicated in a behavioral control experiment (see Supplement and Fig.S7).” (p.7f)

2.3 “the model with the three-way insight*time*ROI interaction did not explain significant variance in representational strength” : please update that sentence as it makes no sense when we don’t know which analysis is conducted. Here, I assume that the authors wanted to say that a model with the three-way interaction did not explain RC better than a model without it.

We changed the sentence as follows to clarify that the statistics from the three-way interaction come from a model comparison where the model with the insight*time*ROI interaction was tested against a model with the insight*time interaction as described in the Method section (p.40):

“Furthermore, the model including the three-way insight*time*ROI interaction did not explain representational strength (RC) significantly better than a model without ROI as an additional interaction term ($Chi^2(3)=1.79$, $p=.62$), suggesting no differences between the two ROIs.” (p.11)

2.4 Hypothesis 2: “there was a main effect for condition” > but what is condition? From the script I understand it’s insight.

The original sentence read:

“As illustrated by Fig.4, there was a main effect for condition ($Chi^2(1)=47.56$, $p<.001$, $\beta=.09$) implying a positive relationship between brain activity and insight over all three ROIs.” suggesting that insight is the condition being tested.

To clarify this, we replaced "condition" with "insight":

“As illustrated by Fig.4, **the full model including insight performed significantly better than the model without insight** ($Chi^2(1)=47.57$, $p<.001$, $\beta=.09$) implying a positive relationship between brain activity and insight over all three ROIs. **Furthermore, the model with the insight*ROI interaction performed better than the model without the interaction term** ($Chi^2(2)=26.91$, $p<.001$) implying differences between the ROIs in this condition.” (p.12f)

2.5 “Posthoc analyses revealed that activity in amygdala activity ...” > please double check the text. “hippocampus significantly predicted insight” > If the dependent variable is activity, then it’s the insight that “predicts” activity, not the reverse. Or the analysis should be done the other way around. In any case the use of the term “predict” suggest too much causality, here, it’s a relationship that has been found.

We thank the reviewer for pointing this out and changed the “direction” of the prediction as well as the phrasing:

“Post-hoc analyses revealed that activity in amygdala activity ($z=7.55$, $p<.001$, $\beta=.18$) and anterior ($z=5.57$, $p<.001$, $\beta=.13$) but not posterior ($z=1.11$, $p=.27$, $\beta=.03$) hippocampus were significantly **associated with** insight (see Fig.4).” (p.13)

2.6 I have small doubts about the degrees of freedom, as the ones in the script seem to not match the manuscript in some cases. To be double-checked.

Reviewer #5 (Remarks on code availability):

The results are reproducible and the code is usable. I would just double-check the degrees of freedom reported in the manuscript as they don't match the ones in the script. I'm not sure why.

Yes, it is possible for the degrees of freedom (DF) to vary slightly when running linear mixed models with lmer in R on different days or different machines, primarily due to the way lmer handles DF estimation for fixed effects and some potential differences in computational environments. First, lmer uses different methods for approximating the DF for fixed effects (e.g., Satterthwaite, Kenward-Roger, or others, depending on the package setup and options). These methods involve iterative computations and numerical approximations, which can vary slightly due to machine-specific floating-point precision or software environment differences. Second, minor updates in the lmerTest or lme4 packages, or even in R itself, may change default settings or improve DF approximations. These updates can impact the exact DF obtained when rerunning the same model on different days if software was updated in the meantime. For this reason, we have described the package versions in the Method section. Finally, linear mixed models, especially those with complex random effects structures, may yield slightly different solutions if the starting points for the optimization algorithms differ due to random seeds or hardware floating-point precision. Although these differences are generally very small, they can sometimes impact the DF. To help minimize these variations, we have now set a random seed at the beginning of the code (`set.seed(123)`) and rerun the analyses to make sure those are the final results (at least on our machine with our package versions) and noted the slight changes accordingly (in yellow, please see manuscript).

We thank the reviewer for the detailed comments, which have greatly helped us clarify our methods. We believe the paper is much improved thanks to the reviewer's comments.